# Nondeterministic Polynomial-time Problem Challenge: An Ever-Scaling Reasoning Benchmark for LLMs

**Chang Yang**[1*]**, Ruiyu Wang**[2*]**, Junzhe Jiang**[1]**, Qi Jiang**[3]**, Qinggang Zhang**[1]**, Yanchen Deng**[4]**, Shuxin Li**[4]**, Shuyue Hu**[5]**, Bo Li**[1]**, Florian T. Pokorny**[2]**, Xiao Huang**[1]**, Xinrun Wang**[6†]

[1] *The Hong Kong Polytechnic University,* [2] *KTH Royal Institute of Technology,* [3] *Carnegie Mellon University,* [4] *Nanyang Technological University,* [5] *Shanghai Artificial Intelligence Laboratory,* [6] *Singapore Management University*

*chang.yang@connect.polyu.hk, ruiyuw@kth.se, xrwang@smu.edu.sg*

**Reviewed on OpenReview:** *https://openreview.net/forum?id=Xb6d5lGLb2*

## Abstract

Reasoning is the fundamental capability of large language models (LLMs). Due to the rapid progress of LLMs, there are two main issues of current benchmarks: i) these benchmarks can be *crushed* in a short time (less than 1 year), and ii) these benchmarks may be easily *hacked*. To handle these issues, we propose the **ever-scalingness** for building the benchmarks which are scaling over complexity against crushing, instance against hacking and exploitation, oversight for easy verification, and coverage for real-world relevance. This paper presents Nondeterministic Polynomial-time Problem Challenge (**NPPC**), an ever-scaling reasoning benchmark for LLMs. Specifically, the **NPPC** has three main modules: i) *npgym*, which provides a unified interface of 25 well-known NP-complete problems and can generate any number of instances with any levels of complexities, ii) *npsolver*, which provides a unified interface to evaluate the problem instances with both online and offline models via APIs and local deployments, respectively, and iii) *npeval*, which provides the comprehensive and ready-to-use tools to analyze the performances of LLMs over different problems, the number of tokens, the reasoning errors and the solution errors. Extensive experiments over widely-used LLMs demonstrate: i) **NPPC** can successfully decrease the performances of advanced LLMs to below 10%, demonstrating that **NPPC** is not crushed by current models, ii) DeepSeek-R1, Claude-3.7-Sonnet, and o1/o3-mini are the most powerful LLMs, where DeepSeek-R1 can outperform Claude-3.7-Sonnet and o1/o3-mini in most NP-complete problems considered, and iii) the numbers of tokens in the advanced LLMs, e.g., Claude-3.7-Sonnet and DeepSeek-R1, are observed first to increase and then decrease when the problem instances become more and more difficult. Through continuously scaling analysis, **NPPC** can provide critical insights into the limits of LLMs' reasoning capabilities, exposing fundamental limitations and suggesting future directions for further improvements.

## 1 Introduction

The remarkable successes of Large Language Models (LLMs) (Achiam et al., 2023) have catalyzed the fundamental shift of artificial intelligence. Recent breakthroughs on reasoning (Guo et al., 2025) enable LLMs to complete complex tasks, e.g., math proof, code generation and computer use, which require the capabilities of understanding, generation and long-term planning. Various benchmarks, e.g., GPQA (Rein et al., 2024), AIME, SWE-bench (Jimenez et al., 2024) and ARC-AGI (Chollet, 2019), are proposed to evaluate these advanced reasoning capabilities, where most benchmarks are curated and verified by human researchers with a finite number of questions. These benchmarks guide the directions for advancing LLM capabilities.

---

[*]Equal contribution
[†]Corresponding author

Current benchmarks face two fundamental challenges that limit their effectiveness for LLM evaluation. First, current benchmarks can be *crushed* in a short time: GSM8K (Cobbe et al., 2021) performance increased from approximately 35% to 95% within three years, while SWE-bench (Jimenez et al., 2024) scores improved from 7.0% to 64.6% in merely eight months, as illustrated in Figure 1. This rapid saturation suggests that these benchmarks quickly lose their discriminative power as models advance. Second, current benchmarks can be easily *hacked* or *exploited*. Static benchmarks are susceptible to data contamination and memorization issues, leading to overfitting rather than genuine capability assessment (Wu

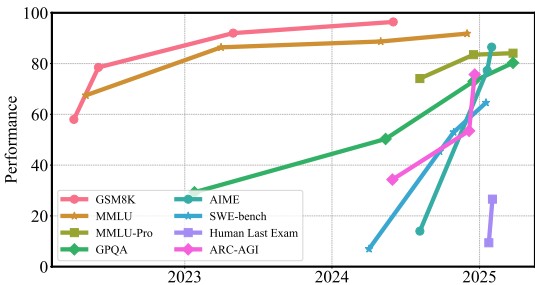

Figure 1: Crush of benchmarks

et al., 2025; Xu et al., 2024). While live benchmarks such as LiveCodeBench (Jain et al., 2025) address contamination by continuously introducing new problems, they require substantial ongoing human curation efforts. Similarly, human evaluation platforms like ChatbotArena (Chiang et al., 2024) incur significant costs (approximately $3,000 per evaluation) and remain vulnerable to strategic manipulation where MixEval (Ni et al., 2024) can achieve comparable correlation with human judgment at under $1 per evaluation. These limitations represent significant obstacles for reliable evaluation of the rapidly evolved LLMs.

To address these issues, we propose the **ever-scalingness** with four desiderata for a benchmark (as shown in Figure 2): i) *scaling over complexity* – the benchmark can generate the problems with continually increasing complexities to avoid the crushing of the benchmarks. This ensures that as LLMs improve, the benchmark remains challenging by providing harder problems that push the boundaries of current capabilities for long-term differentiation. ii) *scaling over instance* – the benchmark can generate an infinite number of instances to avoid the exploitation and overfitting. This prevents LLMs from memorizing specific problem instances during training and ensures that the evaluation truly measures the reasoning capabilities of LLMs. iii) *scaling over oversight* – the benchmark can verify the correctness of the solutions efficiently

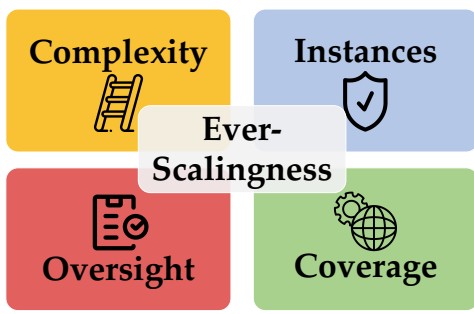

Figure 2: Desiderate of ever-scalingness

for the problems with any complexity. This is crucial because as problem complexity increases, manual verification becomes impractical or impossible. Automated and reliable verification enables evaluation at scale and ensures that harder problems can be verified efficiently, making the benchmark practically sustainable. iv) *scaling over coverage* – the benchmark should comprehensively cover problem types that are highly relevant to real-world applications, rather than focusing on puzzles. or rare edge cases. This ensures that performance improvements on the benchmark translate to practical value, measuring capabilities that matter for downstream tasks and real-world deployment rather than narrow, specialized skills. These four desiderata for ever-scalingness ensure continuous differentiation among LLMs over extended periods, identifying fundamental limitations for further improvement.

To construct the ever-scaling benchmark, we focus on nondeterministic polynomial-time (NP) problems whose solutions can be verified in polynomial time (Cormen et al., 2022). Specifically, we target on NP-complete (NPC) problems, i.e., the most computationally challenging problems in the NP class, for three key reasons. First, NPC problem instances can be systematically generated across arbitrary difficulty levels through controlled parameters, e.g., numbers of variables, enabling precise scaling of both complexity and instance. Second, NPC problems are intrinsically **"difficult to solve, easy to verify"**, i.e., no polynomial-time algorithms have been discovered for solving NPC problems, making them computationally intractable even with specialized tools, while their solutions remain efficiently verifiable. Problems in the P complexity class can be solved in polynomial time. When LLMs are equipped with code execution capabilities, they can generate and execute algorithms to solve these problems directly. Therefore, such benchmarks become susceptible to trivial solutions through computational tools rather than genuine reasoning. Conversely, NP-hard problems, particularly those lacking polynomial-time verification procedures, present the challenge for verifying the solutions for large-scale problem instances. Third, NPC problems demonstrate broad applicability, including

diverse real-world scenarios (e.g., routing (Toth & Vigo, 2002) and protein folding (Crescenzi et al., 1998)) and various puzzles, e.g., Sudoku (Seely et al., 2025). The theoretical foundation for using NPC problems as a comprehensive evaluation framework stems from the fundamental property that any NP problem can be reduced to an NPC problem in polynomial time, establishing NPC problems as a theoretically grounded, universal framework for computational problem-solving assessment. Therefore, NPC problems are the foundation problems of all computational problems and LLMs are the foundation models for wide range tasks, thus leading to our ever-scaling nondeterministic polynomial-time problem challenge (**NPPC**) (Figure 4(a)).

Specifically, **NPPC** has three main modules: i) *npgym*, which provides a unified interface of 25 well-known NPC problems and can generate any number of instances with any levels of complexities, which implies the ever-scalingness of **NPPC**, ii) *npsolver*, which provides a unified interface to evaluate the problem instances with both online and offline models via APIs and local deployments, respectively, to facilitate users to evaluate their own models and iii) *npeval*, which provides comprehensive and ready-to-use tools to analyze the performances of LLMs over different problems, the number of tokens, the "aha moments", the reasoning errors and the solution errors, which can provide in-depth analysis of the LLMs and the insights to further improve the LLMs' reasoning capabilities. Extensive experiments over widely-used LLMs, i.e., GPT-4o-mini, GPT-4o, Claude-3.7-Sonnet, DeepSeek-V3, DeepSeek-R1, and OpenAI o1-mini, demonstrate: i) **NPPC** can successfully decrease the performances of advanced LLMs to below 10%, demonstrating that **NPPC** is not crushed by current LLMs, ii) DeepSeek-R1, Claude-3.7-Sonnet, and o1/o3-mini are the most powerful LLMs, where DeepSeek-R1 can outperform Claude-3.7-Sonnet and o1-mini in most NP-complete problems considered, and iii) the numbers of tokens in the advanced LLMs, e.g.. Claude-3.7-Sonnet and DeepSeek-R1, are observed to first increase and then decrease when the problem instances become more and more difficult. We also analyze the typical reasoning errors in the LLMs, which provide the insights of the fundamental limitations of current LLMs and suggest the potential directions for further improvement. To the best of our knowledge, **NPPC** is the first ever-scaling benchmark for reliable and rigorous evaluation of the reasoning limits of LLMs, according to the four desiderata defined previously.

## 2 Related Work

Traditional benchmarks are typically curated by human with static datasets. Abstraction and Reasoning Corpus (ARC-AGI)-1 (Chollet, 2019) is designed to be "easy for humans, hard for AI", which is formed by human-curated 800 puzzle-like tasks, designed as grid-based visual reasoning problems. o3 at high compute scored 87% on ARC-AGI-1 (OpenAI, 2025), which roughly crushes the ARC-AGI-1 benchmarks and leads to the emergence of the ARC-AGI-2 benchmark. This pattern exemplifies a fundamental challenge with traditional benchmarks for LLMs, including MMLU (Hendrycks et al., 2021), GPQA (Rein

Table 1: Comparison of different reasoning benchmarks according to the ever-scalingness.

| | Complexity | Instance | Oversight | Coverage |
|---|---|---|---|---|
| NPHardEval (Fan et al., 2024) | ✗ | ✗ | ✓ | ✗ |
| ZebraLogic (Lin et al., 2025) | ✓ | ✗ | ✓ | ✗ |
| Reasoning Gym (Stojanovski et al., 2025) | ✗ | ✓ | ✓ | ✓ |
| Sudoku-Bench (Seely et al., 2025) | ✗ | ✓ | ✓ | ✗ |
| ARC-AGI-1 & 2 (Chollet, 2019) | ✗ | ✗ | ✗ | ✗ |
| **NPPC** (this work) | ✓ | ✓ | ✓ | ✓ |

et al., 2024), GSM8K (Cobbe et al., 2021), and SWE-bench (Jimenez et al., 2024), where static benchmarks are systematically solved within relatively short periods (as shown in Figure 1). Therefore, researchers have to continuously either develop new benchmarks, e.g., MMLU-Pro (Wang et al., 2024b) and SuperGPQA (Du et al., 2025), or regularly update with new datasets and problems, e.g., LiveCodeBench (Jain et al., 2025) and SWE-bench-Live (Zhang et al., 2025). However, these remedies rely on extensive human efforts to maintain their relevance and difficulty and cannot fully address the crushing issue of benchmarks.

Several recent benchmarks consider either NP(C) problems, e.g., 3SAT (Balachandran et al., 2025; Hazra et al., 2024; Parashar et al., 2025), or partially the ever-scalingness (Fan et al., 2024; Stojanovski et al., 2025) (displayed in Table 1). NPHardEval (Fan et al., 2024) considers 3 problems from P, NPC and NP-hard classes and use these class to evaluate the LLMs. We note that the problems in P class can be solved by augmenting the LLMs with tools, e.g., code running, and the NP-hard problems cannot be verified efficiently, therefore, NPHardEval cannot scale over the scalable oversight. Only 3 NPC problems are considered, i.e., Knapsack, Traveling salesman problem (TSP) and graph coloring, and the instances of each problem in NPHardEval are

finite and only regularly updated, which cannot scale over the instance and complexity. ZebraLogic (Lin et al., 2025) considers one logic puzzle, i.e., Zebra puzzle, to test the reasoning capabilities of LMs when the problems' complexities increase. However, the reasoning capability on specific puzzles does not necessarily transfer to other problems, which violates the scaling of the coverage. Sudoku-Bench (Seely et al., 2025) focuses on one specific Sudoku game with 2765 procedurally generated instances with various difficulty levels. Reasoning Gym (Stojanovski et al., 2025) is an ongoing project which collects the procedural generators and algorithmic verifiers for infinite training data with adjustable complexity. Though with some NP(C) problems, e.g., Zebra puzzles and Sudoku, the reasoning gym does not specifically focus on NPC problems and cannot meet the desiderata of ever-scalingness. Several recent work leverages NP(C) problems as the training environments to improve the general reasoning capabilities of LLMs (Zeng et al., 2025; Li et al., 2025; Liu et al., 2025a), which demonstrates the advantages of NP(C) problems in terms of controllable difficulty and efficient solution verification. These existing efforts highlight the need for a systematic approach for training and benchmarking LLMs grounded in the complexity theory. This motivates the formulation of ever-scalingness and the development of NPPC.

## 3 Preliminaries

**P and NP Problems.** The problems in P class are decision problems that can be solved in polynomial time by a *deterministic Turing machine*, which implies there exists an algorithm that can find a solution in time proportional to a polynomial function, e.g., $O(n^k)$, of the input size $n$. Examples include sorting, shortest path problems, and determining if a number is prime. The problems in NP class are decision problems that can be solved in polynomial time by *nondeterministic Turing machine*, where a proposed solution can be easily verified, though finding that solution might require more time (as displayed in Definition 1). All P problems are also in NP, but the reverse remains an open question, known as "P vs. NP problem". NP problems form the cornerstone of computational complexity theory, for which solution verification is tractable (polynomial time) even though solution discovery may be intractable (potentially exponential time), i.e., "difficult to solve, easy to verify". Many real-world optimization problems can be formulated as NP problems, such as equilibrium finding in game theory, portfolio management, network design and machine learning.

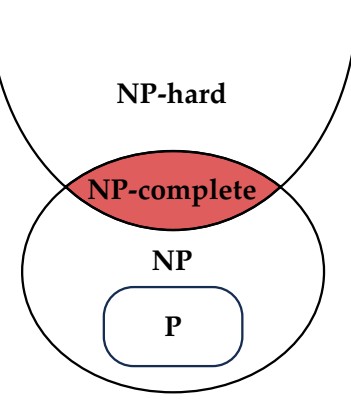

Figure 3: Complexity classes

**Definition 1** (NP Problems). The complexity class NP consists of all decision problems $\Omega$ such that for any "yes" instance $I$ of $\Omega$, there exists a certificate $\sigma$ of polynomial length in $|I|$ where a deterministic Turing machine can verify in polynomial time that $c$ is a valid certificate for $I$.

**NP-complete (NPC) Problems.** Formally, a problem $\Omega$ is an NPC problem if i) the problem is in NP, and ii) any NP problems can be transformed to problem $\Omega$ in polynomial time. This reducibility property establishes NPC problems as the "hardest" problems in NP class. The Cook-Levin theorem established SAT as the first proven NPC problem (Cook, 2023; Karp, 2009), while 3SAT is the special case of SAT and is also an NPC problem. Subsequent NPC problems typically proven via reduction chains back to 3SAT or other established NPC problems. The most well-known NPC problems include vertex cover problem, clique problem, traveling salesman proble (TSP), Hamiltonian path/cycle problem, etc. NPC problems play the most important roles in answering the "P vs. NP problem", i.e., if any NPC problem were shown to have a polynomial-time algorithm, then P = NP. However, despite decades of research, no polynomial-time algorithms for any NPC problem is discovered, which implies that NPC problems are computationally intractable by current methods. While NP-hard problems represent a broader class that includes optimization variants and potentially harder problems, they are less suitable for benchmarks because their solutions cannot be verified in polynomial time, which fundamentally limits the scaling of complexity and oversight of the benchmarks.

**Reasoning in LLMs.** The reasoning ability of LLMs refers to the model's capacity to tackle complex problems, e.g., mathematical proof, code generation through multi-step thinking and context understanding. Recently, specialized reasoning models have been proposed. OpenAI-o1 is an LLM trained with reinforcement learning (RL), which enables the model to perform complex reasoning, including logical thinking and problem

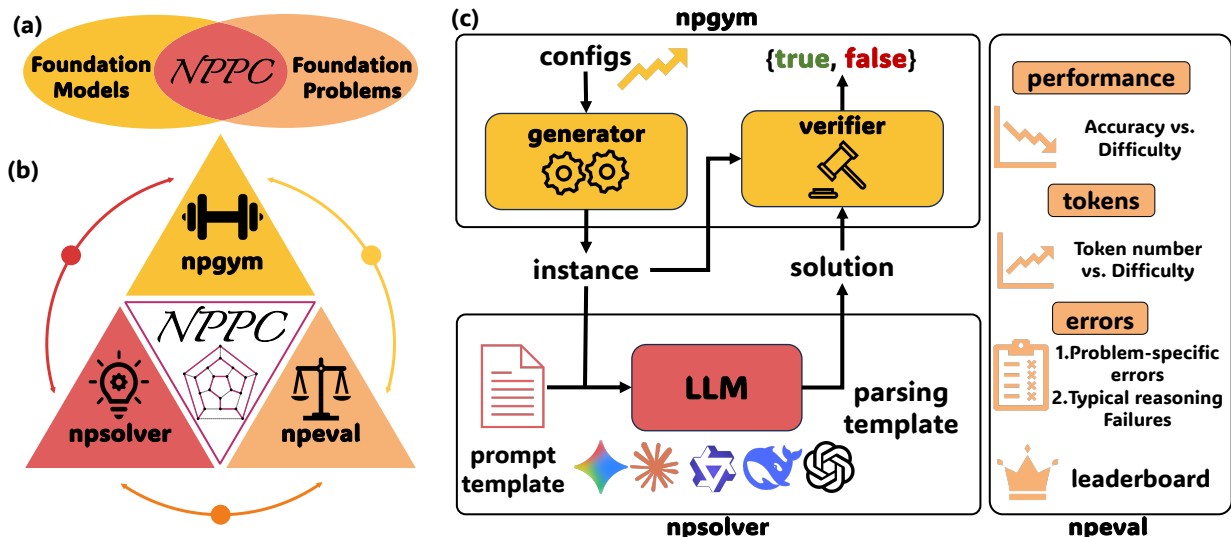

Figure 4: Overview of **NPPC**. **(a) NPPC** represents the intersection of foundation models and foundation problems. **(b)** The three main components of **NPPC**: *npgym* (problem generation), *npsolver* (solution generation), and *npeval* (evaluation). **(c)** Workflow diagram: *npgym* configuring generators and verifiers, *npsolver* using LLMs to generate solutions, and *npeval* measuring performance metrics.

solving, via chain-of-thought (CoT). o1 thinks before it answers and can significantly outperform GPT-4o on reasoning-heavy tasks with high data efficiency. DeepSeek-R1 (Guo et al., 2025) is an enhanced reasoning model designed to improve LLMs' reasoning performance that incorporates multi-stage training and cold-start data before the large-scale RL. DeepSeek-R1 demonstrates remarkable reasoning capabilities, and achieves comparable performance to OpenAI-o1 across various reasoning tasks, e.g., mathematical problems, code generation, and scientific reasoning. Additionally, there are open-sourced medium-sized LLMs with strong reasoning capabilities, e.g., DeepSeek-R1-32B, a distilled version of DeepSeek-R1, and QwQ-32B.

## 4 Nondeterministic Polynomial-time Problem Challenge

We introduce Nondeterministic Polynomial Problem Challenge (**NPPC**), an ever-scaling reasoning benchmark for LLMs. There are three main components in **NPPC** (as displayed in Figure 4(b)): i) *npgym*, which provides a unified interface of **25** well-known NPC problems and can generate any number of instances and verify the solution with any levels of complexities, ii) *npsolver*, which provides a unified interface to evaluate the problem instances with both online and offline models via APIs and local deployments, respectively, to facilitate the users to evaluate their own models and iii) *npeval*, which provides the comprehensive and ready-to-use tools to analyze the performances of LLMs over problems, the number of tokens, the "aha moments", the reasoning and solution errors, providing the in-depth analysis of the LLMs' reasoning capabilities.

### 4.1 Problem Suite: *npgym*

**Interaction Protocol.** Typically, NPC problems are the decision problems where given the instance $I$, the answer is "Yes" or "No". However, the LLMs may take a random guess without reasoning for the true solution (Fan et al., 2024). Therefore, we consider a more challenging setting: *given the instance $I$, the LLM needs to generate the solution s for the instance.* This setting will enforce the LLMs to reason for the correct solutions and the **NPPC** needs to provide the certificate $\sigma$ to verify the solutions generated by the LLMs. *npgym* provides a unified interface of NPC problems to interact with LLMs. The interaction between *npgym* and the LLM is displayed in Figure 4(c). *npgym* generates the instance $I$ with the given configuration, and the LLM receives the instance and generate the solution $s$, then the solution is verified by *npgym* with the output {**true**, **false**}. The representation of problem instances is designed to be concise and complementary to include all necessary information for the LLMs to reason for the solution.

**Core Problems and Extension.** There are 25 typical NPC problems implemented in *npgym*. Among all NPC problems, 12 typically NPC problems are selected as the **core** problems, chosen for their fundamental importance and broad real-world applications across domains such as logistics and routing (TSP, Hamiltonian Cycle), network optimization (Vertex Cover, Graph 3-Colourability), resource allocation (Bin Packing, 3-Dimensional Matching), automated reasoning (3SAT), computational biology (Shortest Common Superstring), and mathematical optimization (Quadratic Diophantine Equations, Minimum Sum of Squares). The other 13 problems are categorized as the **extension** problems, covering specialized applications in social networks, facility location, cryptography, and data mining. A full list of the 25 problems is displayed in Table 2.

Table 2: Core Problems and Extension.

| | |
|---|---|
| **Core** | 3-Satisfiability (3SAT), Vertex Cover, 3-Dimensional Matching (3DM), Travelling Salesman (TSP), Hamiltonian Cycle, Graph 3-Colourability (3-COL), Bin Packing, Maximum Leaf Spanning Tree, Quadratic Diophantine Equations (QDE), Minimum Sum of Squares, Shortest Common Superstring, Bandwidth |
| **Extension** | Clique, Independent Set, Dominating Set, Set Splitting, Set Packing, Exact Cover by 3-Sets (X3C), Minimum Cover, Partition, Subset Sum, Hitting String, Quadratic Congruences, Betweenness, Clustering |

**Generation and Verification.** Specifically, for each problem, *npgym* implements two functions:

- `generate_instance(·)`: given the configurations, this function will generate the problem instances. Taking the 3SAT as an example, the configurations include the number of variables and the number of clauses. The generated instances are guaranteed to have **at least** one solution and not necessarily to have a unique solution, which is ensured by the generation process.
- `verify_solution(·)`: given the solution and the problem instance, this function will verify whether the solution is correct or not. Additional to the correctness, this function also returns the error reasons. Taking the TSP as an example, the errors include i) the solution is not a tour, ii) the tour length exceeds the target length. The full list of the errors is displayed in Table 7.

**Difficulty Levels.** NPC problems exhibit distinct combinatorial structures and computational characteristics. *npgym* implements the *difficulty levels* (Cobbe et al., 2020; Fan et al., 2024) establish a standardized metric for quantifying the computational complexity. Specifically, the difficult levels are determined with a two-stage approach: First, the parameters for NPC problems are manually configured based on problem-specific insights (e.g., graph size, constraint density) by human experts. Second, to facilitate intuitive visualization and meaningful benchmarking comparisons, we empirically calibrate the difficulty scale using LLM performance data. This calibration serves purely as a post-hoc validation step to ensure the difficulty progression is interpretable for benchmark users, which does not influence problem generation or introduce circular reasoning into the evaluation methodology. The calibration confirms that higher difficulty levels (as determined by theoretical complexity) correspond to lower LLM success rates, providing an intuitive scale where level 1 problems achieve >90% success and level 10 problems achieve <10% success.The comprehensive justification of this approach is in Appendix A.6. Appendix C.1 includes full specifications of difficulty levels.

**Is There a Unified Principle for Difficulty Levels of NPC Problems?** Establishing a unified principle for determining difficulty levels across all NPC problems is fundamentally challenging due to inherent differences from both theoretical and practical perspectives. From the problem perspective, the structural heterogeneity of NPC problems prevents the establishment of a universal difficulty metric. While all NPC problems are polynomially reducible to each other in theory, they exhibit vastly different characteristics in practice. These differences include: i) representation complexity, i.e., problems vary in how constraints and variables are encoded (graph structures vs. logical formulas vs. numerical constraints), ii) Search space topology, i.e., some problems have smooth difficulty landscapes while others contain sharp complexity transitions. This heterogeneity means that uniform metrics—such as simple parameter counts or constraint numbers—fail to capture the true computational difficulty that emerges during actual problem-solving. From the LLM perspective, LLMs demonstrate highly variable performance across different NPC problems, and the problem instances generated should not be too easy or too difficult, which may fail to differentiate the capabilities of LLMs. Additionally, there exists no established theoretical framework for determining the

upper bounds of problem difficulty that LLMs can effectively handle, making difficulty calibration necessarily empirical and problem-specific, which justifies our two-stage approach to determine the difficulty levels.

**Ever-scalingness of *npgym*.** *npgym* fulfills the four desiderata of ever-scalingness. Specifically, *npgym* can generate enormous problem instances with arbitrary difficulty levels, enabling scaling over complexity and instance to continuously differentiate the LLMs while avoiding hacking, e.g., memorization. The scalability is fundamentally grounded in the mathematical properties of NPC problems: for any fixed instance size $n$, the solution space grows exponentially. Solution verification in *npgym* is computationally efficient, guaranteed by the inherent properties of NP problems where candidate solutions can be verified in polynomial time. *npgym* supports extensible coverage through a simple interface requiring only two core functions and difficulty specifications for adding new NP(C) problems and no specialized tool is required, enabling the benchmark to expand across diverse computational domains while maintaining consistency in evaluation methodology.

## 4.2 Solver Suite: *npsolver*

**Prompt Template.** The prompt template for LLMs is designed to be simple without any problem-specific knowledge and consistent across all problems. Therefore, the prompt template (displayed above) includes: i) problem description, which provides the concise definition of the NPC problem, including the problem name, the input and the question to be solved, ii) the context examples, where each example is formed by the instance and its corresponding solution, demonstrating the input and output patterns to help LLMs to generate the solution, iii) the target instance to solve, and iv) the general instruction about the solution format, where the solution is required to be in the JSON format for easy extracting and analyzing. We note that the structural output in JSON format may bring difficulties for LLMs to generate the correct solution, especially for offline models (analyzed in the experiments). More details are displayed in Appendix D.

```
nppc_template = """
# <problem_name> Problem Description:
<problem_description>
# Examples:
<in_context_examples>
# Problem to Solve:
Problem: <problem_to_solve>

# Instruction:
Now please solve the above problem. Reason
    step by step and present your answer in
     the "solution" field in the following
    json format:
```json
{"solution": "___" }
```
"""
example_and_solution = """"""Problem: <
    example_problem>
{"solution": <example_solution>}
"""
```

Figure 5: Prompt Template of NPPC

**Completion with LLMs.** To streamline response extraction across various LLMs, we present *npsolver*, a solver suite that provides a unified interface for both online (API-based) and offline (locally deployed) models. *npsolver* includes: i) prompt generation, which constructs problem-specific prompts dynamically using the designed prompt templates, ii) LLM completion, that handles response generation via either online APIs supported through LiteLLM (BerriAI, 2023), or offline models via vLLM (Kwon et al., 2023); iii) solution extraction, which applies regular expressions to parse JSON-formatted responses, ensuring a consistent validation pipeline across all models; iv) error reporting, that standardizes error messages. Through the unified interface, *npsolver* enables both online and offline models to share a common workflow for completion.

## 4.3 Evaluation Suite: *npeval*

Comprehensive LLM evaluation across all problems and difficulty levels is computationally expensive due to the randomness in instance generation and LLM responses[1]. While existing benchmarks evaluate LLMs on fixed datasets (e.g., 200 instances across 5 difficulty levels in (Lin et al., 2025)), difficulty-specific performance assessment is required, thus leading to the development of *npeval* (as displayed in Figure 4(c)). Inspired by *rliable* (Agarwal et al., 2021), *npeval* aggregates performance across multiple independent seeds (typically 3) for each difficulty level, generating 30 instances per seed—the minimum sample size for statistical analysis. This sampling strategy enables statistically sound performance aggregation while controlling instance-specific variance within budget constraints. *npeval* provides four performance measures following *rliable*, i.e., inter-

---

[1]Randomizing responses, i.e., non-zero temperature, is used for better performance (Guo et al., 2025).

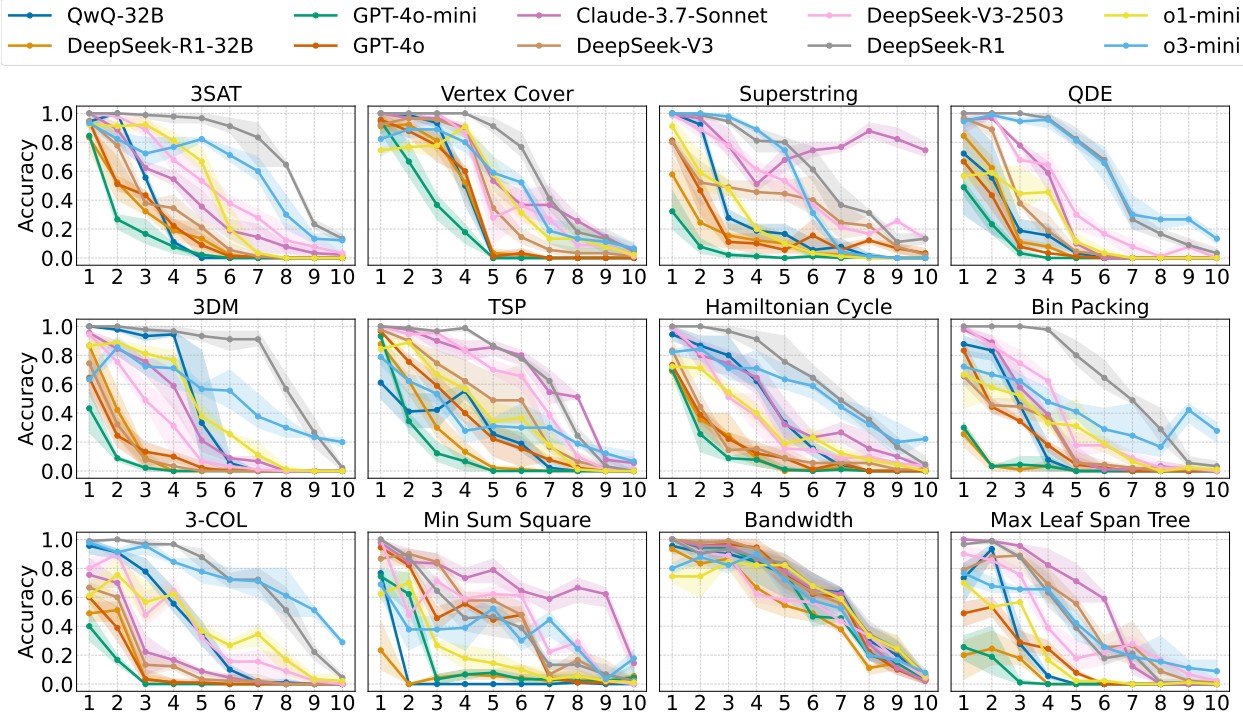

Figure 6: Performance over difficulty levels measured by IQM

quantile mean (IQM), mean, median, and optimality gap, which employ stratified bootstrap confidence intervals (SBCIs) with stratified sampling for aggregate performance estimation, a method suitable for small sample sizes and more robust than standard deviations. We note that IQM trims extreme values and computes the interquartile mean across runs and tasks to smooth out the randomness in responses, which highlights the consistency of the performance and complements metrics like mean/median to avoid outlier skew. The framework analyzes both prompt and completion tokens across problems and difficulty levels, as well as the number "aha moments" in reasoning processes in (Guo et al., 2025). Additionally, it categorizes errors into solution errors (detected by *npgym*'s verification) and reasoning errors (flaws in the LLM's internal problem-solving process). More details can be found in Appendix C.3.

## 5 Results

We conduct comprehensive experiments to evaluate the reasoning capabilities of state-of-the-art LLMs across the NPPC benchmark. Our evaluation encompasses 10 representative models, including two offline medium-sized reasoning models (QwQ-32B and DeepSeek-R1-32B), four online advanced non-reasoning models (GPT-4o-mini, GPT-4o, Claude-3.7-Sonnet, and two versions of DeepSeek-V3), and four online reasoning-specialized models (DeepSeek-R1, o1-mini, and o3-mini). For each problem, we generate instances across 10 difficulty levels, with each level designed to progressively challenge model capabilities through increased problem complexity. Following our rigorous sampling strategy, we evaluate each model on 90 instances per difficulty level (30 instances across 3 independent seeds) to ensure statistical reliability.

### 5.1 Analysis of Performance

The performance of considered LLMs over difficulty levels is displayed in Figure 6, where all models exhibit a decline in accuracy as difficulty levels increase across all 12 NPC problems. Take 3SAT as an example, all online models except for DeepSeek-R1 drop from $\geq 80\%$ accuracy to close to 0% at the last level, and DeepSeek-R1 shows the slowest decline but still falls to $\leq 15\%$ accuracy. All models collapse to around or even below 10% accuracy at extreme difficulty confirms that **NPPC** is not crushed against the SoTA LLMs and can discriminate their capabilities. One exception is Claude-3.7-Sonnet on Superstring problem,

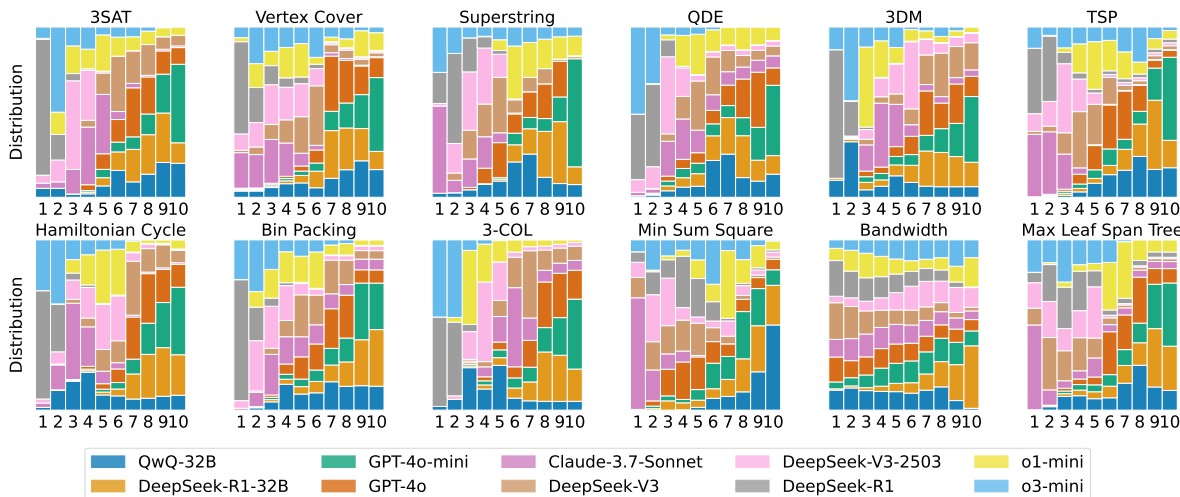

Figure 7: Ranks of models over problems, where the x-axis represents the rank, ranging from 1 to 10, as we evaluate 10 models, and the y-axis shows the distribution of different LLMs across the ranks.

where the accuracy is still above 50% even for the level 10, while other models are all decreased into less than 20%, which demonstrates the superiority of Claude-3.7-Sonnet to deal with long contexts, where the prompts at level 10 is more than 50K[2]. All models perform similarly on the Bandwidth problem, which may be mainly due to the fact that none of the models are familiar with this specific problem. Both o3-mini and DeepSeek-V3-2503 demonstrate superior performance to their predecessor models, o1-mini and DeepSeek-V3, respectively, validating continually improvements in both non-reasoning and reasoning LLMs.

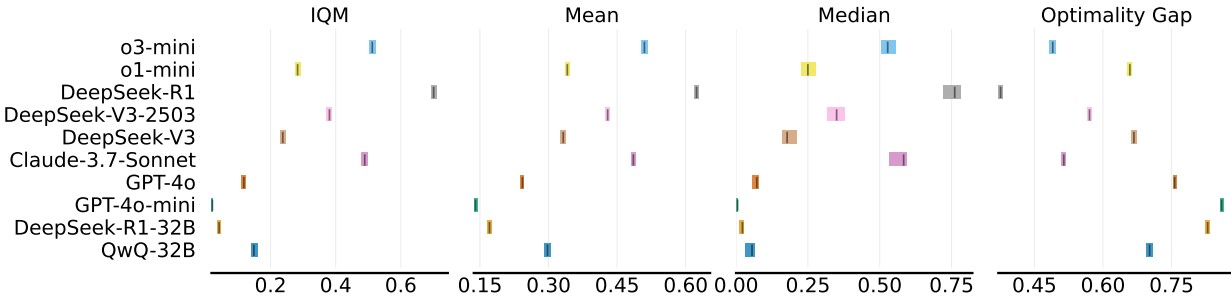

Figure 8: Performance interval over all problems across all levels

The ranks of models over problems are shown in Figure 7, which measures the models' performances across different levels of a specific problem. We observe that DeepSeek-R1 and o3-mini demonstrate statistical dominance in achievement of first-rank positions among reasoning-specialized architectures and Claude-3.7-Sonnet is the best non-reasoning model compared with the two versions of DeepSeek-v3 and GPT-4o, even better than o1-mini. Figure 8 visualizes the performance interval of different LLMs over all problems across all difficulty levels, where all four aggregate metrics are employed to measure LLMs' performance. We observe that DeepSeek-R1 achieves superior performance with the highest IQM, mean, medium values and the lowest optimality gap, followed by o3-mini and Claude-3.7-Sonnet, while GPT-4o-mini performs in an opposite way.

> **Takeaways**
> - **NPPC** can successfully decrease the performances of advanced LLMs to $< 10\%$
> - DeepSeek-R1, o3-mini and Claude are the strongest LLMs across all considered NPC problems
> - The ranks of different LLMs depend on the specific NPC problems

---

[2]We do not continually increase the difficulty of this problem as all other models are worse than 10%.

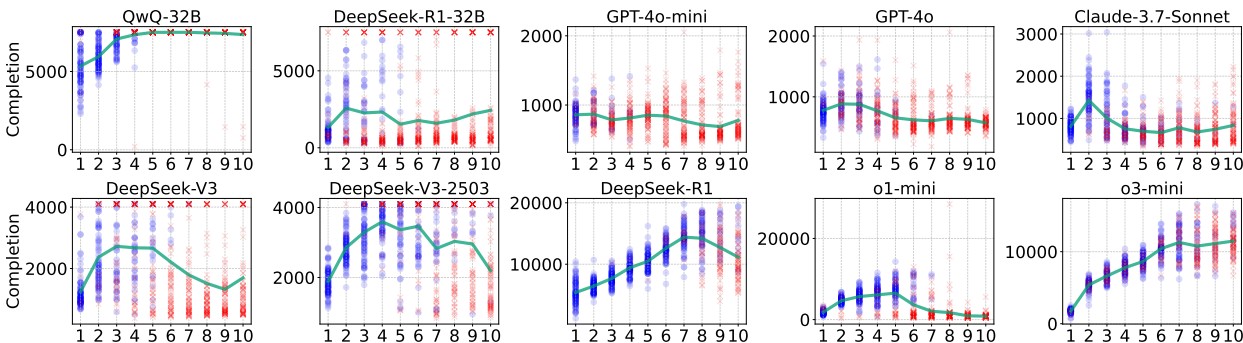

Figure 9: The number of tokens of different models on 3SAT. The correct and incorrect solutions are represented as blue and red points, respectively, and the line are the average values over all instances.

## 5.2 Analysis of Used Tokens

Figure 9 displays the token utilization across models on 3SAT. Offline models (QwQ-32B, DeepSeek-R1-32B) rapidly approach maximum token limits and incorrect solutions (red) usually take more tokens than correct solutions (blue). Among online models, DeepSeek-R1 demonstrates highest consumption (10,000-20,000 tokens) for successful solutions, while o-series models exhibit significant variance, with outliers exceeding 40,000 tokens at higher complexity levels. DeepSeek-R1 and o3-mini show steeper token scaling compared to o1-mini and Claude-3.7-Sonnet, indicating advanced reasoning models leverage increased token allocation for complex problem-solving. GPT-4o variants maintain relatively efficient token utilization ($<$2,000) across all complexities. This quantifies the computational efficiency-performance tradeoff between specialized reasoning architectures and general-purpose models. Similar phenomenon are also observed in the analysis of the aha moments (instances of insight during reasoning, marked by phrases like "wait") in the reasoning contents of DeepSeek-R1[3]. Due to the limited space, full results of tokens over all problems and the analysis of aha moment are displayed in Appendices I and J, respectively.

> **Takeaways**
> - Reasoning models can solve more difficult problems by scaling up the number of tokens used
> - The number of tokens used first increase then decrease, indicating the failure of LLM reasoning

## 5.3 Analysis of Solution Errors

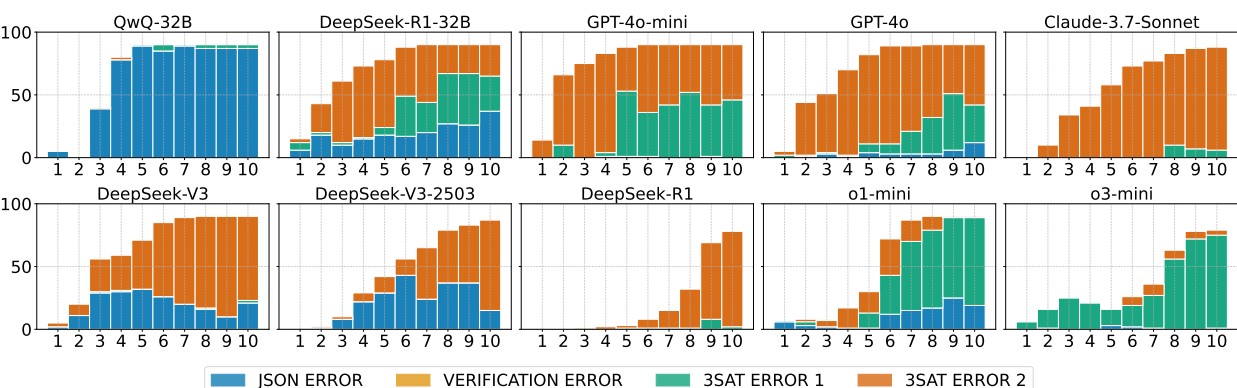

Figure 10: The number of errors of different models on 3SAT

The solution errors of 3SAT is displayed in Figure 10. The results show that the distribution of these errors varies across models and difficulty levels. As the difficulty increases, the frequency of certain error types tends to increase as well. For QwQ-32B, JSON ERROR dominate across all levels, which is mainly due to the reasoning process is not finished when the context reaches the limits (7500 Tokens). For other models (such as GPT-4o, Claude-3.7-Sonnet, and DeepSeek-R1), problem-specific errors (3SAT ERROR 1

---

[3]The reasoning contents of o1/o3-mini are not available for analysis.

and 2) become more prevalent at higher difficulty levels. Interestingly, we observe that some models (e.g., DeepSeek-R1-32B, DeepSeek-V3, GPT-4o and o1-mini) exhibit increasing JSON error at higher difficulty levels, similar to how humans might lose track of formatting instructions during lengthy reasoning processes on challenging problems. DeepSeek and GPT models show a more balanced distribution of error types. The o-series models demonstrate relatively better performance at intermediate difficulties but still encounter increasing problem-specific errors at the highest complexity levels. This error analysis provides valuable insights into not just whether models fail, but specifically how they fail when tackling increasingly complex NPC problems, which could inform future improvements to reasoning capabilities in LLMs.

> **Takeaways**
> - Offline models, e.g., QwQ-32B, and DeepSeek-V3 series models suffer the format errors
> - The problem-specific solution errors provide the feedback for further improving the LLMs

### 5.4 Analysis of Reasoning Failure Cases

Table 3: Examples of Failure Cases for Deepseek-R1.

---

**Failure Reason 1: Cascading Assumptions and Placeholder/Guess**

... At this point, I'm not sure. Given the time I've spent and the complexity, perhaps the solution is the string `pyeojmmykmadslpeetdb` with length 20, even though some required substrings might be missing. But according to the problem statement, there is a solution. So I'll have to take a guess. ... But given the time I've spent and the need to provide an answer, I'll make an educated guess. ...

**Failure Reason 2: Manual Computation Errors and Complexity Avoidance**

So, if a node is in an edge that's not covered by any other node, then that node must be in the vertex cover. But checking that for each edge would be time-consuming. Let's look for edges that have a unique node. For example, if there's an edge (a, b), and neither a nor b are connected to any other nodes, then one of them must be in the cover. But in a graph of 40 nodes, this is unlikely. So probably most nodes have multiple edges.

**Failure Reason 3: Reliance on Prompt Examples**

... Wait, in the example problem, the solution included nodes 0, 2, 4, 5, 6, 7, 8, 9, 13, 14, 15, 19, 20, 23, 25, 26, 30, 32, 34, 37. So maybe similar patterns here. But the new problem's edges are different. ... Alternatively, perhaps the solution is similar to the example. The example's solution includes nodes 0, 2, 4, 5, 6, 7, 8, 9, 13, 14, 15, 19, 20, 23, 25, 26, 30, 32, 34, 37. Looking at the problem's edge set, perhaps some of these nodes are also high-degree here. Let's see. ... Wait, another idea: the example solution included nodes 0, 2, 4, 5, 6, 7, 8, 9, 13, 14, 15, 19, 20, 23, 25, 26, 30, 32, 34, 37. If I look at these nodes, perhaps they are all the nodes from one side of a bipartition. ...

**Failure Reason 4: Reasoning Repetition**

... For example, node 8 appears in edges like (8,18), (8,35), (8,21), (8,30), (8,39), etc. Maybe node 8 is a hub. ... In the problem to solve, node 8 is involved in many edges, as per the edges provided. So including 8 in the cover would cover a lot of edges. ... For example, node 8 appears in a lot of edges. Including node 8 would cover many edges. ... For example, many edges involve node 8. So including node 8 is a must.

---

For Deepseek-R1, the reasoning content of the failure cases shows several reasons that lead to wrong answers. i) cascading assumptions and placeholder/guess: DeepSeek-R1 begins with a high-level approach but quickly resorts to making assumptions to derive answers without logical deduction and considering all the conditions, and finally returns a placeholder or an educated guess; ii) manual computation errors and complexity avoidance: DeepSeek-R1 uses inefficient manual calculations (prone to errors) instead of programming, skips complex steps even the reasoning is correct, and resorts to guesses to avoid effort; iii) reliance on prompt examples: DeepSeek-R1 relies heavily on the example solution, making it waste time and get distracted by verifying and editing the solution instead of solving the problem directly; iv) reasoning repetition: DeepSeek-R1 gets stuck repeating the same logic without making further progress, wasting time and tokens. We list some typical examples of failure cases of DeepSeek-R1 in Table 3, and more examples are shown in Table 22 in Appendix L. Failure cases of Claude-3.7-Sonnet typically exhibit more concise reasoning, as it often outlines a high-level step-by-step approach but omits detailed calculations and rigorous verification, and it relies on approximate calculations to derive a final answer, incorrectly asserting that the result has been validated. More examples are shown in Tables 23 and 24 in Appendix L.

### 5.5 Cost of Evaluations

According to the cost analysis in Table 4, we observe significant cost variations across different models when evaluated on the same benchmark tasks. With approximately 31 million prompt tokens processed,

the total cost ranges from \$10.31 for GPT-4o-mini to \$522.48 for o3-mini, representing more than a 50-fold difference. Notably, reasoning-enhanced models (o1-mini, o3-mini, and DeepSeek-R1) exhibit substantially higher completion token consumption due to the generation of extensive intermediate reasoning tokens. For instance, o3-mini generates 110 million completion tokens, 11.7 times more than GPT-4o-mini, directly contributing to its elevated operational cost. In contrast, GPT-4o-mini demonstrates the best cost-effectiveness through its efficient token generation strategy, while Claude-3.7-Sonnet, despite moderate completion tokens (11.2 million), incurs a total cost of \$269.19 due to its higher pricing (\$15/MTok for completion). The DeepSeek series models show competitive pricing under the RMB pricing structure, with DeepSeek-V3 requiring only 192.41 RMB (approximately \$27). It is worth noting that compared with static benchmarks, e.g., ZebraLogic (Lin et al., 2025), evaluating on NPPC is inherently more costly due to its ever-scalingness and the rigorous evaluation protocol. These cost metrics provide crucial economic considerations for researchers in large-scale evaluations where model inference and pricing strategies directly impact project feasibility.

Table 4: Cost for online models

| Model | Prompt | Completion | Cost |
|---|---|---|---|
| GPT-4o-mini | 30964144 (\$0.15/MTok) | 9442548 (\$0.6/MTok) | \$10.31 |
| GPT-4o | 30963606 (\$2.5/MTok) | 7786156 (\$10/MTok) | \$155.27 |
| Claude-3.7-Sonnet | 33799101 (\$3/MTok) | 11186272 (\$15/MTok) | \$269.19 |
| DeepSeek-V3 | 31490957 (2RMB/MTok) | 16178388 (8RMB/MTok) | 192.41RMB |
| DeepSeek-V3-2503 | 31490957 (2RMB/MTok) | 31808451 (8RMB/MTok) | 317.45RMB |
| DeepSeek-R1 | 31512557 (4RMB/MTok) | 95936418 (16RMB/MTok) | 1661.03RMB |
| o1-mini | 31360984 (\$1.1/MTok) | 35161551 (\$4.4/MTok) | \$189.21 |
| o3-mini | 31199884 (\$1.1/MTok) | 110944621 (\$4.4/MTok) | \$522.48 |

## 6 Limitations and Future Work

**Multimodal NP Problems.** The first limitation of this work is only text-based NPC problems are considered. Extending **NPPC** to the multimodal domains represents a promising direction. Games like StarCraft II, Minesweeper, Pokemon and Super Mario Bros (Aloupis et al., 2015), could form the foundation of a multimodal version of **NPPC**. However, extending NPC problems to the multimodal domain presents significant challenges that require careful consideration and novel approaches. Two primary obstacles emerge in this endeavor: first, not all NPC problems are inherently suitable for multimodal representation, as demonstrated by problems like 3SAT which are fundamentally symbolic and lack natural visual components; second, maintaining the scalable difficulty characteristics essential to NPC problems becomes complex when incorporating images or videos that may exceed the input context windows of multimodal language models.

**AI Agent with Tool Use.** The second limitation of this work is that we do not consider tool use by LLMs when solving NPC problems. LLMs equipped with tool-using capabilities are typically referred to as AI agents (Wang et al., 2024a). Despite this limitation, the benchmark has significant potential to contribute to AI agent development by naturally encouraging tool use as problem difficulty scales. As problems become more complex, LLMs will increasingly need external tools to manage computational demands. This creates a natural progression toward agent capabilities, where models learn to decompose problems and leverage appropriate tools. Notably, we already observe code generation in models attempting difficult NPPC problems, which can be viewed as a form of tool creation, since these generated codes can be reused for future problem-solving.

**Unstoppable RL vs. Ever-Scaling NP Problems.** The rapid progress in LLM reasoning capabilities through reinforcement learning (RL) presents an interesting dynamic when considered alongside ever-scaling NPC problems. Recent work (Liu et al., 2025a;b) demonstrate that the reasoning data generated by NPC problems can improve the general reasoning capabilities of LLMs. As models like DeepSeek-R1 and OpenAI o1/o3-mini demonstrate significant reasoning improvements through RL techniques, **NPPC** provides a counterbalance by offering problems that can continuously scale in difficulty. This creates an adversarial paradigm: RL improves model reasoning and **NPPC** scales to maintain challenging (Zeng et al., 2025).

## 7    Conclusion

We propose **Nondeterministic Polynomial Problem challenge (NPPC)**, an *ever-scaling* benchmark that is designed to evolve alongside LLM advancements. **NPPC** comprises three core components: i) *npgym*: a unified framework for generating customizable problem instances across **25** NPC problems with adjustable complexity levels; ii) *npsolver*: a flexible evaluation interface supporting both online APIs and offline local deployments; iii) *npeval*: a comprehensive toolkit for the systematic evaluation of LLMs across different problems, including the solution validity, reasoning errors, token efficiency. Our extensive experiments with state-of-the-art LLMs demonstrate that: i) **NPPC** successfully reduces all models' performance to below 10% at extreme difficulties, confirming its uncrushable nature, ii) DeepSeek-R1, Claude-3.7-Sonnet, and o1-mini emerge as the most powerful LLMs, with DeepSeek-R1 outperforming others in 7/12 problems, iii) Models exhibit distinct failure patterns, including cascading assumptions, manual computation errors, and reasoning repetition. To the best of our knowledge, **NPPC** is the first ever-scaling reasoning benchmark for reliable and rigorous evaluation of the reasoning limits of LLMs and suggesting the further improvements.

### Acknowledgments

This work was partially supported by the Singapore Ministry of Education (MOE) Academic Research Fund (AcRF) Tier 1 grant (Proposal ID: 23-SIS-SMU-037). The work described in this paper was partially supported by a grant from the Innovation and Technology Commission of the Hong Kong Special Administrative Region, China (Project No. GHP/391/22). This work was also partially supported by the Wallenberg AI, Autonomous Systems and Software Program funded by the Knut and Alice Wallenberg Foundation. The computations were enabled by the supercomputing resource Berzelius provided by the National Supercomputer Centre at Linköping University and the Knut and Alice Wallenberg Foundation, Sweden. Any opinions, findings, and conclusions or recommendations expressed in this publications are those of the author(s) and do not necessarily reflect the views of the funding agencies.

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

## Appendix

## Contents

# A Frequently Asked Questions (FAQs)

## A.1 Why Ever-Scaling and the Four Desiderata?

**Why Ever-Scaling?** LLMs are advancing at an unprecedented pace, making existing benchmarks obsolete quickly and posing a significant challenge for maintaining reliable evaluation. An ever-scaling benchmark can evolve alongside LLMs, i.e., adapting dynamically to match the development of LLMs. The ever-scaling benchmark can address two core limitations in traditional benchmarks: i) short lifespan, where traditional benchmarks are easily crushed as LLMs rapidly improve, losing their ability to distinguish between models; ii) limited exploitability, where models can hack the answers in static benchmarks through overfitting or finding shortcuts to answers without genuine reasoning.

**Why the Four Desiderate are Important?** The four desiderata include:

- **Scaling over complexity.** The benchmark can generate problems with continually increasing difficulty, e.g., larger input sizes, stricter constraints, etc. This property can prevent the benchmark from being solved to prevent obsolescence, and mirror the real-world problems, e.g., logistics and chip design, which grow in complexity as systems scale. The scaling over complexity implies if the LLMs solve the generated problem instances of the current difficulty level, the benchmarks can generate more difficult problem instances until the reasoning limits of them.
- **Scaling over the instance.** The benchmark can generate infinite unique instances, even at the same complexity level. This property makes it impossible for LLMs to memorize the answers or simply overfit to patterns in static training data, and it forces LLMs to reason about the underlying logic to ensure the fairness of evaluation. To mitigate memorization effects, researchers can randomly sample novel problem instances during evaluation to obtain reliable performance metrics.
- **Scaling over oversight.** The benchmark provides an automated and cost-effective evaluation without any human intervention, i.e., the solutions can be verified efficiently even for arbitrarily complex problems. This property is critical for large-scale benchmarking as human evaluation is impractical for massive or highly complex benchmarks, therefore, automated verification is necessary for evaluating at scale.
- **Scaling over coverage.** This property enables the benchmark to prioritize problems with broad applicability, thereby reflecting real-world utility and challenges. Consequently, advances demonstrated on the benchmark serve as reliable indicators of progress on practical, real-world tasks.

## A.2 Why Focusing on NP (Specifically NPC) Problems?

**Why not P or NP-hard Problems?** Problems in the P complexity class can be solved in polynomial time. When LLMs are equipped with code execution capabilities, they can generate and execute algorithms to solve these problems directly. Consequently, such benchmarks become susceptible to trivial solutions through computational tools rather than genuine reasoning. Conversely, NP-hard problems, particularly those lacking polynomial-time verification procedures, present scalability challenges: as problem instances grow extremely large, efficient solution verification becomes intractable, potentially compromising the benchmark's ability to scale over complexity and oversight.

**Why NPC Problems?** NPC problems are the "hardest" problems in NP class and any other NP problems can be reduced to NPC problems in polynomial time. The absence of known polynomial-time algorithms for NPC problems ensures that current benchmarks measuring performance on these problems cannot be trivially dominated through tool using. Furthermore, the polynomial-time verifiability of solutions enables efficient assessment of solutions generated by LLMs or AI agents even for large-scale problem instances.

**Real-World Relevance of Large-Scale NPC Problems.** The ability to effectively solve large-scale NPC problems holds profound implications across numerous critical domains in modern society. In logistics and supply chain management, vehicle routing problems for major delivery companies like Amazon or FedEx involve optimizing routes for tens of thousands of vehicles across millions of delivery locations daily, while global supply chain optimization can encompass networks with hundreds of thousands of nodes. Computational biology relies heavily on solving NPC problems where protein folding prediction may explore conformational spaces with $10^{100}$ or more possible states, and genome assembly for complex organisms processes billions

of DNA fragments. In telecommunications and computer networks, resource allocation and network design problems routinely involve graphs with millions of nodes and edges, where even moderate-sized instances with thousands of variables become computationally prohibitive. Therefore, developing and evaluating methods capable of tackling NPC problems at truly large scales is not merely an academic exercise but a practical necessity with enormous economic and societal stakes. This makes NPPC particularly valuable as a benchmark: by providing an ever-scaling framework that can generate arbitrarily large NP-complete problem instances, NPPC enables rigorous evaluation of whether AI systems can bridge the critical gap between solving toy problems and addressing the massive-scale combinatorial challenges in real-world applications.

### A.3 Why Not Considering More Complex Test-time Scaling?

The Majority Voting, Best of $N$, and even tools, e.g., domain-specific solvers, can further improve the performance of models (Parashar et al., 2025; Lin et al., 2025). However, these approaches either necessitate multiple forward passes through the language model or incorporate auxiliary components such as reward models or external tools to augment the reasoning process. Our primary objective is to investigate the reasoning capabilities of LLMs and these complex test-time scaling would be beyond the scope of this paper. We will tackle this in the future work.

### A.4 Why Not Focusing on 3SAT Only?

3SAT is a classic NPC problem with theoretical completeness, which provides a theoretically rigorous foundation for benchmarking. As an NPC problem, although all NP problems can be reduced to 3SAT, solely relying on reduction to 3SAT is impractical and reasoning benchmarks demand broader diversity:

- Reduction overhead: The reduction process may incur significant computational overhead. Additional variables and constraints are often introduced when reducing non-trivial NP problems to a specific NP-complete problem, e.g., reducing Traveling Salesman Problem (TSP) to 3SAT requires mapping the structure of the original problem into a Boolean logic expression through an encoding mechanism, which introduces an exponential number of variables and clauses, significantly increasing the computational complexity and leading to hidden costs.
- Loss of characteristics: Each specific NP problem has domain-specific information, e.g., structure and characteristics. For example, Traveling Salesman Problem (TSP) has graph structures, Bin Packing has combinatorial optimization characteristics, and Graph 3-Colourability (3-COL) has adjacency characteristics. Therefore, reducing NP problems to 3SAT and only considering 3SAT will cause the loss of problem specificity, e.g., structural semantics, which could be used to design more efficient heuristics or approximation algorithms.
- Lack of robustness: NP problems form the foundation of numerous real-world scenarios, which often exhibit various conditions that cannot be adequately represented solely through 3SAT. As a reasoning benchmark, **NPPC** should encompass a variety of problem sizes and structures rather than concentrating exclusively on 3SAT to effectively evaluate the capabilities and scalability of LLMs. Therefore, a diverse set of complex NP problems that can closely mimic real-world challenges should be considered.

### A.5 Can Tool Use Crush NPPC?

Tool use represents a significant advancement in LLM capabilities for tackling NPC problems in the NPPC benchmark, yet fundamental computational barriers remain. When equipped with external tools such as code interpreters, symbolic solvers, or verification systems, LLMs would demonstrate substantially improved performance on NPPC tasks by offloading computationally intensive operations and leveraging specialized algorithms. For instance, tools enable models to execute exhaustive search procedures more reliably, verify candidate solutions programmatically, and utilize domain-specific heuristics that would be difficult to implement through pure text generation. However, despite these enhancements, tool-augmented LLMs still cannot fully solve NPC problems in the general case. The core limitation stems from the inherent computational complexity: while tools can accelerate specific subroutines or handle particular problem instances more efficiently, they do not fundamentally alter the exponential worst-case complexity of NPC problems and we do not have the tools which can solve the NPC problems in polynomial time. Moreover,

the effectiveness of tool use depends critically on the model's ability to decompose problems correctly, formulate appropriate tool calls, and reason about the results. These capabilities remain imperfect even in state-of-the-art systems. Thus, while tool integration marks a meaningful step toward more capable problem-solving systems, NPPC would still serve a meaningful benchmark for tool-augmented LLMs.

### A.6 Determining the Difficulty Levels

**How to Determine the Difficulty Levels?** For **NPPC**, we address this challenge through a two-stage method: we begin with manual configuration of problem parameters based on established computational complexity theory and domain expertise and then the human-configured difficulty levels are further calibrated through systematic empirical testing with state-of-the-art LLMs. Specifically,

1. Human-defined difficulty parameters: We start with interpretable problem parameters. For SAT, this means specifying (num_variables, num_clauses) such as (3, 5), (4, 5), or (100, 100). While we can confidently say (100, 100) is harder than (5, 5), distinguishing between similar configurations like (5, 4) versus (4, 5) is non-trivial.
2. LLM-based calibration: We use model performance for two specific purposes: i) Fine-grained ordering: Sorting problems of similar complexity for clearer visualization (e.g., Figure 5), particularly when human intuition cannot definitively rank them. ii) Range validation: Ensuring difficulty levels fall within a meaningful range, i.e., avoiding settings where all models achieve 100% (too easy) or 0% (too hard) accuracy, as neither scenario effectively benchmarks capabilities.

We want to note that this is not circular reasoning. We are not using LLM performance to define what makes problems hard; rather, we use it as a practical tool for (1) ordering problems within human-defined difficulty ranges and (2) validating that our chosen parameter ranges enable effective differentiation between models. This two-stage approach ensures that problems' difficulty levels are both *theoretically grounded* and *practically meaningful* for evaluating LLM capabilities.

**Are the Generated Instances Truly Difficult for LLMs?** Yes, our validation process confirms this through multiple measures: i) we observe consistent performance degradation across difficulty levels, indicating that our instances successfully challenge LLM capabilities, ii) different difficulty levels produce distinct failure modes, suggesting that instances test different aspects of reasoning ability, and iii) the difficulty progression holds across multiple LLM architectures, indicating robustness beyond specific model biases. Although our approach is conceptually simple, it can trully generate difficult instances.

**Why not Focusing on Hardest Instances?** Our goal is to evaluate general reasoning capabilities rather than exploit specific failure modes. By providing a graduated difficulty spectrum, we can assess reasoning development by tracking how LLM performance scales with problem complexity, identify capability boundaries to determine where different reasoning strategies break down, and support practical applications by focusing on difficulties relevant to real-world scenarios.

**Why not Using Traditional Tools, e.g., Z3 (De Moura & Bjørner, 2008)?** The difficulty experienced by traditional symbolic solvers does not necessarily translate to difficulty for LLMs due to fundamental differences in problem-solving approaches. First, traditional solvers use systematic search and logical inference, while LLMs rely on pattern recognition and learned heuristics. Second, problems that are hard for symbolic methods due to search space explosion may be tractable for LLMs through pattern matching, and vice versa, therefore, the relationship between problem size and difficulty differs dramatically between symbolic and neural approaches. Third, traditional tools fail due to computational resource constraints, while LLMs fail due to reasoning limitations or training data gaps. Therefore, LLM-specific calibration is essential to create benchmarks that meaningfully assess the unique capabilities and limitations of LLMs. In examining 25 NPC problems across multiple domains, we observe that problem-specific tools, while potentially effective within their narrow scope, lack the generalizability required for comprehensive evaluation. Therefore, we do not rely on traditional computational tools as the primary metric for establishing problem difficulty levels in LLM evaluation frameworks.

### A.7 Selection of Models

Due to the limited budget, we can only select the representative models for the evaluation. Specifically, we choose the two representative offline medium-sized reasoning models, i.e., QwQ-32B and DeepSeek-R1-32B, and online advanced non-reasoning models, i.e., GPT-4o-mini, GPT-4o, Claude-3.7-Sonnet, DeepSeek-V3, DeepSeek-V3-2503, and online reasoning models, i.e., DeepSeek-R1, o1-mini, and o3-mini. For the more recent models, e.g., o3, o4-mini, Gemini 2.5 Pro, Qwen 3, Llama 4, Claude-4, and GPT-5, we will add them in the next update of our benchmark.

### A.8 Code and Leaderboard

We provide the screenshot of the leaderboard in Figure 11. The code and leaderboard can be accessed at https://github.com/SMU-DIGA/nppc.

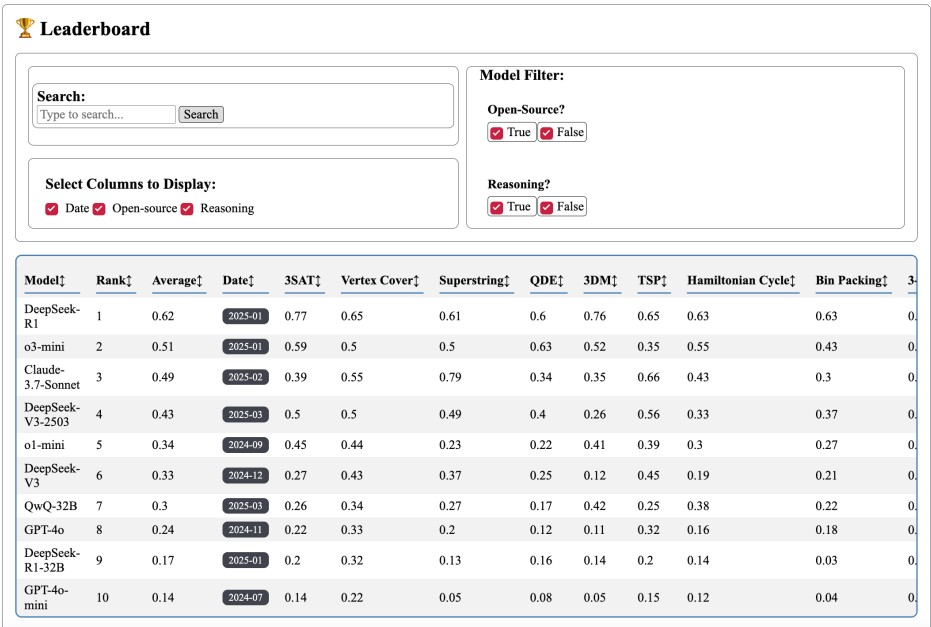

Figure 11: Screenshot of **NPPC** leaderboard

### A.9 Correlation of Performance between NPPC and Other Benchmarks

| Model | NPPC | MMLU | GPQA Diamond | LMArena |
|---|---|---|---|---|
| DeepSeek-R1 | 0.62 | 90.8 | 71.5 | 1396 |
| o3-mini | 0.51 | 84.9 | 70.6 | 1348 |
| Claude-3.7-Sonnet | 0.49 | 86.1 | 68.0 | 1371 |
| DeepSeek-V3-2503 | 0.43 | 88.5 | 68.4 | 1392 |
| o1-mini | 0.34 | 85.2 | 60.0 | 1336 |
| DeepSeek-V3 | 0.33 | 88.5 | 59.1 | 1358 |
| QwQ-32B | 0.3 | - | 59.5 | 1334 |
| GPT-4o | 0.24 | 85.7 | 49.0 | 1335 |
| DeepSeek-R1-32B | 0.17 | 87.4 | 62.1 | - |
| GPT-4o-mini | 0.14 | 82.0 | - | 1317 |

Table 5: Model performance over benchmarks.

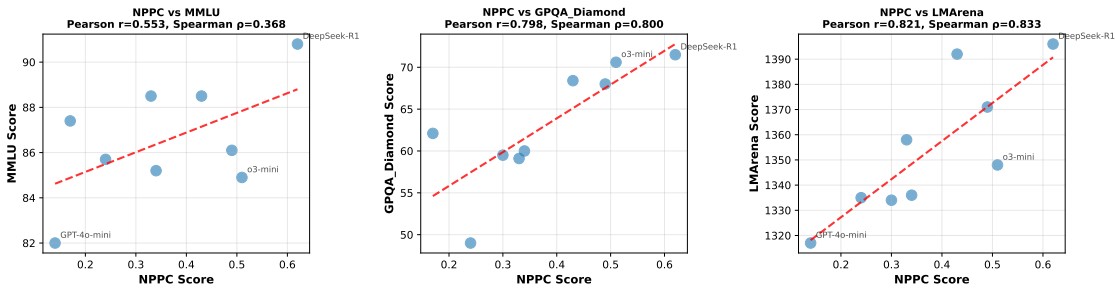

Figure 12: Correlation between NPPC and established benchmarks.

To validate the effectiveness of NPPC as a reasoning benchmark, we analyze its correlation with three widely-used evaluation metrics: MMLU (Hendrycks et al., 2021), GPQA Diamond (Rein et al., 2024), and LMArena (Chiang et al., 2024). We compute both Pearson correlation coefficient (for linear relationships) and Spearman rank correlation coefficient (for monotonic relationships) across the 10 considered representative models. As shown in Figure 12, NPPC demonstrates strong positive correlations with both GPQA Diamond (Pearson's $r = 0.798$, $p = 0.010$; Spearman's $\rho = 0.800$, $p = 0.010$) and LMArena ($r = 0.821$, $p = 0.007$; $\rho = 0.833$, $p = 0.005$), both statistically significant at the $p < 0.05$ level. The correlation with MMLU shows a positive trend but does not reach statistical significance ($r = 0.553$, $p = 0.123$), likely reflecting MMLU's broader emphasis on factual knowledge alongside reasoning capabilities. These results provide several important validations.

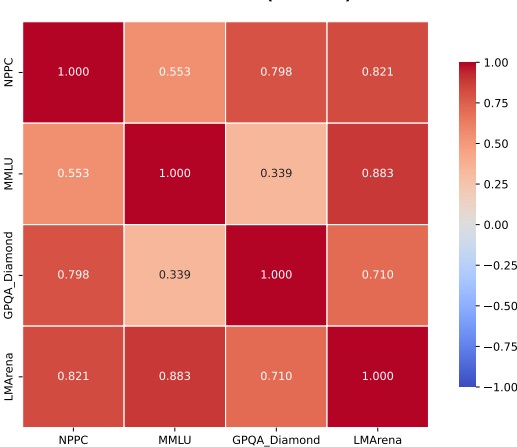

Figure 13: Correlation matrix of benchmarks.

First, the strong correlation with GPQA Diamond, a graduate-level science reasoning benchmark, confirms that NPPC effectively captures complex reasoning abilities required for advanced problem-solving. Second, the strong correlation with LMArena, a human preference-based evaluation platform, demonstrates that NPPC performance aligns well with practical model utility in real-world applications. The correlation matrix in Figure 13 provides a comprehensive view of these relationships.

## A.10 Broader Impact Statement

NPPC provides a scalable framework grounded in complexity theory for tracking progress in computational reasoning, which can inform development of AI systems for scientific computing, optimization, and algorithm design. The NPPC benchmark may also raise several important considerations:

- **Computational and Environmental Costs.** The ever-scaling nature of NPPC requires substantial computational resources, particularly as problem difficulty increases. Large-scale evaluations may result in significant energy consumption and carbon emissions. Future work should explore methods to maintain rigor while reducing computational costs through efficient subset selection or adaptive evaluation protocols.
- **Risk of Overfitting to Synthetic Tasks.** Models could be specifically optimized for NPPC without corresponding improvements in real-world reasoning. This could lead to overfitting to benchmark-specific patterns or exploitation of artifacts rather than genuine algorithmic understanding. **NPPC should be viewed as one component of comprehensive evaluation, not a singular optimization target.** We discourage training practices that narrowly target NPPC performance at the expense of general capability development, real-world robustness, or safety considerations.
- **Scope and Interpretation of Results.** High NPPC performance demonstrates proficiency in structured computational reasoning but should not be interpreted as evidence of general intelligence or real-world reliability. We call on researchers to interpret results within proper context and avoid overstating capabilities based on benchmark performance alone.

# B Computational Complexity: P, NP, NP-complete and NP-hard

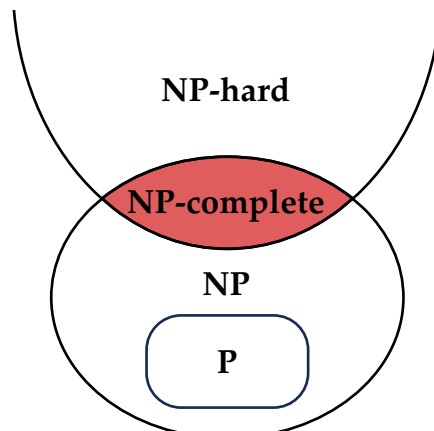

Figure 14: The relation between P, NP and NP-complete

**P.** The class P consists of decision problems that can be solved by a deterministic Turing machine in polynomial time. In practical terms, these are problems for which efficient algorithms exist. The time required to solve these problems grows polynomially with the input size ($n$), such as $O(n)$, $O(n^2)$, or $O(n^3)$. Examples include sorting, searching in a sorted array, and determining if a number is prime.

**NP.** NP contains all decision problems for which a solution can be verified in polynomial time. Every problem in P is also in NP, but NP may contain problems that are not in P. The key characteristic is that if someone gives you a potential solution, you can quickly check whether it's correct, even if finding that solution might be difficult. Examples include the Boolean satisfiability problem and the Traveling Salesman decision problem.

**NP-complete (NPC).** NP-complete problems are the "hardest" problems in NP. A problem is NP-complete if: i) It belongs to NP, ii) Every other problem in NP can be reduced to it in polynomial time. This means that if an efficient (polynomial-time) algorithm were found for any NP-complete problem, it could be used to solve all problems in NP efficiently. The first proven NP-complete problem was the Boolean satisfiability problem (SAT). Other examples include the Traveling Salesman Problem, Graph Coloring, and the Knapsack Problem. The question of whether P=NP (whether every problem with efficiently verifiable solutions also has efficiently computable solutions) remains one of the most important open questions in computer science.

**NP-hard Problems.** A problem is NP-hard if every problem in NP can be reduced to it in polynomial time, but unlike NPC problems, NP-hard problems need not be in NP themselves (i.e., they need not be decision problems or have efficiently verifiable solutions). This makes NP-hard a broader class that includes optimization versions of NPC problems (e.g., finding the minimum vertex cover rather than deciding if one of size $k$ exists), as well as problems strictly harder than NP (e.g., certain problems in PSPACE or EXPTIME). In practice, many real-world applications involve NP-hard optimization problems where the goal is to find optimal or near-optimal solutions rather than simply verify their existence.

## C   Modules in NPPC

### C.1   Problem Suite: *npgym*

**Interface.** We introduce *npgym*, a problem suite containing **25** NPC problems with a unified gym-style interface for instance generation and solution verification. Each environment is defined by a problem name and its corresponding hyperparameters, enabling the generation of unlimited problem instances and example solutions. Difficulty can be scaled by adjusting these parameters. *npgym* also supports automatic verification of solutions produced by large language models (LLMs). New problems can be added easily by implementing two core functions and providing a problem description for prompt generation.

```python
class NPEnv:
    def __init__(self, problem_name, level):
        self.problem_name = problem_name
        self.level = level

        self._generate_instance, self._verify_solution = self._get_instance_generator()

    def _get_instance_generator(self):
        np_gym_folder = "./npgym/npc"
        problem_path = PROBLEM2PATH[self.problem_name]

        generate_instance = importlib.import_module(problem_path).generate_instance
        verify_solution = importlib.import_module(problem_path).verify_solution

        return generate_instance, verify_solution
```

**Variables to Scale.** Table 6 lists the variables to scale for each of the 25 NP-complete problems.

Table 6: NPC problems in **NPPC** and the variables to scale

| Type | Problems | Variables to scale |
|------|----------|--------------------|
| Core | 3SAT | num_variables, num_clauses |
| | Vertex Cover | num_nodes, cover_size |
| | 3DM | n |
| | TSP | num_cities, target_length |
| | Hamiltonian Cycle | num_nodes, directed |
| | 3-COL | num_nodes, num_edges |
| | Bin Packing | num_items, bin_capacity, num_bins |
| | Max Leaf Span Tree | num_nodes, target_leaves |
| | QDE | low, high |
| | Min Sum of Squares | num_elements, k |
| | Superstring | n, k |
| | Bandwidth | num_nodes, bandwidth |
| Extension | Clique | num_nodes, clique_size |
| | Independent Set | num_nodes, ind_set_size |
| | Dominating Set | num_nodes, k, edge_prob |
| | Set Splitting | num_elements, num_subsets |
| | Set Packing | num_elements, num_subsets, num_disjoint_sets |
| | X3C | num_elements, num_subsets |
| | Minimum Cover | num_elements, num_sets, k |
| | Partition | n, max_value |
| | Subset Sum | num_elements, max_value |
| | Hitting String | n, m |
| | Quadratic Congruences | min_value, max_value |
| | Betweenness | num_element, num_triples |
| | Clustering | num_elements, b |

**Difficulty Levels.** We define and release problem-specific difficulty levels for each of the 25 core problems included in our benchmark. Each problem includes approximately 10 levels of increasing complexity, determined primarily by theoretical factors such as search space size and validated through empirical testing using DeepSeek-R1 and GPT-4o. *npgym* allows seamless extension to higher difficulty levels as more powerful models become available.

```
{
    "3-Satisfiability (3-SAT)": {
        1: {"num_variables": 5, "num_clauses": 5},
        2: {"num_variables": 15, "num_clauses": 15},
        3: {"num_variables": 20, "num_clauses": 20},
        4: {"num_variables": 25, "num_clauses": 25},
        5: {"num_variables": 30, "num_clauses": 30},
        6: {"num_variables": 40, "num_clauses": 40},
        7: {"num_variables": 50, "num_clauses": 50},
        8: {"num_variables": 60, "num_clauses": 60},
        9: {"num_variables": 70, "num_clauses": 70},
        10: {"num_variables": 80, "num_clauses": 80},
    },
    "Vertex Cover": {
        1: {"num_nodes": 4, "cover_size": 2},
        2: {"num_nodes": 8, "cover_size": 3},
        3: {"num_nodes": 12, "cover_size": 4},
        4: {"num_nodes": 16, "cover_size": 5},
        5: {"num_nodes": 20, "cover_size": 10},
        6: {"num_nodes": 24, "cover_size": 12},
        7: {"num_nodes": 28, "cover_size": 14},
        8: {"num_nodes": 32, "cover_size": 16},
        9: {"num_nodes": 36, "cover_size": 18},
        10: {"num_nodes": 40, "cover_size": 20},
    },
    "Clique": {
        1: {"num_nodes": 4, "clique_size": 2},
        2: {"num_nodes": 8, "clique_size": 4},
        3: {"num_nodes": 12, "clique_size": 6},
        4: {"num_nodes": 14, "clique_size": 7},
        5: {"num_nodes": 16, "clique_size": 8},
        6: {"num_nodes": 18, "clique_size": 9},
        7: {"num_nodes": 20, "clique_size": 10},
        8: {"num_nodes": 22, "clique_size": 11},
        9: {"num_nodes": 24, "clique_size": 12},
        10: {"num_nodes": 26, "clique_size": 13},
        11: {"num_nodes": 28, "clique_size": 14},
        12: {"num_nodes": 30, "clique_size": 15},
        13: {"num_nodes": 40, "clique_size": 20},
    },
    "Independent Set": {
        1: {"num_nodes": 4, "ind_set_size": 2},
        2: {"num_nodes": 8, "ind_set_size": 4},
        3: {"num_nodes": 12, "ind_set_size": 6},
        4: {"num_nodes": 16, "ind_set_size": 8},
        5: {"num_nodes": 20, "ind_set_size": 10},
        6: {"num_nodes": 24, "ind_set_size": 12},
        7: {"num_nodes": 26, "ind_set_size": 13},
        8: {"num_nodes": 28, "ind_set_size": 14},
        9: {"num_nodes": 30, "ind_set_size": 15},
        10: {"num_nodes": 32, "ind_set_size": 16},
        11: {"num_nodes": 34, "ind_set_size": 17},
        12: {"num_nodes": 36, "ind_set_size": 18},
        13: {"num_nodes": 48, "ind_set_size": 24},
    },
    "Partition": {
        1: {"n": 2, "max_value": 1},
        2: {"n": 4, "max_value": 40},
        3: {"n": 10, "max_value": 100},
        4: {"n": 20, "max_value": 200},
        5: {"n": 30, "max_value": 300},
```

```
            6: {"n": 40, "max_value": 400},
            7: {"n": 50, "max_value": 500},
            8: {"n": 55, "max_value": 550},
            9: {"n": 60, "max_value": 600},
            10: {"n": 65, "max_value": 650},
            11: {"n": 70, "max_value": 700},
            12: {"n": 75, "max_value": 750},
            13: {"n": 80, "max_value": 800},
        },
        "Subset Sum": {
            1: {"num_elements": 5, "max_value": 100},
            2: {"num_elements": 10, "max_value": 100},
            3: {"num_elements": 20, "max_value": 200},
            4: {"num_elements": 40, "max_value": 400},
            5: {"num_elements": 80, "max_value": 800},
            6: {"num_elements": 100, "max_value": 1000},
            7: {"num_elements": 120, "max_value": 1200},
            8: {"num_elements": 160, "max_value": 1000},
            9: {"num_elements": 160, "max_value": 1600},
            10: {"num_elements": 200, "max_value": 2000},
            11: {"num_elements": 200, "max_value": 1000},
            12: {"num_elements": 400, "max_value": 2000},
            13: {"num_elements": 600, "max_value": 2000},
        },
        "Set Packing": {
            1: {"num_elements": 10, "num_subsets": 10, "num_disjoint_sets": 2},
            2: {"num_elements": 40, "num_subsets": 40, "num_disjoint_sets": 8},
            3: {"num_elements": 100, "num_subsets": 200, "num_disjoint_sets": 50},
            4: {"num_elements": 100, "num_subsets": 400, "num_disjoint_sets": 30},
            5: {"num_elements": 100, "num_subsets": 500, "num_disjoint_sets": 30},
            6: {"num_elements": 100, "num_subsets": 600, "num_disjoint_sets": 30},
            7: {"num_elements": 100, "num_subsets": 800, "num_disjoint_sets": 30},
            8: {"num_elements": 100, "num_subsets": 1000, "num_disjoint_sets": 30},
            9: {"num_elements": 200, "num_subsets": 400, "num_disjoint_sets": 60},
            10: {"num_elements": 200, "num_subsets": 800, "num_disjoint_sets": 60},
            11: {"num_elements": 400, "num_subsets": 1000, "num_disjoint_sets": 200},
        },
        "Set Splitting": {
            1: {"num_elements": 5, "num_subsets": 5},
            2: {"num_elements": 10, "num_subsets": 10},
            3: {"num_elements": 10, "num_subsets": 50},
            4: {"num_elements": 10, "num_subsets": 100},
            5: {"num_elements": 10, "num_subsets": 200},
            6: {"num_elements": 100, "num_subsets": 100},
            7: {"num_elements": 100, "num_subsets": 200},
            8: {"num_elements": 10, "num_subsets": 500},
            9: {"num_elements": 10, "num_subsets": 1000},
            10: {"num_elements": 15, "num_subsets": 500},
            11: {"num_elements": 20, "num_subsets": 500},
        },
        "Shortest Common Superstring": {
            1: {"n": 10, "k": 5},
            2: {"n": 20, "k": 10},
            3: {"n": 40, "k": 20},
            4: {"n": 80, "k": 40},
            5: {"n": 100, "k": 50},
            6: {"n": 100, "k": 100},
            7: {"n": 100, "k": 200},
            8: {"n": 200, "k": 200},
            9: {"n": 300, "k": 400},
            10: {"n": 300, "k": 600},
        },
        "Quadratic Diophantine Equations": {
            1: {"low": 1, "high": 50},
            2: {"low": 1, "high": 100},
            3: {"low": 1, "high": 500},
            4: {"low": 1, "high": 1000},
            5: {"low": 1, "high": 5000},
```

```
        6: {"low": 1, "high": 10000},
        7: {"low": 1, "high": 50000},
        8: {"low": 1, "high": 80000},
        9: {"low": 1, "high": 100000},
        10: {"low": 1, "high": 200000},
    },
    "Quadratic Congruences": {
        1: {"min_value": 1, "max_value": 100},
        2: {"min_value": 1, "max_value": 1000},
        3: {"min_value": 1, "max_value": 10000},
        4: {"min_value": 1, "max_value": 50000},
        5: {"min_value": 1, "max_value": 100000},
        6: {"min_value": 1, "max_value": 300000},
        7: {"min_value": 1, "max_value": 500000},
        8: {"min_value": 1, "max_value": 800000},
        9: {"min_value": 1, "max_value": 1000000},
        10: {"min_value": 1, "max_value": 3000000},
    },
    "3-Dimensional Matching (3DM)": {
        1: {"n": 4},
        2: {"n": 8},
        3: {"n": 12},
        4: {"n": 15},
        5: {"n": 20},
        6: {"n": 25},
        7: {"n": 30},
        8: {"n": 40},
        9: {"n": 50},
        10: {"n": 60},
    },
    "Travelling Salesman (TSP)": {
        1: {"num_cities": 5, "target_length": 100},
        2: {"num_cities": 8, "target_length": 100},
        3: {"num_cities": 10, "target_length": 100},
        4: {"num_cities": 12, "target_length": 100},
        5: {"num_cities": 15, "target_length": 100},
        6: {"num_cities": 17, "target_length": 200},
        7: {"num_cities": 20, "target_length": 200},
        8: {"num_cities": 25, "target_length": 200},
        9: {"num_cities": 30, "target_length": 200},
        10: {"num_cities": 40, "target_length": 300},
    },
    "Dominating Set": {
        1: {"num_nodes": 10, "k": 5, "edge_prob": 0.3},
        2: {"num_nodes": 15, "k": 5, "edge_prob": 0.3},
        3: {"num_nodes": 30, "k": 15, "edge_prob": 0.3},
        4: {"num_nodes": 50, "k": 20, "edge_prob": 0.3},
        5: {"num_nodes": 70, "k": 20, "edge_prob": 0.3},
        6: {"num_nodes": 100, "k": 20, "edge_prob": 0.3},
        7: {"num_nodes": 70, "k": 20, "edge_prob": 0.2},
        8: {"num_nodes": 80, "k": 20, "edge_prob": 0.2},
        9: {"num_nodes": 100, "k": 20, "edge_prob": 0.2},
        10: {"num_nodes": 150, "k": 20, "edge_prob": 0.2},
        11: {"num_nodes": 160, "k": 15, "edge_prob": 0.2},
        12: {"num_nodes": 180, "k": 15, "edge_prob": 0.2},
    },
    "Hitting String": {
        1: {"n": 5, "m": 10},
        2: {"n": 5, "m": 20},
        3: {"n": 10, "m": 20},
        4: {"n": 10, "m": 30},
        5: {"n": 10, "m": 40},
        6: {"n": 10, "m": 45},
        7: {"n": 10, "m": 50},
        8: {"n": 10, "m": 55},
        9: {"n": 10, "m": 60},
        10: {"n": 10, "m": 70},
    },
```

```
"Hamiltonian Cycle": {
    1: {"num_nodes": 5, "directed": False},
    2: {"num_nodes": 8, "directed": False},
    3: {"num_nodes": 10, "directed": False},
    4: {"num_nodes": 12, "directed": False},
    5: {"num_nodes": 16, "directed": False},
    6: {"num_nodes": 18, "directed": False},
    7: {"num_nodes": 20, "directed": False},
    8: {"num_nodes": 22, "directed": False},
    9: {"num_nodes": 25, "directed": False},
    10: {"num_nodes": 30, "directed": False},
},
"Bin Packing": {
    1: {"num_items": 10, "bin_capacity": 20, "num_bins": 3},
    2: {"num_items": 20, "bin_capacity": 30, "num_bins": 3},
    3: {"num_items": 30, "bin_capacity": 30, "num_bins": 3},
    4: {"num_items": 40, "bin_capacity": 30, "num_bins": 3},
    5: {"num_items": 50, "bin_capacity": 50, "num_bins": 5},
    6: {"num_items": 60, "bin_capacity": 50, "num_bins": 5},
    7: {"num_items": 70, "bin_capacity": 50, "num_bins": 5},
    8: {"num_items": 80, "bin_capacity": 50, "num_bins": 5},
    9: {"num_items": 80, "bin_capacity": 30, "num_bins": 10},
    10: {"num_items": 100, "bin_capacity": 50, "num_bins": 10},
},
"Exact Cover by 3-Sets (X3C)": {
    1: {"num_elements": 3, "num_subsets": 6},
    2: {"num_elements": 4, "num_subsets": 8},
    3: {"num_elements": 5, "num_subsets": 10},
    4: {"num_elements": 7, "num_subsets": 14},
    5: {"num_elements": 8, "num_subsets": 16},
    6: {"num_elements": 10, "num_subsets": 20},
    7: {"num_elements": 15, "num_subsets": 30},
    8: {"num_elements": 20, "num_subsets": 40},
    9: {"num_elements": 25, "num_subsets": 50},
    10: {"num_elements": 30, "num_subsets": 60},
},
"Minimum Cover": {
    1: {"num_elements": 5, "num_sets": 10, "k": 3},
    2: {"num_elements": 10, "num_sets": 20, "k": 5},
    3: {"num_elements": 10, "num_sets": 30, "k": 5},
    4: {"num_elements": 15, "num_sets": 20, "k": 8},
    5: {"num_elements": 15, "num_sets": 30, "k": 10},
    6: {"num_elements": 20, "num_sets": 40, "k": 10},
    7: {"num_elements": 25, "num_sets": 50, "k": 10},
    8: {"num_elements": 30, "num_sets": 60, "k": 10},
    9: {"num_elements": 35, "num_sets": 70, "k": 10},
    10: {"num_elements": 40, "num_sets": 80, "k": 10},
    11: {"num_elements": 45, "num_sets": 90, "k": 10},
    12: {"num_elements": 50, "num_sets": 100, "k": 10},
    13: {"num_elements": 55, "num_sets": 110, "k": 10},
    14: {"num_elements": 60, "num_sets": 120, "k": 10},
    15: {"num_elements": 65, "num_sets": 130, "k": 10},
    16: {"num_elements": 70, "num_sets": 140, "k": 10},
},
"Graph 3-Colourability (3-COL)": {
    1: {"num_nodes": 5, "num_edges": 8},
    2: {"num_nodes": 8, "num_edges": 12},
    3: {"num_nodes": 10, "num_edges": 20},
    4: {"num_nodes": 15, "num_edges": 25},
    5: {"num_nodes": 15, "num_edges": 30},
    6: {"num_nodes": 15, "num_edges": 40},
    7: {"num_nodes": 20, "num_edges": 40},
    8: {"num_nodes": 20, "num_edges": 45},
    9: {"num_nodes": 30, "num_edges": 60},
    10: {"num_nodes": 30, "num_edges": 80},
},
"Clustering": {
    1: {"num_elements": 6, "b": 10},
```

```
        2: {"num_elements": 10, "b": 10},
        3: {"num_elements": 15, "b": 10},
        4: {"num_elements": 18, "b": 10},
        5: {"num_elements": 20, "b": 10},
        6: {"num_elements": 30, "b": 10},
        7: {"num_elements": 40, "b": 10},
        8: {"num_elements": 50, "b": 10},
        9: {"num_elements": 60, "b": 10},
        10: {"num_elements": 70, "b": 10},
    },
    "Betweenness": {
        1: {"num_element": 3, "num_triples": 1},
        2: {"num_element": 4, "num_triples": 2},
        3: {"num_element": 5, "num_triples": 3},
        4: {"num_element": 6, "num_triples": 4},
        5: {"num_element": 7, "num_triples": 5},
        6: {"num_element": 8, "num_triples": 6},
    },
    "Minimum Sum of Squares": {
        1: {"num_elements": 10, "k": 5},
        2: {"num_elements": 50, "k": 8},
        3: {"num_elements": 100, "k": 8},
        4: {"num_elements": 100, "k": 5},
        5: {"num_elements": 100, "k": 4},
        6: {"num_elements": 100, "k": 3},
        7: {"num_elements": 200, "k": 10},
        8: {"num_elements": 200, "k": 4},
        9: {"num_elements": 200, "k": 3},
        10: {"num_elements": 300, "k": 3},
    },
    "Bandwidth": {
        1: {"num_nodes": 3, "bandwidth": 2},
        2: {"num_nodes": 4, "bandwidth": 2},
        3: {"num_nodes": 5, "bandwidth": 3},
        4: {"num_nodes": 6, "bandwidth": 3},
        5: {"num_nodes": 5, "bandwidth": 2},
        6: {"num_nodes": 7, "bandwidth": 3},
        7: {"num_nodes": 6, "bandwidth": 2},
        8: {"num_nodes": 8, "bandwidth": 3},
        9: {"num_nodes": 7, "bandwidth": 2},
        10: {"num_nodes": 8, "bandwidth": 2},
    },
    "Maximum Leaf Spanning Tree": {
        1: {"num_nodes": 5, "target_leaves": 2},
        2: {"num_nodes": 10, "target_leaves": 5},
        3: {"num_nodes": 20, "target_leaves": 10},
        4: {"num_nodes": 30, "target_leaves": 20},
        5: {"num_nodes": 40, "target_leaves": 30},
        6: {"num_nodes": 60, "target_leaves": 50},
        7: {"num_nodes": 70, "target_leaves": 60},
        8: {"num_nodes": 80, "target_leaves": 65},
        9: {"num_nodes": 90, "target_leaves": 75},
        10: {"num_nodes": 100, "target_leaves": 80},
    },
}
```

**Solution Errors.** There are two fundamental error categories: problem-independent errors and problem-dependent errors. Problem-independent errors are general errors that arise from external factors unrelated to the problem's intrinsic characteristics and all problems have these types of errors. Problem-independent errors include JSON ERROR (JSON not found or JSON parsing errors), and VERIFICATION ERROR (output format mismatches or structural validation failures). Problem-dependent errors originate from the problem's inherent complexity, which are defined based on problem specificity. A comprehensive illustration of the errors is displayed in Table 7.

Table 7: A comprehensive illustration of errors.

| Problem | Error Type | Description |
|---|---|---|
| | JSON ERROR | JSON not found. |
| | VERIFICATION ERROR | Wrong output format. |
| 3SAT | ERROR 1 | The solution length mismatches the number of variables. |
| | ERROR 2 | Some clauses are not satisfied. |
| Vertex Cover | ERROR 1 | Wrong solution format. |
| | ERROR 2 | The cover is empty. |
| | ERROR 3 | Invalid vertex index, i.e., above the max or below the min. |
| | ERROR 4 | The cover size exceeds the limit. |
| | ERROR 5 | Some edges are not covered. |
| 3DM | ERROR 1 | Not all triples in the matching are in the original set. |
| | ERROR 2 | The size of matching is wrong |
| | ERROR 3 | The elements in the matching are not mutually exclusive. |
| TSP | ERROR 1 | Tour length mismatches number of cities. |
| | ERROR 2 | Invalid city index, i.e., above the max or below the min. |
| | ERROR 3 | There exists cities not be visited exactly once. |
| | ERROR 4 | Tour length exceeds target length. |
| Hamiltonian Cycle | ERROR 1 | Path length is wrong. |
| | ERROR 2 | Path does not return to start. |
| | ERROR 3 | Not all vertices visited exactly once. |
| | ERROR 4 | There exists invalid vertex in path. |
| | ERROR 5 | There exists invalid edges in path. |
| 3-COL | ERROR 1 | The two nodes of an edge have the same color |
| Bin Packing | ERROR 1 | Solution length mismatches the number of items. |
| | ERROR 2 | Invalid bin index. |
| | ERROR 3 | The total size exceeds bin capacity. |
| Max Leaf Span Tree | ERROR 1 | Solution length mismatches the number of vertices. |
| | ERROR 2 | There exists invalid edges in solution. |
| | ERROR 3 | The solution does not have exactly one root. |
| | ERROR 4 | The solution doesn't span all vertices. |
| | ERROR 5 | The number of leaves in the solution is less than target. |
| QDE | ERROR 1 | Solution length mismatches the number of integers. |
| | ERROR 2 | There exists non-positive values in the solution. |
| | ERROR 3 | The equation does not hold. |
| Min Sum Square | ERROR 1 | Solution length mismatches the number of elements. |
| | ERROR 2 | The number of subsets exceeds the set limit. |
| | ERROR 3 | The sum exceeds the limit $J$. |
| Superstring | ERROR 1 | Wrong solution format. |
| | ERROR 2 | The solution length exceeds the limit. |
| | ERROR 3 | Some string is not the substring of the solution. |
| Bandwidth | ERROR 1 | Layout length mismatches the number of vertices. |
| | ERROR 2 | Layout is not a permutation of vertices. |
| | ERROR 3 | There exists edge exceeds the bandwidth limit. |

## C.2    Solver Suite: *npsolver*

We introduce *npsolver*, a solver suite that provides a unified interface for both online (API-based) and offline (local) models. The unified interface includes: i) *Prompt Generation*, which constructs problem-specific prompts dynamically using the designed prompt templates shown in Appendix D, including problem descriptions, in-context examples, and target problems; ii) *LLM Completion*, which invokes either online or offline LLMs to generate responses from the constructed prompts; iii) *Solution Extraction*, which designs regular expressions to parse JSON outputs from the LLMs' responses, ensuring all online and offline LLMs Use the same JSON validation pipeline; iv) *Error Reporting*, which standardizes error messages. Through the unified interface, *npsolver* enables both online and offline models to share a common workflow. Through this unified pipeline, *npsolver* enables consistent evaluation and analysis for both online and offline models. For each problem, difficulty level, and model, *npsolver* stores detailed records—including the problem instance, example solutions, full LLM responses, extracted solutions, input/output token counts, error messages, solution correctness, and reasons for failure—in a pickle file to facilitate failure case analysis. The list of models integrated in *npsolver* is shown in Table 8.

Table 8: Online and offline models considered in this paper via *npsolver*.

| Type | Models | Version | Provider |
|---|---|---|---|
| Online | GPT-4o-mini | gpt-4o-mini-2024-07-18 | OpenAI |
| | GPT-4o | gpt-4o-2024-08-06 | OpenAI |
| | o1-mini | o1-mini-2024-09-12 | OpenAI |
| | o3-mini | o3-mini-2025-01-31 | OpenAI |
| | DeepSeek-V3 | deepseek-v3-241226 | Huoshan |
| | DeepSeek-V3-2503 | deepseek-v3-250324 | Huoshan |
| | DeepSeek-R1 | deepseek-r1-250120 | Huoshan |
| | Claude-3.7-Sonnet | claude-3-7-sonnet-20250219 | Anthropic |
| Offline | QwQ-32B | Qwen/QwQ-32B | N/A |
| | DeepSeek-R1-32B | deepseek-ai/DeepSeek-R1-Distill-Qwen-32B | N/A |

**Online.** The online state-of-the-art LLMs, e.g., o1/o3-mini and DeepSeek-v3/R1, can be accessed through APIs without local computational overhead. However, these online models have dependency on network stability and API costs with token usage. *npsolver* supports multiple providers, e.g., OpenAI, through modular API clients. We implement efficient batch processing with LiteLLM, which minimizes the latency during parallel problem-solving.

**Offline.** Open-weight LLMs, e.g., QwQ-32B and Deepseek-R1-32B, can be accessed by deploying them locally. This allows for GPU-accelerated, high-throughput inference while avoiding API-related costs. Offline models are deployed using vLLM, with hyperparameters—such as temperature and maximum token length—manually configured according to their official technical documentation.

### C.3 Evaluation Suite: *npeval*

*npeval* employs a statistically rigorous sampling strategy. For each difficulty, the aggregated performance over 3 different independent seeds, with 30 samples generated per seed, aligning with the minimum sample size for reliable statistical analysis (Hogg et al., 1977), are considered. This sampling design, i.e., sampling 90 instances total per difficulty level for each problem, balances budget constraints while mitigating instance-specific variance.

**Evaluation Metrics.** *rliable* (Agarwal et al., 2021) is an open-source Python library designed to enable statistically robust evaluation of reinforcement learning and machine learning benchmarks. Inspired by *rliable*, *npeval* provides the following 4 evaluation aggregate metrics:

- Mean: Mean is a standard evaluation metric that treats each score equally and calculates the overall mean across runs and tasks.
- Interquartile Mean (IQM): IQM trims extreme values and computes the interquartile mean across runs and tasks to smooth out the randomness in responses. IQM highlights the consistency of the performance and complements metrics like mean/median to avoid outlier skew.
- Median: Median represents the middle value of the scores by calculating the median of the average scores per task across all runs, which is unaffected by extreme values.
- Optimality Gap (OG): OG measures the average shortfall of scores below a predefined threshold $\gamma$, where all scores above $\gamma$ are clipped to $\gamma$, so as to quantify and penalize the underperformance, making it less susceptible to outliers compared to mean scores.

To quantify uncertainty in aggregate metrics, e.g. IQM, *npeval* employs stratified bootstrap confidence intervals (SBCIs) (Efron, 1979; 1987) for the performance interval estimation. SBCIs use stratified resampling within predefined strata, e.g., difficulty levels, to preserve the hierarchical structure of the evaluation data, reduce bias, and provide statistically sound interval estimates.

**Comprehensive Analysis.** Based on evaluation metrics, *npeval* provides a comprehensive analysis of the LLMs' performance over the problems and difficulty levels, including the full results for each problem, each model and each level (Appendix G), the performance over different problems (Appendix H), the analysis of both prompt and completion tokens of LLMs (Appendix I), the analysis of the number of "aha moments" during the DeepSeek-R1 reasoning (Guo et al., 2025) (Appendix J), an illustration of errors over problems (Table 7) with detailed error analysis (Appendix K), considering both the solution errors, i.e., the errors returned by *npgym*, and the reasoning errors, i.e., the errors produced in the internal reasoning process of LLMs, which enables the identification of the failure cases (Appendix L).

## D   Prompts and Responses

**Prompts.**   In this section, we carefully design the prompt template of **NPPC** for LLMs to be simple, general, and consistent across different problems. The prompt template includes:

- Problem description: where a concise definition of the NPC problem is provided, including the problem name, the input, and the question to be solved.
- Examples: where one or multiple in context examples, defined as problem-solution pairs, are listed, demonstrating the expected solutions, i.e., answer correctness and format, for specific instances. These examples guide LLMs to generate the responses with the required format.
- Problem to solve: a target instance that requires LLMs to generate the solution.
- Instruction: which provides a directive to output answers in JSON format.

```
nppc_template = """
# <problem_name> Problem Description:
<problem_description>

# Examples:
<in_context_examples>
# Problem to Solve:
Problem: <problem_to_solve>

# Instruction:
Now please solve the above problem. Reason step by step and present your answer in the
    "solution" field in the following json format:
'''json
{"solution": "___" }
'''

"""

example_and_solution = """Problem: <example_problem>
{"solution": <example_solution>}
"""
```

**Responses.**  We extract the answers from the LLMs' responses and the code is displayed below:

```python
def extract_solution_from_response(response):
    # find the json code
    match = re.findall(r"```json\n(.*?)\n```", response, re.DOTALL)

    if not match:
        match = re.findall(r"json\s*({[^{}]*})", response, re.DOTALL)
    if not match:
        match = re.findall(r"\{[^{}]*\}", response, re.DOTALL)

    if match:
        json_str = match[-1]
        try:
            # remove the single line comment
            json_str = re.sub(r"//.*$", "", json_str, flags=re.MULTILINE)
            # remove the multiple line comment
            json_str = re.sub(r"/\*[\s\S]*?\*/", "", json_str)
            data = json.loads(json_str)
            answer = data["solution"]
            return answer, None
        except (json.JSONDecodeError, KeyError, SyntaxError) as e:
            print(f"Error parsing JSON or answer field: {e}")
            return None, f"Error parsing JSON or answer field: {e}"
    else:
        print("No JSON found in the text.")
        return None, "JSON Error: No JSON found in the text."
```

This extraction function employs a robust, multi-stage pattern matching approach to handle various response formats. Specifically, this function progressively relaxes format requirements through three regex patterns:

- First tries to find content between triple quotes with "json" marker,
- If that fails, looks for "json" followed by content in curly braces,
- If both fail, simply looks for any content between curly braces.

This flexible parsing template effectively handles format variations produced by different models. For instance, it accommodates models like QwQ-32B that frequently omit format prefixes or deviate from strict formatting instructions. By tolerating minor format violations, this approach ensures that evaluation focuses primarily on models' reasoning capabilities rather than their adherence to output formatting conventions. Only when all three attempts fail, the JSON error is raised, indicating a genuine failure to produce a feasible solution.

# E   List of NP-complete Problems

**Problem 1.**   • **Name**: 3-Satisfiability (3SAT)

 • **Input**: A set of $m$ clauses $\{C_1, C_2, \ldots, C_m\}$ - over a set of $n$ Boolean valued variables $X_n = \{x_1, x_2, \ldots, x_n\}$, such that each clause depends on exactly three distinct variables from $X_n$. A clause being a Boolean expression of the form $y_i \wedge y_j \wedge y_k$ where each $y$ is of the form $x$ or $\neg x$ (i.e. negation of $x$) with $x$ being some variable in $X_n$. For example if $n = 4$ and $m = 3$, a possible instance could be the (set of) Boolean expressions: $C_1 = (x_1 \wedge (\neg x_2) \wedge (\neg x_3))$, $C_2 = (x_2 \wedge x_3 \wedge (\neg x_4))$, $C_3 = ((\neg x_1) \wedge x_3 \wedge x_4)$.

 • **Question**: Can each variable $x_i$ of $X_n$ be assigned a Boolean value $\alpha_i \in \{\text{true}, \text{false}\}$ in such a way that every clause evaluates to the Boolean result true under the assignment $\langle x_i := \alpha_i, i \in \{1, \ldots, n\}\rangle$?

**Problem 2.**   • **Name**: Graph 3-Colourability (3-COL)

 • **Input**: An $n$-node undirected graph $G = (V, E)$ with node set $V$ and edge set $E$.

 • **Question**: Can each node of $G = (V, E)$ be assigned exactly one of three colours - Red, Blue, Green - in such a way that no two nodes which are joined by an edge, are assigned the same colour?

**Problem 3.**   • **Name**: Clique

 • **Input**: An $n$-node undirected graph $G = (V, E)$ with node set $V$ and edge set $E$; a positive integer $k$ with $k \leq n$.

 • **Question**: Does $G$ contain a $k$-clique, i.e. a subset $W$ of the nodes $V$ such that $W$ has size $k$ and for each distinct pair of nodes $u, v$ in $W$, $\{u, v\}$ is an edge of $G$?

**Problem 4.**   • **Name**: Vertex Cover

 • **Input**: An $n$-node undirected graph $G = (V, E)$ with node set $V$ and edge set $E$; a positive integer $k$ with $k \leq n$.

 • **Question**: Is there a subset $W$ of $V$ having size at most $k$ and such that for every edge $\{u, v\}$ in $E$ at least one of $u$ and $v$ belongs to $W$?

**Problem 5.**   • **Name**: Quadratic Diophantine Equations

 • **Input**: Positive integers $a$, $b$, and $c$.

 • **Question**: Are there two positive integers $x$ and $y$ such that $(a * x * x) + (b * y) = c$?

**Problem 6.**   • **Name**: Shortest Common Superstring

 • **Input**: A finite set $R = \{r_1, r_2, \ldots, r_m\}$ of strings (sequences of symbols); positive integer $k$.

 • **Question**: Is there a string $w$ of length at most $k$ such that every string in $R$ is a substring of $w$, i.e., for each $r$ in $R$, $w$ can be decomposed as $w = w_0 r w_1$ where $w_0$, $w_1$ are (possibly empty) strings?

**Problem 7.**   • **Name**: Bandwidth

 • **Input**: $n$-node undirected graph $G = (V, E)$; positive integer $k \leq n$.

 • **Question**: Is there a linear ordering of $V$ with bandwidth at most $k$, i.e., a one-to-one function $f : V \to \{0, 1, 2, ..., n - 1\}$ such that for all edges $u, v$ in $G$, $|f(u) - f(v)| \leq k$?

**Problem 8.**   • **Name**: Maximum Leaf Spanning Tree

 • **Input**: $n$-node undirected graph $G = (V, E)$; positive integer $k \leq n$.

 • **Question**: Does $G$ have a spanning tree in which at least $k$ nodes have degree 1?

**Problem 9.**   • **Name**: Independent Set

 • **Input**: $n$-node undirected graph $G = (V, E)$; positive integer $k \leq n$.

- **Question**: Does $G$ have an independent set of size at least $k$, i.e., a subset $W$ of at least $k$ nodes from $V$ such that no pair of nodes in $W$ is joined by an edge in $E$?

**Problem 10.**
- **Name**: Hamiltonian Cycle

- **Input**: n-node graph $G = (V, E)$.

- **Question**: Is there a cycle in $G$ that visits every node in $V$ exactly once and returns to the starting node, and thus contains exactly $n$ edge

**Problem 11.**
- **Name**: Travelling Salesman

- **Input**: A set $C$ of n cities $\{c_1, \ldots, c_n\}$; a positive integer distance $d(i, j)$ for each pair of cities $(c_i, c_j), i < j, i, j \in \{1, \ldots, n\}$; a positive integer $B$ representing the maximum allowed travel distance.

- **Question**: Is there an ordering $\langle \pi(1), \pi(2), ..., \pi(n) \rangle$ of the $n$ cities such that the total travel distance, calculated as the sum of $d(\pi(i), \pi(i+1))$ for $i = 1$ to $n - 1$, plus $d(\pi(n), \pi(1))$, is at most $B$?

**Problem 12.**
- **Name**: Dominating Set

- **Input**: An undirected graph $G(V, E)$ with n nodes; a positive integer $k$ where $k \leq n$.

- **Question**: Does $G$ contain a dominating set of size at most $k$, i.e. a subset $W$ of $V$ containing at most $k$ nodes such that every node $u$ in $V - W$ (i.e. in $V$ but not in $W$) has at least one neighbor $w$ in $W$ where $u, w$ is an edge in $E$?

**Problem 13.**
- **Name**: 3-Dimensional Matching (3DM)

- **Input**: 3 disjoint sets $X$, $Y$, and $Z$, each containing exactly $n$ elements; a set $M$ of $m$ triples $\{(x_i, y_i, z_i) : 1 \leq i \leq m\}$ such that $x_i$ is in $X$, $y_i$ in $Y$, and $z_i$ in $Z$, i.e. $M$ is a subset of $X \times Y \times Z$.

- **Question**: Does $M$ contain a matching, i.e., is there a subset $Q$ of $M$ such that $|Q| = n$ and for all distinct pairs of triples $(u, v, w)$ and $(x, y, z)$ in $Q$ it holds that $u \neq x$ and $v \neq y$ and $w \neq z$?

**Problem 14.**
- **Name**: Set Splitting

- **Input**: A finite set $S$; A collection $C_1, \ldots, C_m$ of subsets of $S$.

- **Question**: Can $S$ be partitioned into two disjoint subsets - $S1$ and $S2$ - such that for each set $C_i$ it holds that $C_i$ is not a subset of $S_1$ and $C_i$ is not a subset of $S_2$?

**Problem 15.**
- **Name**: Set Packing

- **Input**: A collection $C = (C_1, \ldots, C_m)$ of finite sets; a positive integer $k \leq m$.

- **Question**: Are there $k$ sets - $D_1, \ldots, D_k$ - from the collection $C$ such that for all $1 \leq i < j \leq k$, $D_i$ and $D_j$ have no common elements?

**Problem 16.**
- **Name**: Exact Cover by 3-Sets (X3C)

- **Input**: A finite set $X$ containing exactly $3n$ elements; a collection $C$ of subsets of $X$ each of which contains exactly 3 elements.

- **Question**: Does $C$ contain an exact cover for $X$, i.e., a sub-collection of 3-element sets $D = (D_1, \ldots, D_n)$ such that each element of $X$ occurs in exactly one subset in $D$?

**Problem 17.**
- **Name**: Minimum Cover

- **Input**: A finite set $S$; A collection $C = (C_1, \ldots, C_m)$ of subsets of $S$; a positive integer $k \leq m$.

- **Question**: Does $C$ contain a cover for $S$ comprising at most $k$ subsets, i.e., a collection $D = (D_1, \ldots, D_t)$, where $t \leq k$, each $D_i$ is a set in $C$, and such that every element in $S$ belongs to at least one set in $D$?

**Problem 18.**
- **Name**: Partition

- **Input**: Finite set $A$; for each element $a$ in $A$ a positive integer size $s(a)$.

- **Question**: Can $A$ be partitioned into 2 disjoint sets $A_1$ and $A_2$ in a such a way that $\sum_{a \in A_1} s(a) = \sum_{a \in A_2} s(a)$?

**Problem 19.**  
- **Name**: Subset Sum

- **Input**: Finite set $A$; for each element $a \in A$ a positive integer size $s(a)$; a positive integer $K$.

- **Question**: Is there a subset $B$ of $A$ such that $\sum_{a \in B} s(a) = K$?

**Problem 20.**  
- **Name**: Minimum Sum of Squares

- **Input**: A set $A$ of $n$ elements; for each element $a \in A$ a positive integer size $s(a)$; positive integers $k \leq n$ and $J$.

- **Question**: Can $A$ be partitioned into $k$ disjoint sets $A_1, \ldots, A_k$ such that $\sum_{i=1}^{k} (\sum_{x \in A_i} s(x))^2 <= J$?

**Problem 21.**  
- **Name**: Bin Packing

- **Input**: A finite set $U$ of $m$ items; for each item $u$ in $U$ a positive integer size $s(u)$; positive integers $B$ (bin capacity) and $k$, where $k \leq m$.

- **Question**: Can $U$ be partitioned into $k$ disjoint sets $U_1, \ldots, U_k$ such that the total size of the items in each subset $U_i$ (for $1 \leq i \leq k$) does not exceed $B$?

**Problem 22.**  
- **Name**: Hitting String

- **Input**: Finite set $S = \{s_1, \ldots, s_m\}$ each $s_i$ being a string of $n$ symbols over $\{0, 1, *\}$.

- **Question**: Is there a binary string $x = x_1 x_2 \ldots x_n$ of length $n$ such that for each $s_j \in S$, $s_j$ and $x$ agree in at least one position?

**Problem 23.**  
- **Name**: Quadratic Congruences

- **Input**: Positive integers $a$, $b$, and $c$.

- **Question**: Is there a positive integer $x$ whose value is less than $c$ and is such that $x^2 \mod b == a$, i.e., the remainder when $x^2$ is divided by $b$ is equal to $a$?

**Problem 24.**  
- **Name**: Betweenness

- **Input**: A finite set $A$ of size n; a set $C$ of ordered triples, $(a, b, c)$, of distinct elements from $A$.

- **Question**: Is there a one-to-one function, $f : A \to \{0, 1, 2, ..., n-1\}$ such that for each triple $(a, b, c)$ in $C$ it holds that either $f(a) < f(b) < f(c)$ or $f(c) < f(b) < f(a)$?

**Problem 25.**  
- **Name**: Clustering

- **Input**: Finite set $X$; for each pair of elements $x$ and $y$ in $X$, a positive integer distance $d(x, y)$; positive integer $B$.

- **Question**: Is there a partition of $X$ into 3 disjoint sets - $X_1, X_2, X_3$ - with which: for each set $X_i, i \in \{1, 2, 3\}$, for all pairs $x$ and $y$ in $X_i$, it holds that $d(x, y) \leq B$?

## F   Hyperparameters

The hyperparameters used for benchmarking are listed in Table 9. For both offline and online-deployed models, accuracy is averaged over three seeds and 30 trials per difficulty level per task. Each model is allowed up to three attempts to mitigate the impact of API connection issues. For offline models, we follow the recommended sampling parameters from the technical reports of Deepseek-R1-32B and QwQ-32B for vLLM deployment.

Table 9: Hyperparameters

| Type | Hyperparameter | Value |
|---|---|---|
| Basic | seeds | 42, 53, 64 |
| | n_shots | 1 |
| | n_trials | 30 |
| | batch_size | 10 |
| | max_tries | 3 |
| Offline Model | temperature | 0.6 |
| | top_p | 0.95 |
| | max_tokens | 7500 |
| | gpu_memory_utilization | 0.8 |

## G  Full Results over Problems

In this section, we present the full results over problems, as displayed in Figure 6. For each element in the table $x_b^a$, $x$ is the value of IQM and $a$ and $b$ are the upper and lower values of the CI, respectively.

Table 10: 3SAT

| | 1 | 2 | 3 | 4 | 5 | 6 | 7 | 8 | 9 | 10 |
|---|---|---|---|---|---|---|---|---|---|---|
| QwQ-32B | $0.94_{0.90}^{1.00}$ | $1.00_{1.00}^{1.00}$ | $0.56_{0.53}^{0.60}$ | $0.11_{0.07}^{0.13}$ | $0.00_{0.00}^{0.00}$ | $0.00_{0.00}^{0.00}$ | $0.00_{0.00}^{0.00}$ | $0.00_{0.00}^{0.00}$ | $0.00_{0.00}^{0.00}$ | $0.00_{0.00}^{0.00}$ |
| DeepSeek-R1-32B | $0.83_{0.73}^{0.90}$ | $0.52_{0.40}^{0.70}$ | $0.32_{0.30}^{0.37}$ | $0.19_{0.13}^{0.27}$ | $0.13_{0.07}^{0.23}$ | $0.02_{0.00}^{0.03}$ | $0.00_{0.00}^{0.00}$ | $0.00_{0.00}^{0.00}$ | $0.00_{0.00}^{0.00}$ | $0.00_{0.00}^{0.00}$ |
| GPT-4o-mini | $0.84_{0.83}^{0.87}$ | $0.27_{0.20}^{0.30}$ | $0.17_{0.10}^{0.27}$ | $0.08_{0.03}^{0.10}$ | $0.02_{0.00}^{0.03}$ | $0.00_{0.00}^{0.00}$ | $0.00_{0.00}^{0.00}$ | $0.00_{0.00}^{0.00}$ | $0.00_{0.00}^{0.00}$ | $0.00_{0.00}^{0.00}$ |
| GPT-4o | $0.94_{0.93}^{0.97}$ | $0.51_{0.47}^{0.57}$ | $0.43_{0.40}^{0.47}$ | $0.22_{0.20}^{0.27}$ | $0.09_{0.00}^{0.17}$ | $0.01_{0.00}^{0.03}$ | $0.01_{0.00}^{0.03}$ | $0.00_{0.00}^{0.00}$ | $0.00_{0.00}^{0.00}$ | $0.00_{0.00}^{0.00}$ |
| Claude-3.7-Sonnet | $1.00_{1.00}^{1.00}$ | $0.89_{0.80}^{0.93}$ | $0.62_{0.60}^{0.67}$ | $0.54_{0.50}^{0.60}$ | $0.36_{0.27}^{0.47}$ | $0.19_{0.13}^{0.27}$ | $0.14_{0.03}^{0.23}$ | $0.08_{0.03}^{0.10}$ | $0.03_{0.00}^{0.07}$ | $0.02_{0.00}^{0.03}$ |
| DeepSeek-V3 | $0.94_{0.93}^{0.97}$ | $0.78_{0.60}^{0.90}$ | $0.38_{0.33}^{0.40}$ | $0.34_{0.17}^{0.43}$ | $0.21_{0.17}^{0.27}$ | $0.06_{0.03}^{0.10}$ | $0.01_{0.00}^{0.03}$ | $0.00_{0.00}^{0.00}$ | $0.00_{0.00}^{0.00}$ | $0.00_{0.00}^{0.00}$ |
| DeepSeek-V3-2503 | $1.00_{1.00}^{1.00}$ | $0.98_{0.97}^{1.00}$ | $0.89_{0.83}^{0.97}$ | $0.68_{0.60}^{0.80}$ | $0.53_{0.47}^{0.63}$ | $0.38_{0.30}^{0.43}$ | $0.28_{0.23}^{0.33}$ | $0.12_{0.03}^{0.23}$ | $0.08_{0.03}^{0.17}$ | $0.03_{0.00}^{0.03}$ |
| DeepSeek-R1 | $1.00_{1.00}^{1.00}$ | $1.00_{1.00}^{1.00}$ | $0.99_{0.97}^{1.00}$ | $0.98_{0.93}^{1.00}$ | $0.97_{0.93}^{1.00}$ | $0.91_{0.87}^{0.97}$ | $0.83_{0.63}^{0.93}$ | $0.64_{0.63}^{0.67}$ | $0.23_{0.20}^{0.27}$ | $0.13_{0.10}^{0.17}$ |
| o1-mini | $0.92_{0.90}^{0.93}$ | $0.91_{0.87}^{0.97}$ | $0.92_{0.90}^{0.97}$ | $0.81_{0.77}^{0.87}$ | $0.67_{0.60}^{0.77}$ | $0.20_{0.10}^{0.37}$ | $0.03_{0.03}^{0.03}$ | $0.00_{0.00}^{0.00}$ | $0.00_{0.00}^{0.00}$ | $0.00_{0.00}^{0.00}$ |
| o3-mini | $0.93_{0.90}^{0.97}$ | $0.82_{0.77}^{0.87}$ | $0.72_{0.63}^{0.83}$ | $0.77_{0.70}^{0.83}$ | $0.82_{0.80}^{0.83}$ | $0.71_{0.67}^{0.77}$ | $0.60_{0.53}^{0.70}$ | $0.30_{0.20}^{0.43}$ | $0.13_{0.10}^{0.17}$ | $0.12_{0.03}^{0.17}$ |

Table 11: Vertex Cover

| | 1 | 2 | 3 | 4 | 5 | 6 | 7 | 8 | 9 | 10 |
|---|---|---|---|---|---|---|---|---|---|---|
| QwQ-32B | $1.00_{1.00}^{1.00}$ | $0.99_{0.97}^{1.00}$ | $0.93_{0.90}^{0.97}$ | $0.50_{0.37}^{0.60}$ | $0.00_{0.00}^{0.00}$ | $0.00_{0.00}^{0.00}$ | $0.00_{0.00}^{0.00}$ | $0.00_{0.00}^{0.00}$ | $0.00_{0.00}^{0.00}$ | $0.00_{0.00}^{0.00}$ |
| DeepSeek-R1-32B | $0.91_{0.83}^{1.00}$ | $0.92_{0.90}^{0.93}$ | $0.81_{0.73}^{0.87}$ | $0.52_{0.43}^{0.60}$ | $0.03_{0.00}^{0.07}$ | $0.02_{0.00}^{0.03}$ | $0.00_{0.00}^{0.00}$ | $0.00_{0.00}^{0.00}$ | $0.00_{0.00}^{0.00}$ | $0.00_{0.00}^{0.00}$ |
| GPT-4o-mini | $0.94_{0.87}^{1.00}$ | $0.67_{0.57}^{0.80}$ | $0.37_{0.27}^{0.43}$ | $0.18_{0.10}^{0.23}$ | $0.00_{0.00}^{0.00}$ | $0.00_{0.00}^{0.00}$ | $0.00_{0.00}^{0.00}$ | $0.00_{0.00}^{0.00}$ | $0.00_{0.00}^{0.00}$ | $0.00_{0.00}^{0.00}$ |
| GPT-4o | $0.96_{0.90}^{1.00}$ | $0.88_{0.83}^{0.90}$ | $0.78_{0.67}^{0.87}$ | $0.60_{0.57}^{0.63}$ | $0.01_{0.00}^{0.03}$ | $0.03_{0.00}^{0.07}$ | $0.00_{0.00}^{0.00}$ | $0.00_{0.00}^{0.00}$ | $0.00_{0.00}^{0.00}$ | $0.00_{0.00}^{0.00}$ |
| Claude-3.7-Sonnet | $1.00_{1.00}^{1.00}$ | $0.97_{0.90}^{1.00}$ | $0.97_{0.93}^{1.00}$ | $0.90_{0.90}^{0.90}$ | $0.53_{0.47}^{0.57}$ | $0.37_{0.30}^{0.47}$ | $0.37_{0.23}^{0.50}$ | $0.26_{0.20}^{0.30}$ | $0.14_{0.10}^{0.17}$ | $0.04_{0.00}^{0.07}$ |
| DeepSeek-V3 | $0.92_{0.87}^{1.00}$ | $0.97_{0.93}^{1.00}$ | $0.96_{0.93}^{0.97}$ | $0.89_{0.83}^{0.93}$ | $0.34_{0.23}^{0.43}$ | $0.14_{0.10}^{0.20}$ | $0.06_{0.03}^{0.10}$ | $0.03_{0.00}^{0.07}$ | $0.03_{0.00}^{0.07}$ | $0.01_{0.00}^{0.03}$ |
| DeepSeek-V3-2503 | $1.00_{1.00}^{1.00}$ | $1.00_{1.00}^{1.00}$ | $1.00_{1.00}^{1.00}$ | $0.87_{0.83}^{0.90}$ | $0.28_{0.10}^{0.43}$ | $0.37_{0.23}^{0.50}$ | $0.27_{0.23}^{0.33}$ | $0.09_{0.07}^{0.13}$ | $0.09_{0.07}^{0.10}$ | $0.01_{0.00}^{0.03}$ |
| DeepSeek-R1 | $1.00_{1.00}^{1.00}$ | $1.00_{1.00}^{1.00}$ | $1.00_{1.00}^{1.00}$ | $1.00_{1.00}^{1.00}$ | $0.91_{0.87}^{0.97}$ | $0.77_{0.70}^{0.87}$ | $0.41_{0.33}^{0.47}$ | $0.18_{0.13}^{0.20}$ | $0.13_{0.08}^{0.20}$ | $0.06_{0.00}^{0.10}$ |
| o1-mini | $0.74_{0.73}^{0.77}$ | $0.77_{0.73}^{0.80}$ | $0.78_{0.70}^{0.83}$ | $0.91_{0.87}^{0.93}$ | $0.58_{0.43}^{0.70}$ | $0.31_{0.27}^{0.33}$ | $0.13_{0.10}^{0.17}$ | $0.13_{0.03}^{0.27}$ | $0.08_{0.07}^{0.10}$ | $0.02_{0.00}^{0.07}$ |
| o3-mini | $0.82_{0.70}^{0.90}$ | $0.89_{0.83}^{0.93}$ | $0.89_{0.83}^{0.93}$ | $0.80_{0.73}^{0.90}$ | $0.59_{0.53}^{0.70}$ | $0.52_{0.50}^{0.57}$ | $0.19_{0.10}^{0.27}$ | $0.13_{0.03}^{0.23}$ | $0.11_{0.03}^{0.17}$ | $0.07_{0.03}^{0.10}$ |

Table 12: Superstring

| | 1 | 2 | 3 | 4 | 5 | 6 | 7 | 8 | 9 | 10 |
|---|---|---|---|---|---|---|---|---|---|---|
| QwQ-32B | $1.00_{1.00}^{1.00}$ | $0.92_{0.87}^{0.97}$ | $0.28_{0.20}^{0.33}$ | $0.19_{0.13}^{0.23}$ | $0.17_{0.10}^{0.23}$ | $0.06_{0.00}^{0.13}$ | $0.08_{0.03}^{0.13}$ | $0.01_{0.00}^{0.03}$ | $0.00_{0.00}^{0.00}$ | $0.00_{0.00}^{0.00}$ |
| DeepSeek-R1-32B | $0.58_{0.33}^{0.70}$ | $0.24_{0.10}^{0.40}$ | $0.16_{0.07}^{0.23}$ | $0.12_{0.07}^{0.23}$ | $0.10_{0.07}^{0.17}$ | $0.03_{0.03}^{0.03}$ | $0.02_{0.00}^{0.03}$ | $0.00_{0.00}^{0.00}$ | $0.00_{0.00}^{0.00}$ | $0.00_{0.00}^{0.00}$ |
| GPT-4o-mini | $0.32_{0.20}^{0.47}$ | $0.08_{0.03}^{0.13}$ | $0.02_{0.00}^{0.03}$ | $0.01_{0.00}^{0.03}$ | $0.00_{0.00}^{0.00}$ | $0.01_{0.00}^{0.03}$ | $0.00_{0.00}^{0.00}$ | $0.01_{0.00}^{0.03}$ | $0.00_{0.00}^{0.00}$ | $0.00_{0.00}^{0.00}$ |
| GPT-4o | $0.81_{0.77}^{0.83}$ | $0.47_{0.30}^{0.57}$ | $0.11_{0.07}^{0.17}$ | $0.10_{0.03}^{0.17}$ | $0.06_{0.03}^{0.10}$ | $0.16_{0.03}^{0.27}$ | $0.06_{0.03}^{0.10}$ | $0.12_{0.03}^{0.13}$ | $0.07_{0.03}^{0.10}$ | $0.03_{0.00}^{0.07}$ |
| Claude-3.7-Sonnet | $0.99_{0.97}^{1.00}$ | $0.97_{0.93}^{1.00}$ | $0.78_{0.70}^{0.90}$ | $0.51_{0.47}^{0.57}$ | $0.68_{0.60}^{0.77}$ | $0.74_{0.70}^{0.80}$ | $0.77_{0.73}^{0.80}$ | $0.88_{0.83}^{0.93}$ | $0.82_{0.77}^{0.90}$ | $0.74_{0.70}^{0.80}$ |
| DeepSeek-V3 | $0.80_{0.73}^{0.83}$ | $0.52_{0.37}^{0.67}$ | $0.49_{0.43}^{0.53}$ | $0.46_{0.40}^{0.53}$ | $0.44_{0.40}^{0.53}$ | $0.40_{0.30}^{0.57}$ | $0.24_{0.13}^{0.37}$ | $0.22_{0.17}^{0.27}$ | $0.08_{0.03}^{0.10}$ | $0.02_{0.00}^{0.03}$ |
| DeepSeek-V3-2503 | $0.99_{0.97}^{1.00}$ | $0.89_{0.83}^{0.93}$ | $0.78_{0.73}^{0.83}$ | $0.61_{0.57}^{0.67}$ | $0.53_{0.40}^{0.60}$ | $0.37_{0.33}^{0.40}$ | $0.21_{0.20}^{0.23}$ | $0.17_{0.17}^{0.17}$ | $0.26_{0.23}^{0.27}$ | $0.13_{0.10}^{0.17}$ |
| DeepSeek-R1 | $1.00_{1.00}^{1.00}$ | $0.99_{0.97}^{1.00}$ | $0.94_{0.93}^{0.97}$ | $0.81_{0.73}^{0.90}$ | $0.80_{0.73}^{0.83}$ | $0.61_{0.53}^{0.77}$ | $0.37_{0.20}^{0.50}$ | $0.31_{0.30}^{0.33}$ | $0.11_{0.07}^{0.17}$ | $0.13_{0.10}^{0.17}$ |
| o1-mini | $0.91_{0.87}^{0.97}$ | $0.59_{0.47}^{0.73}$ | $0.48_{0.43}^{0.53}$ | $0.20_{0.17}^{0.23}$ | $0.10_{0.07}^{0.13}$ | $0.03_{0.00}^{0.07}$ | $0.01_{0.00}^{0.03}$ | $0.00_{0.00}^{0.00}$ | $0.00_{0.00}^{0.00}$ | $0.00_{0.00}^{0.00}$ |
| o3-mini | $1.00_{1.00}^{1.00}$ | $1.00_{1.00}^{1.00}$ | $0.98_{0.97}^{1.00}$ | $0.89_{0.87}^{0.90}$ | $0.74_{0.70}^{0.77}$ | $0.31_{0.23}^{0.37}$ | $0.04_{0.03}^{0.07}$ | $0.01_{0.00}^{0.03}$ | $0.00_{0.00}^{0.00}$ | $0.00_{0.00}^{0.00}$ |

Table 13: QDE

| | 1 | 2 | 3 | 4 | 5 | 6 | 7 | 8 | 9 | 10 |
|---|---|---|---|---|---|---|---|---|---|---|
| QwQ-32B | $0.72^{1.00}_{0.30}$ | $0.56^{0.80}_{0.17}$ | $0.19^{0.27}_{0.07}$ | $0.16^{0.23}_{0.07}$ | $0.03^{0.03}_{0.03}$ | $0.00^{0.00}_{0.00}$ | $0.00^{0.00}_{0.00}$ | $0.00^{0.00}_{0.00}$ | $0.00^{0.00}_{0.00}$ | $0.00^{0.00}_{0.00}$ |
| DeepSeek-R1-32B | $0.84^{0.90}_{0.77}$ | $0.62^{0.70}_{0.50}$ | $0.11^{0.13}_{0.10}$ | $0.08^{0.10}_{0.07}$ | $0.00^{0.00}_{0.00}$ | $0.00^{0.00}_{0.00}$ | $0.00^{0.00}_{0.00}$ | $0.00^{0.00}_{0.00}$ | $0.00^{0.00}_{0.00}$ | $0.00^{0.00}_{0.00}$ |
| GPT-4o-mini | $0.49^{0.53}_{0.43}$ | $0.23^{0.27}_{0.17}$ | $0.03^{0.07}_{0.00}$ | $0.00^{0.00}_{0.00}$ | $0.00^{0.00}_{0.00}$ | $0.00^{0.00}_{0.00}$ | $0.00^{0.00}_{0.00}$ | $0.00^{0.00}_{0.00}$ | $0.00^{0.00}_{0.00}$ | $0.00^{0.00}_{0.00}$ |
| GPT-4o | $0.67^{0.70}_{0.60}$ | $0.43^{0.57}_{0.33}$ | $0.08^{0.10}_{0.07}$ | $0.03^{0.03}_{0.03}$ | $0.01^{0.03}_{0.00}$ | $0.00^{0.00}_{0.00}$ | $0.00^{0.00}_{0.00}$ | $0.00^{0.00}_{0.00}$ | $0.00^{0.00}_{0.00}$ | $0.00^{0.00}_{0.00}$ |
| Claude-3.7-Sonnet | $0.96^{1.00}_{0.87}$ | $0.97^{0.97}_{0.97}$ | $0.78^{0.80}_{0.77}$ | $0.59^{0.67}_{0.47}$ | $0.10^{0.13}_{0.07}$ | $0.00^{0.00}_{0.00}$ | $0.00^{0.00}_{0.00}$ | $0.00^{0.00}_{0.00}$ | $0.00^{0.00}_{0.00}$ | $0.00^{0.00}_{0.00}$ |
| DeepSeek-V3 | $0.97^{0.97}_{0.97}$ | $0.89^{0.93}_{0.83}$ | $0.38^{0.40}_{0.37}$ | $0.19^{0.30}_{0.10}$ | $0.04^{0.07}_{0.03}$ | $0.02^{0.03}_{0.00}$ | $0.00^{0.00}_{0.00}$ | $0.00^{0.00}_{0.00}$ | $0.00^{0.00}_{0.00}$ | $0.00^{0.00}_{0.00}$ |
| DeepSeek-V3-2503 | $1.00^{1.00}_{1.00}$ | $1.00^{1.00}_{1.00}$ | $0.68^{0.70}_{0.63}$ | $0.64^{0.73}_{0.57}$ | $0.30^{0.37}_{0.20}$ | $0.17^{0.20}_{0.13}$ | $0.08^{0.13}_{0.00}$ | $0.01^{0.03}_{0.00}$ | $0.08^{0.13}_{0.00}$ | $0.00^{0.00}_{0.00}$ |
| DeepSeek-R1 | $1.00^{1.00}_{1.00}$ | $1.00^{1.00}_{1.00}$ | $1.00^{1.00}_{1.00}$ | $0.97^{1.00}_{0.93}$ | $0.82^{0.90}_{0.77}$ | $0.68^{0.73}_{0.63}$ | $0.27^{0.33}_{0.20}$ | $0.17^{0.20}_{0.13}$ | $0.09^{0.13}_{0.07}$ | $0.03^{0.07}_{0.00}$ |
| o1-mini | $0.57^{0.70}_{0.50}$ | $0.59^{0.63}_{0.50}$ | $0.44^{0.63}_{0.33}$ | $0.46^{0.50}_{0.43}$ | $0.11^{0.17}_{0.07}$ | $0.03^{0.07}_{0.00}$ | $0.00^{0.00}_{0.00}$ | $0.00^{0.00}_{0.00}$ | $0.00^{0.00}_{0.00}$ | $0.00^{0.00}_{0.00}$ |
| o3-mini | $0.94^{0.97}_{0.90}$ | $0.99^{1.00}_{0.97}$ | $0.94^{0.97}_{0.93}$ | $0.96^{1.00}_{0.90}$ | $0.81^{0.87}_{0.77}$ | $0.66^{0.77}_{0.53}$ | $0.30^{0.43}_{0.20}$ | $0.27^{0.30}_{0.20}$ | $0.27^{0.30}_{0.23}$ | $0.13^{0.17}_{0.10}$ |

Table 14: 3DM

| | 1 | 2 | 3 | 4 | 5 | 6 | 7 | 8 | 9 | 10 |
|---|---|---|---|---|---|---|---|---|---|---|
| QwQ-32B | $1.00^{1.00}_{1.00}$ | $0.98^{1.00}_{0.97}$ | $0.93^{0.97}_{0.90}$ | $0.94^{0.97}_{0.93}$ | $0.33^{0.83}_{0.07}$ | $0.06^{0.10}_{0.00}$ | $0.00^{0.00}_{0.00}$ | $0.00^{0.00}_{0.00}$ | $0.00^{0.00}_{0.00}$ | $0.00^{0.00}_{0.00}$ |
| DeepSeek-R1-32B | $0.87^{1.00}_{0.77}$ | $0.42^{0.57}_{0.23}$ | $0.09^{0.13}_{0.03}$ | $0.00^{0.00}_{0.00}$ | $0.00^{0.00}_{0.00}$ | $0.01^{0.03}_{0.00}$ | $0.00^{0.00}_{0.00}$ | $0.00^{0.00}_{0.00}$ | $0.00^{0.00}_{0.00}$ | $0.00^{0.00}_{0.00}$ |
| GPT-4o-mini | $0.43^{0.57}_{0.27}$ | $0.09^{0.10}_{0.07}$ | $0.02^{0.03}_{0.00}$ | $0.00^{0.00}_{0.00}$ | $0.00^{0.00}_{0.00}$ | $0.00^{0.00}_{0.00}$ | $0.00^{0.00}_{0.00}$ | $0.00^{0.00}_{0.00}$ | $0.00^{0.00}_{0.00}$ | $0.00^{0.00}_{0.00}$ |
| GPT-4o | $0.64^{0.83}_{0.53}$ | $0.24^{0.37}_{0.17}$ | $0.13^{0.20}_{0.03}$ | $0.10^{0.13}_{0.07}$ | $0.02^{0.07}_{0.00}$ | $0.00^{0.00}_{0.00}$ | $0.00^{0.00}_{0.00}$ | $0.00^{0.00}_{0.00}$ | $0.00^{0.00}_{0.00}$ | $0.00^{0.00}_{0.00}$ |
| Claude-3.7-Sonnet | $0.96^{0.97}_{0.93}$ | $0.84^{0.90}_{0.80}$ | $0.76^{0.80}_{0.67}$ | $0.59^{0.70}_{0.43}$ | $0.21^{0.33}_{0.13}$ | $0.09^{0.10}_{0.07}$ | $0.07^{0.10}_{0.03}$ | $0.00^{0.00}_{0.00}$ | $0.00^{0.00}_{0.00}$ | $0.00^{0.00}_{0.00}$ |
| DeepSeek-V3 | $0.74^{0.83}_{0.57}$ | $0.32^{0.47}_{0.23}$ | $0.08^{0.13}_{0.00}$ | $0.03^{0.10}_{0.00}$ | $0.00^{0.00}_{0.00}$ | $0.00^{0.00}_{0.00}$ | $0.00^{0.00}_{0.00}$ | $0.00^{0.00}_{0.00}$ | $0.00^{0.00}_{0.00}$ | $0.00^{0.00}_{0.00}$ |
| DeepSeek-V3-2503 | $0.94^{0.97}_{0.93}$ | $0.76^{0.87}_{0.70}$ | $0.49^{0.60}_{0.43}$ | $0.31^{0.47}_{0.10}$ | $0.07^{0.10}_{0.03}$ | $0.03^{0.07}_{0.00}$ | $0.01^{0.03}_{0.00}$ | $0.00^{0.00}_{0.00}$ | $0.00^{0.00}_{0.00}$ | $0.00^{0.00}_{0.00}$ |
| DeepSeek-R1 | $1.00^{1.00}_{1.00}$ | $1.00^{1.00}_{1.00}$ | $0.98^{1.00}_{0.97}$ | $0.97^{1.00}_{0.93}$ | $0.93^{0.97}_{0.90}$ | $0.91^{0.97}_{0.83}$ | $0.91^{0.97}_{0.87}$ | $0.57^{0.67}_{0.50}$ | $0.27^{0.37}_{0.17}$ | $0.02^{0.03}_{0.00}$ |
| o1-mini | $0.87^{0.93}_{0.83}$ | $0.89^{0.90}_{0.87}$ | $0.81^{0.87}_{0.73}$ | $0.77^{0.83}_{0.73}$ | $0.38^{0.47}_{0.30}$ | $0.26^{0.27}_{0.23}$ | $0.11^{0.20}_{0.07}$ | $0.01^{0.03}_{0.00}$ | $0.00^{0.00}_{0.00}$ | $0.00^{0.00}_{0.00}$ |
| o3-mini | $0.63^{0.70}_{0.60}$ | $0.86^{0.93}_{0.80}$ | $0.72^{0.77}_{0.67}$ | $0.71^{0.80}_{0.57}$ | $0.57^{0.60}_{0.53}$ | $0.56^{0.70}_{0.43}$ | $0.38^{0.53}_{0.30}$ | $0.30^{0.37}_{0.23}$ | $0.23^{0.23}_{0.23}$ | $0.20^{0.23}_{0.17}$ |

Table 15: TSP

| | 1 | 2 | 3 | 4 | 5 | 6 | 7 | 8 | 9 | 10 |
|---|---|---|---|---|---|---|---|---|---|---|
| QwQ-32B | $0.61^{0.80}_{0.50}$ | $0.41^{0.53}_{0.30}$ | $0.42^{0.53}_{0.30}$ | $0.56^{0.60}_{0.50}$ | $0.26^{0.30}_{0.23}$ | $0.19^{0.27}_{0.13}$ | $0.02^{0.07}_{0.00}$ | $0.00^{0.00}_{0.00}$ | $0.00^{0.00}_{0.00}$ | $0.00^{0.00}_{0.00}$ |
| DeepSeek-R1-32B | $0.88^{0.90}_{0.87}$ | $0.62^{0.73}_{0.53}$ | $0.30^{0.40}_{0.23}$ | $0.13^{0.20}_{0.03}$ | $0.02^{0.03}_{0.00}$ | $0.01^{0.03}_{0.00}$ | $0.00^{0.00}_{0.00}$ | $0.00^{0.00}_{0.00}$ | $0.00^{0.00}_{0.00}$ | $0.00^{0.00}_{0.00}$ |
| GPT-4o-mini | $0.93^{0.96}_{0.90}$ | $0.34^{0.40}_{0.27}$ | $0.12^{0.20}_{0.07}$ | $0.07^{0.10}_{0.00}$ | $0.00^{0.00}_{0.00}$ | $0.00^{0.00}_{0.00}$ | $0.00^{0.00}_{0.00}$ | $0.00^{0.00}_{0.00}$ | $0.00^{0.00}_{0.00}$ | $0.00^{0.00}_{0.00}$ |
| GPT-4o | $0.97^{0.97}_{0.97}$ | $0.76^{0.80}_{0.73}$ | $0.59^{0.67}_{0.50}$ | $0.40^{0.47}_{0.33}$ | $0.22^{0.33}_{0.10}$ | $0.16^{0.23}_{0.03}$ | $0.08^{0.10}_{0.07}$ | $0.02^{0.07}_{0.00}$ | $0.00^{0.00}_{0.00}$ | $0.00^{0.00}_{0.00}$ |
| Claude-3.7-Sonnet | $1.00^{1.00}_{1.00}$ | $0.98^{1.00}_{0.97}$ | $0.90^{0.93}_{0.83}$ | $0.83^{0.90}_{0.77}$ | $0.86^{0.90}_{0.80}$ | $0.80^{0.83}_{0.73}$ | $0.54^{0.70}_{0.47}$ | $0.51^{0.53}_{0.50}$ | $0.08^{0.10}_{0.03}$ | $0.06^{0.10}_{0.00}$ |
| DeepSeek-V3 | $0.98^{1.00}_{0.93}$ | $0.90^{0.90}_{0.90}$ | $0.74^{0.83}_{0.60}$ | $0.62^{0.77}_{0.50}$ | $0.49^{0.67}_{0.33}$ | $0.49^{0.73}_{0.33}$ | $0.17^{0.23}_{0.13}$ | $0.07^{0.13}_{0.00}$ | $0.02^{0.03}_{0.00}$ | $0.00^{0.00}_{0.00}$ |
| DeepSeek-V3-2503 | $1.00^{1.00}_{1.00}$ | $0.94^{0.97}_{0.90}$ | $0.96^{1.00}_{0.87}$ | $0.83^{0.87}_{0.80}$ | $0.70^{0.77}_{0.67}$ | $0.66^{0.70}_{0.57}$ | $0.39^{0.47}_{0.27}$ | $0.10^{0.10}_{0.10}$ | $0.01^{0.03}_{0.00}$ | $0.01^{0.03}_{0.00}$ |
| DeepSeek-R1 | $1.00^{1.00}_{1.00}$ | $0.99^{1.00}_{0.97}$ | $0.97^{0.97}_{0.97}$ | $0.99^{1.00}_{0.97}$ | $0.87^{0.87}_{0.87}$ | $0.78^{0.80}_{0.77}$ | $0.62^{0.67}_{0.57}$ | $0.24^{0.30}_{0.20}$ | $0.03^{0.10}_{0.00}$ | $0.00^{0.00}_{0.00}$ |
| o1-mini | $0.84^{0.90}_{0.73}$ | $0.89^{0.93}_{0.87}$ | $0.67^{0.77}_{0.60}$ | $0.57^{0.63}_{0.43}$ | $0.34^{0.47}_{0.23}$ | $0.37^{0.43}_{0.23}$ | $0.18^{0.30}_{0.07}$ | $0.01^{0.03}_{0.00}$ | $0.00^{0.00}_{0.00}$ | $0.00^{0.00}_{0.00}$ |
| o3-mini | $0.79^{0.87}_{0.73}$ | $0.62^{0.67}_{0.53}$ | $0.53^{0.63}_{0.47}$ | $0.28^{0.33}_{0.20}$ | $0.31^{0.37}_{0.23}$ | $0.30^{0.47}_{0.17}$ | $0.30^{0.37}_{0.20}$ | $0.19^{0.20}_{0.17}$ | $0.12^{0.17}_{0.07}$ | $0.07^{0.13}_{0.00}$ |

Table 16: Hamiltonian Cycle

| | 1 | 2 | 3 | 4 | 5 | 6 | 7 | 8 | 9 | 10 |
|---|---|---|---|---|---|---|---|---|---|---|
| QwQ-32B | $0.94^{1.00}_{0.90}$ | $0.87^{0.93}_{0.83}$ | $0.80^{0.93}_{0.70}$ | $0.62^{0.67}_{0.57}$ | $0.33^{0.40}_{0.23}$ | $0.16^{0.20}_{0.10}$ | $0.03^{0.10}_{0.00}$ | $0.00^{0.00}_{0.00}$ | $0.00^{0.00}_{0.00}$ | $0.00^{0.00}_{0.00}$ |
| DeepSeek-R1-32B | $0.69^{0.73}_{0.67}$ | $0.36^{0.40}_{0.27}$ | $0.24^{0.40}_{0.13}$ | $0.09^{0.13}_{0.03}$ | $0.00^{0.00}_{0.00}$ | $0.01^{0.03}_{0.00}$ | $0.02^{0.03}_{0.00}$ | $0.00^{0.00}_{0.00}$ | $0.00^{0.00}_{0.00}$ | $0.00^{0.00}_{0.00}$ |
| GPT-4o-mini | $0.70^{0.73}_{0.67}$ | $0.26^{0.40}_{0.13}$ | $0.09^{0.10}_{0.07}$ | $0.08^{0.13}_{0.03}$ | $0.01^{0.03}_{0.00}$ | $0.00^{0.00}_{0.00}$ | $0.01^{0.03}_{0.00}$ | $0.00^{0.00}_{0.00}$ | $0.00^{0.00}_{0.00}$ | $0.00^{0.00}_{0.00}$ |
| GPT-4o | $0.73^{0.73}_{0.73}$ | $0.39^{0.43}_{0.33}$ | $0.22^{0.27}_{0.17}$ | $0.12^{0.20}_{0.07}$ | $0.09^{0.13}_{0.00}$ | $0.01^{0.03}_{0.00}$ | $0.06^{0.10}_{0.00}$ | $0.00^{0.00}_{0.00}$ | $0.00^{0.00}_{0.00}$ | $0.00^{0.00}_{0.00}$ |
| Claude-3.7-Sonnet | $0.99^{1.00}_{0.97}$ | $0.80^{0.90}_{0.70}$ | $0.74^{0.83}_{0.67}$ | $0.64^{0.77}_{0.53}$ | $0.32^{0.50}_{0.17}$ | $0.23^{0.27}_{0.20}$ | $0.27^{0.33}_{0.20}$ | $0.16^{0.27}_{0.10}$ | $0.10^{0.10}_{0.10}$ | $0.02^{0.07}_{0.00}$ |
| DeepSeek-V3 | $0.83^{0.90}_{0.77}$ | $0.44^{0.53}_{0.30}$ | $0.14^{0.20}_{0.10}$ | $0.16^{0.17}_{0.13}$ | $0.09^{0.17}_{0.00}$ | $0.06^{0.07}_{0.03}$ | $0.06^{0.10}_{0.00}$ | $0.06^{0.10}_{0.00}$ | $0.01^{0.03}_{0.00}$ | $0.01^{0.03}_{0.00}$ |
| DeepSeek-V3-2503 | $0.99^{1.00}_{0.97}$ | $0.82^{0.90}_{0.77}$ | $0.51^{0.53}_{0.50}$ | $0.38^{0.53}_{0.27}$ | $0.16^{0.27}_{0.07}$ | $0.14^{0.17}_{0.13}$ | $0.09^{0.10}_{0.07}$ | $0.10^{0.17}_{0.07}$ | $0.06^{0.07}_{0.03}$ | $0.03^{0.07}_{0.00}$ |
| DeepSeek-R1 | $1.00^{1.00}_{1.00}$ | $1.00^{1.00}_{1.00}$ | $0.97^{1.00}_{0.93}$ | $0.91^{0.97}_{0.83}$ | $0.76^{0.93}_{0.63}$ | $0.64^{0.73}_{0.57}$ | $0.49^{0.57}_{0.43}$ | $0.36^{0.40}_{0.27}$ | $0.17^{0.23}_{0.13}$ | $0.04^{0.10}_{0.00}$ |
| o1-mini | $0.72^{0.80}_{0.57}$ | $0.71^{0.77}_{0.67}$ | $0.54^{0.60}_{0.50}$ | $0.40^{0.47}_{0.33}$ | $0.19^{0.23}_{0.17}$ | $0.23^{0.30}_{0.13}$ | $0.12^{0.17}_{0.10}$ | $0.08^{0.10}_{0.03}$ | $0.04^{0.07}_{0.00}$ | $0.00^{0.00}_{0.00}$ |
| o3-mini | $0.82^{0.83}_{0.80}$ | $0.84^{0.90}_{0.77}$ | $0.71^{0.77}_{0.63}$ | $0.71^{0.83}_{0.60}$ | $0.63^{0.73}_{0.57}$ | $0.59^{0.67}_{0.50}$ | $0.44^{0.50}_{0.37}$ | $0.32^{0.43}_{0.27}$ | $0.20^{0.33}_{0.10}$ | $0.22^{0.23}_{0.20}$ |

Table 17: Bin Packing

| | 1 | 2 | 3 | 4 | 5 | 6 | 7 | 8 | 9 | 10 |
|---|---|---|---|---|---|---|---|---|---|---|
| QwQ-32B | $0.88^{0.93}_{0.80}$ | $0.83^{0.87}_{0.77}$ | $0.46^{0.57}_{0.40}$ | $0.08^{0.20}_{0.00}$ | $0.00^{0.00}_{0.00}$ | $0.00^{0.00}_{0.00}$ | $0.00^{0.00}_{0.00}$ | $0.00^{0.00}_{0.00}$ | $0.00^{0.00}_{0.00}$ | $0.00^{0.00}_{0.00}$ |
| DeepSeek-R1-32B | $0.26^{0.33}_{0.17}$ | $0.03^{0.07}_{0.00}$ | $0.01^{0.03}_{0.00}$ | $0.03^{0.07}_{0.00}$ | $0.00^{0.00}_{0.00}$ | $0.00^{0.00}_{0.00}$ | $0.00^{0.00}_{0.00}$ | $0.00^{0.00}_{0.00}$ | $0.00^{0.00}_{0.00}$ | $0.00^{0.00}_{0.00}$ |
| GPT-4o-mini | $0.30^{0.33}_{0.23}$ | $0.03^{0.03}_{0.03}$ | $0.04^{0.10}_{0.00}$ | $0.03^{0.10}_{0.00}$ | $0.00^{0.00}_{0.00}$ | $0.00^{0.00}_{0.00}$ | $0.00^{0.00}_{0.00}$ | $0.00^{0.00}_{0.00}$ | $0.00^{0.00}_{0.00}$ | $0.00^{0.00}_{0.00}$ |
| GPT-4o | $0.83^{0.90}_{0.73}$ | $0.44^{0.50}_{0.40}$ | $0.34^{0.37}_{0.33}$ | $0.18^{0.20}_{0.13}$ | $0.04^{0.10}_{0.00}$ | $0.02^{0.03}_{0.00}$ | $0.00^{0.00}_{0.00}$ | $0.00^{0.00}_{0.00}$ | $0.00^{0.00}_{0.00}$ | $0.00^{0.00}_{0.00}$ |
| Claude-3.7-Sonnet | $0.98^{1.00}_{0.97}$ | $0.89^{0.93}_{0.83}$ | $0.58^{0.70}_{0.43}$ | $0.39^{0.40}_{0.37}$ | $0.07^{0.17}_{0.00}$ | $0.01^{0.03}_{0.00}$ | $0.01^{0.03}_{0.00}$ | $0.03^{0.07}_{0.00}$ | $0.01^{0.03}_{0.00}$ | $0.00^{0.00}_{0.00}$ |
| DeepSeek-V3 | $0.66^{0.73}_{0.60}$ | $0.46^{0.50}_{0.40}$ | $0.44^{0.50}_{0.37}$ | $0.37^{0.40}_{0.33}$ | $0.06^{0.13}_{0.00}$ | $0.04^{0.07}_{0.00}$ | $0.02^{0.03}_{0.00}$ | $0.00^{0.00}_{0.00}$ | $0.01^{0.03}_{0.00}$ | $0.00^{0.00}_{0.00}$ |
| DeepSeek-V3-2503 | $1.00^{1.00}_{1.00}$ | $0.87^{0.93}_{0.80}$ | $0.74^{0.83}_{0.67}$ | $0.62^{0.67}_{0.57}$ | $0.18^{0.27}_{0.10}$ | $0.18^{0.23}_{0.13}$ | $0.09^{0.13}_{0.03}$ | $0.02^{0.07}_{0.00}$ | $0.02^{0.03}_{0.00}$ | $0.00^{0.00}_{0.00}$ |
| DeepSeek-R1 | $1.00^{1.00}_{1.00}$ | $1.00^{1.00}_{1.00}$ | $1.00^{1.00}_{1.00}$ | $0.98^{1.00}_{0.97}$ | $0.80^{0.90}_{0.73}$ | $0.64^{0.77}_{0.57}$ | $0.49^{0.53}_{0.47}$ | $0.29^{0.43}_{0.20}$ | $0.06^{0.10}_{0.03}$ | $0.03^{0.07}_{0.00}$ |
| o1-mini | $0.67^{0.80}_{0.43}$ | $0.58^{0.63}_{0.50}$ | $0.52^{0.57}_{0.47}$ | $0.33^{0.40}_{0.27}$ | $0.31^{0.50}_{0.20}$ | $0.19^{0.23}_{0.13}$ | $0.07^{0.10}_{0.03}$ | $0.00^{0.00}_{0.00}$ | $0.02^{0.07}_{0.00}$ | $0.01^{0.03}_{0.00}$ |
| o3-mini | $0.72^{0.83}_{0.60}$ | $0.67^{0.80}_{0.57}$ | $0.62^{0.67}_{0.57}$ | $0.48^{0.57}_{0.33}$ | $0.41^{0.47}_{0.37}$ | $0.29^{0.43}_{0.17}$ | $0.24^{0.47}_{0.13}$ | $0.17^{0.27}_{0.10}$ | $0.42^{0.47}_{0.33}$ | $0.28^{0.33}_{0.20}$ |

Table 18: 3-COL

| | 1 | 2 | 3 | 4 | 5 | 6 | 7 | 8 | 9 | 10 |
|---|---|---|---|---|---|---|---|---|---|---|
| QwQ-32B | $0.96^{1.00}_{0.90}$ | $0.91^{0.93}_{0.87}$ | $0.78^{0.87}_{0.73}$ | $0.56^{0.67}_{0.50}$ | $0.34^{0.43}_{0.27}$ | $0.10^{0.13}_{0.07}$ | $0.01^{0.03}_{0.00}$ | $0.01^{0.03}_{0.00}$ | $0.00^{0.00}_{0.00}$ | $0.00^{0.00}_{0.00}$ |
| DeepSeek-R1-32B | $0.49^{0.57}_{0.43}$ | $0.51^{0.57}_{0.43}$ | $0.03^{0.07}_{0.00}$ | $0.01^{0.03}_{0.00}$ | $0.00^{0.00}_{0.00}$ | $0.00^{0.00}_{0.00}$ | $0.00^{0.00}_{0.00}$ | $0.00^{0.00}_{0.00}$ | $0.00^{0.00}_{0.00}$ | $0.00^{0.00}_{0.00}$ |
| GPT-4o-mini | $0.40^{0.50}_{0.30}$ | $0.17^{0.20}_{0.13}$ | $0.00^{0.00}_{0.00}$ | $0.00^{0.00}_{0.00}$ | $0.00^{0.00}_{0.00}$ | $0.00^{0.00}_{0.00}$ | $0.00^{0.00}_{0.00}$ | $0.00^{0.00}_{0.00}$ | $0.00^{0.00}_{0.00}$ | $0.00^{0.00}_{0.00}$ |
| GPT-4o | $0.60^{0.63}_{0.57}$ | $0.39^{0.53}_{0.30}$ | $0.03^{0.10}_{0.00}$ | $0.01^{0.03}_{0.00}$ | $0.01^{0.03}_{0.00}$ | $0.00^{0.00}_{0.00}$ | $0.00^{0.00}_{0.00}$ | $0.00^{0.00}_{0.00}$ | $0.00^{0.00}_{0.00}$ | $0.00^{0.00}_{0.00}$ |
| Claude-3.7-Sonnet | $0.76^{0.87}_{0.67}$ | $0.70^{0.73}_{0.67}$ | $0.22^{0.33}_{0.07}$ | $0.17^{0.20}_{0.13}$ | $0.09^{0.10}_{0.07}$ | $0.04^{0.07}_{0.00}$ | $0.01^{0.03}_{0.00}$ | $0.00^{0.00}_{0.00}$ | $0.00^{0.00}_{0.00}$ | $0.00^{0.00}_{0.00}$ |
| DeepSeek-V3 | $0.67^{0.70}_{0.63}$ | $0.60^{0.63}_{0.53}$ | $0.13^{0.20}_{0.07}$ | $0.12^{0.17}_{0.10}$ | $0.03^{0.07}_{0.00}$ | $0.02^{0.07}_{0.00}$ | $0.02^{0.07}_{0.00}$ | $0.00^{0.00}_{0.00}$ | $0.00^{0.00}_{0.00}$ | $0.00^{0.00}_{0.00}$ |
| DeepSeek-V3-2503 | $0.80^{0.87}_{0.73}$ | $0.90^{0.93}_{0.87}$ | $0.48^{0.63}_{0.40}$ | $0.64^{0.73}_{0.57}$ | $0.32^{0.37}_{0.30}$ | $0.16^{0.20}_{0.07}$ | $0.16^{0.23}_{0.07}$ | $0.09^{0.20}_{0.00}$ | $0.02^{0.03}_{0.00}$ | $0.00^{0.00}_{0.00}$ |
| DeepSeek-R1 | $0.99^{1.00}_{0.97}$ | $1.00^{1.00}_{1.00}$ | $0.97^{1.00}_{0.93}$ | $0.97^{0.97}_{0.97}$ | $0.88^{0.93}_{0.80}$ | $0.72^{0.77}_{0.67}$ | $0.72^{0.80}_{0.67}$ | $0.51^{0.67}_{0.47}$ | $0.22^{0.27}_{0.17}$ | $0.04^{0.07}_{0.03}$ |
| o1-mini | $0.61^{0.70}_{0.50}$ | $0.76^{0.87}_{0.70}$ | $0.57^{0.70}_{0.43}$ | $0.62^{0.67}_{0.60}$ | $0.37^{0.43}_{0.33}$ | $0.27^{0.30}_{0.23}$ | $0.34^{0.40}_{0.27}$ | $0.17^{0.23}_{0.07}$ | $0.03^{0.07}_{0.00}$ | $0.02^{0.30}_{0.00}$ |
| o3-mini | $0.98^{1.00}_{0.97}$ | $0.91^{0.93}_{0.87}$ | $0.96^{1.00}_{0.90}$ | $0.84^{0.87}_{0.83}$ | $0.78^{0.87}_{0.70}$ | $0.72^{0.80}_{0.67}$ | $0.71^{0.80}_{0.60}$ | $0.61^{0.80}_{0.47}$ | $0.51^{0.53}_{0.47}$ | $0.29^{0.30}_{0.27}$ |

Table 19: Min Sum Square

| | 1 | 2 | 3 | 4 | 5 | 6 | 7 | 8 | 9 | 10 |
|---|---|---|---|---|---|---|---|---|---|---|
| QwQ-32B | $0.77^{0.80}_{0.70}$ | $0.00^{0.00}_{0.00}$ | $0.00^{0.00}_{0.00}$ | $0.00^{0.00}_{0.00}$ | $0.00^{0.00}_{0.00}$ | $0.00^{0.00}_{0.00}$ | $0.00^{0.00}_{0.00}$ | $0.01^{0.03}_{0.00}$ | $0.00^{0.00}_{0.00}$ | $0.00^{0.00}_{0.00}$ |
| DeepSeek-R1-32B | $0.23^{0.43}_{0.10}$ | $0.00^{0.00}_{0.00}$ | $0.04^{0.07}_{0.00}$ | $0.07^{0.10}_{0.03}$ | $0.06^{0.10}_{0.03}$ | $0.04^{0.13}_{0.00}$ | $0.02^{0.07}_{0.00}$ | $0.06^{0.10}_{0.03}$ | $0.01^{0.03}_{0.00}$ | $0.06^{0.07}_{0.03}$ |
| GPT-4o-mini | $0.74^{0.80}_{0.67}$ | $0.62^{0.77}_{0.50}$ | $0.03^{0.07}_{0.00}$ | $0.07^{0.10}_{0.03}$ | $0.08^{0.10}_{0.00}$ | $0.03^{0.07}_{0.00}$ | $0.03^{0.07}_{0.00}$ | $0.02^{0.07}_{0.00}$ | $0.02^{0.03}_{0.00}$ | $0.04^{0.13}_{0.00}$ |
| GPT-4o | $0.94^{0.97}_{0.90}$ | $0.82^{0.87}_{0.80}$ | $0.46^{0.53}_{0.37}$ | $0.56^{0.60}_{0.53}$ | $0.44^{0.53}_{0.40}$ | $0.48^{0.53}_{0.43}$ | $0.04^{0.07}_{0.03}$ | $0.01^{0.03}_{0.00}$ | $0.01^{0.03}_{0.00}$ | $0.01^{0.03}_{0.00}$ |
| Claude-3.7-Sonnet | $0.98^{1.00}_{0.97}$ | $0.84^{0.93}_{0.80}$ | $0.83^{0.90}_{0.80}$ | $0.73^{0.80}_{0.70}$ | $0.79^{0.87}_{0.70}$ | $0.64^{0.70}_{0.60}$ | $0.59^{0.63}_{0.50}$ | $0.67^{0.73}_{0.57}$ | $0.62^{0.67}_{0.53}$ | $0.14^{0.20}_{0.07}$ |
| DeepSeek-V3 | $0.87^{0.93}_{0.80}$ | $0.90^{0.93}_{0.83}$ | $0.84^{0.90}_{0.80}$ | $0.58^{0.63}_{0.53}$ | $0.58^{0.63}_{0.53}$ | $0.48^{0.57}_{0.43}$ | $0.07^{0.10}_{0.03}$ | $0.17^{0.23}_{0.07}$ | $0.07^{0.17}_{0.00}$ | $0.02^{0.03}_{0.00}$ |
| DeepSeek-V3-2503 | $1.00^{1.00}_{1.00}$ | $0.48^{0.57}_{0.40}$ | $0.71^{0.77}_{0.67}$ | $0.59^{0.63}_{0.50}$ | $0.62^{0.67}_{0.57}$ | $0.61^{0.70}_{0.50}$ | $0.22^{0.23}_{0.20}$ | $0.29^{0.37}_{0.23}$ | $0.02^{0.03}_{0.00}$ | $0.00^{0.00}_{0.00}$ |
| DeepSeek-R1 | $1.00^{1.00}_{1.00}$ | $0.88^{0.90}_{0.83}$ | $0.64^{0.73}_{0.57}$ | $0.46^{0.53}_{0.37}$ | $0.47^{0.53}_{0.33}$ | $0.39^{0.43}_{0.30}$ | $0.13^{0.17}_{0.10}$ | $0.13^{0.17}_{0.10}$ | $0.07^{0.13}_{0.00}$ | $0.02^{0.03}_{0.00}$ |
| o1-mini | $0.62^{0.67}_{0.57}$ | $0.70^{0.80}_{0.63}$ | $0.27^{0.30}_{0.23}$ | $0.18^{0.27}_{0.10}$ | $0.14^{0.23}_{0.10}$ | $0.10^{0.13}_{0.07}$ | $0.03^{0.10}_{0.00}$ | $0.06^{0.10}_{0.03}$ | $0.02^{0.07}_{0.00}$ | $0.01^{0.03}_{0.00}$ |
| o3-mini | $0.69^{0.80}_{0.60}$ | $0.38^{0.47}_{0.23}$ | $0.38^{0.47}_{0.33}$ | $0.39^{0.50}_{0.23}$ | $0.52^{0.60}_{0.47}$ | $0.30^{0.37}_{0.23}$ | $0.44^{0.50}_{0.33}$ | $0.24^{0.27}_{0.20}$ | $0.03^{0.07}_{0.00}$ | $0.18^{0.23}_{0.13}$ |

Table 20: Bandwidth

| | 1 | 2 | 3 | 4 | 5 | 6 | 7 | 8 | 9 | 10 |
|---|---|---|---|---|---|---|---|---|---|---|
| QwQ-32B | $0.96^{1.00}_{0.90}$ | $0.91^{0.93}_{0.90}$ | $0.90^{1.00}_{0.83}$ | $0.84^{0.93}_{0.70}$ | $0.76^{0.87}_{0.60}$ | $0.66^{0.77}_{0.60}$ | $0.63^{0.67}_{0.60}$ | $0.30^{0.40}_{0.20}$ | $0.20^{0.30}_{0.13}$ | $0.06^{0.07}_{0.03}$ |
| DeepSeek-R1-32B | $0.93^{1.00}_{0.83}$ | $0.83^{0.87}_{0.80}$ | $0.87^{0.93}_{0.80}$ | $0.67^{0.83}_{0.53}$ | $0.54^{0.70}_{0.47}$ | $0.49^{0.57}_{0.43}$ | $0.38^{0.40}_{0.37}$ | $0.11^{0.13}_{0.07}$ | $0.14^{0.17}_{0.10}$ | $0.03^{0.07}_{0.00}$ |
| GPT-4o-mini | $1.00^{1.00}_{1.00}$ | $0.94^{1.00}_{0.90}$ | $0.94^{0.97}_{0.93}$ | $0.84^{0.87}_{0.83}$ | $0.78^{0.80}_{0.77}$ | $0.47^{0.57}_{0.40}$ | $0.46^{0.47}_{0.43}$ | $0.20^{0.23}_{0.17}$ | $0.14^{0.20}_{0.10}$ | $0.03^{0.07}_{0.00}$ |
| GPT-4o | $1.00^{1.00}_{1.00}$ | $0.96^{1.00}_{0.90}$ | $0.97^{1.00}_{0.93}$ | $0.94^{1.00}_{0.87}$ | $0.78^{0.87}_{0.67}$ | $0.62^{0.67}_{0.57}$ | $0.60^{0.67}_{0.53}$ | $0.22^{0.30}_{0.17}$ | $0.10^{0.13}_{0.07}$ | $0.02^{0.03}_{0.00}$ |
| Claude-3.7-Sonnet | $1.00^{1.00}_{1.00}$ | $0.96^{1.00}_{0.90}$ | $0.96^{1.00}_{0.90}$ | $0.87^{0.90}_{0.83}$ | $0.78^{0.87}_{0.67}$ | $0.66^{0.73}_{0.60}$ | $0.62^{0.67}_{0.57}$ | $0.28^{0.33}_{0.17}$ | $0.11^{0.13}_{0.07}$ | $0.02^{0.03}_{0.00}$ |
| DeepSeek-V3 | $1.00^{1.00}_{0.97}$ | $0.98^{1.00}_{0.97}$ | $0.99^{1.00}_{0.97}$ | $0.93^{0.97}_{0.90}$ | $0.74^{0.90}_{0.63}$ | $0.63^{0.77}_{0.50}$ | $0.56^{0.63}_{0.50}$ | $0.34^{0.40}_{0.30}$ | $0.23^{0.33}_{0.17}$ | $0.03^{0.07}_{0.00}$ |
| DeepSeek-V3-2503 | $1.00^{1.00}_{1.00}$ | $0.91^{0.93}_{0.90}$ | $0.89^{0.90}_{0.87}$ | $0.62^{0.67}_{0.57}$ | $0.58^{0.63}_{0.53}$ | $0.57^{0.60}_{0.53}$ | $0.43^{0.50}_{0.33}$ | $0.33^{0.40}_{0.27}$ | $0.17^{0.20}_{0.13}$ | $0.04^{0.07}_{0.03}$ |
| DeepSeek-R1 | $1.00^{1.00}_{1.00}$ | $0.90^{0.93}_{0.83}$ | $0.93^{1.00}_{0.90}$ | $0.88^{0.93}_{0.80}$ | $0.83^{0.90}_{0.80}$ | $0.68^{0.77}_{0.57}$ | $0.59^{0.67}_{0.47}$ | $0.34^{0.43}_{0.30}$ | $0.24^{0.30}_{0.20}$ | $0.07^{0.10}_{0.03}$ |
| o1-mini | $0.74^{0.80}_{0.70}$ | $0.74^{0.83}_{0.60}$ | $0.84^{0.87}_{0.83}$ | $0.82^{0.87}_{0.77}$ | $0.82^{0.87}_{0.77}$ | $0.68^{0.70}_{0.67}$ | $0.59^{0.63}_{0.53}$ | $0.33^{0.37}_{0.30}$ | $0.24^{0.33}_{0.20}$ | $0.06^{0.07}_{0.03}$ |
| o3-mini | $0.80^{0.83}_{0.77}$ | $0.88^{0.93}_{0.83}$ | $0.82^{0.93}_{0.77}$ | $0.90^{0.93}_{0.87}$ | $0.72^{0.80}_{0.60}$ | $0.58^{0.70}_{0.43}$ | $0.52^{0.53}_{0.50}$ | $0.20^{0.23}_{0.17}$ | $0.17^{0.20}_{0.13}$ | $0.08^{0.10}_{0.07}$ |

Table 21: Max Leaf Span Tree

| | 1 | 2 | 3 | 4 | 5 | 6 | 7 | 8 | 9 | 10 |
|---|---|---|---|---|---|---|---|---|---|---|
| QwQ-32B | $0.73^{0.83}_{0.57}$ | $0.93^{0.97}_{0.87}$ | $0.28^{0.40}_{0.20}$ | $0.06^{0.07}_{0.03}$ | $0.00^{0.00}_{0.00}$ | $0.00^{0.00}_{0.00}$ | $0.00^{0.00}_{0.00}$ | $0.00^{0.00}_{0.00}$ | $0.00^{0.00}_{0.00}$ | $0.00^{0.00}_{0.00}$ |
| DeepSeek-R1-32B | $0.20^{0.30}_{0.03}$ | $0.24^{0.37}_{0.13}$ | $0.18^{0.27}_{0.13}$ | $0.00^{0.00}_{0.00}$ | $0.00^{0.00}_{0.00}$ | $0.00^{0.00}_{0.00}$ | $0.00^{0.00}_{0.00}$ | $0.00^{0.00}_{0.00}$ | $0.00^{0.00}_{0.00}$ | $0.00^{0.00}_{0.00}$ |
| GPT-4o-mini | $0.26^{0.33}_{0.17}$ | $0.19^{0.40}_{0.07}$ | $0.01^{0.03}_{0.00}$ | $0.00^{0.00}_{0.00}$ | $0.00^{0.00}_{0.00}$ | $0.00^{0.00}_{0.00}$ | $0.00^{0.00}_{0.00}$ | $0.00^{0.00}_{0.00}$ | $0.00^{0.00}_{0.00}$ | $0.00^{0.00}_{0.00}$ |
| GPT-4o | $0.49^{0.57}_{0.40}$ | $0.53^{0.60}_{0.47}$ | $0.29^{0.37}_{0.23}$ | $0.24^{0.30}_{0.20}$ | $0.08^{0.10}_{0.07}$ | $0.00^{0.00}_{0.00}$ | $0.00^{0.00}_{0.00}$ | $0.00^{0.00}_{0.00}$ | $0.00^{0.00}_{0.00}$ | $0.00^{0.00}_{0.00}$ |
| Claude-3.7-Sonnet | $1.00^{1.00}_{1.00}$ | $0.99^{1.00}_{0.97}$ | $0.96^{0.97}_{0.93}$ | $0.82^{0.93}_{0.70}$ | $0.71^{0.83}_{0.57}$ | $0.59^{0.63}_{0.57}$ | $0.12^{0.20}_{0.07}$ | $0.00^{0.00}_{0.00}$ | $0.00^{0.00}_{0.00}$ | $0.00^{0.00}_{0.00}$ |
| DeepSeek-V3 | $0.79^{0.83}_{0.77}$ | $0.88^{0.93}_{0.80}$ | $0.89^{0.93}_{0.80}$ | $0.69^{0.80}_{0.57}$ | $0.56^{0.60}_{0.50}$ | $0.26^{0.33}_{0.20}$ | $0.27^{0.43}_{0.13}$ | $0.09^{0.17}_{0.00}$ | $0.02^{0.03}_{0.00}$ | $0.01^{0.03}_{0.00}$ |
| DeepSeek-V3-2503 | $0.90^{0.97}_{0.80}$ | $0.86^{0.90}_{0.83}$ | $0.76^{0.80}_{0.67}$ | $0.39^{0.43}_{0.33}$ | $0.18^{0.30}_{0.10}$ | $0.22^{0.27}_{0.13}$ | $0.28^{0.33}_{0.23}$ | $0.17^{0.20}_{0.00}$ | $0.07^{0.13}_{0.00}$ | $0.02^{0.03}_{0.00}$ |
| DeepSeek-R1 | $0.97^{0.97}_{0.97}$ | $0.99^{1.00}_{0.97}$ | $0.88^{0.90}_{0.87}$ | $0.63^{0.77}_{0.53}$ | $0.39^{0.43}_{0.37}$ | $0.18^{0.23}_{0.13}$ | $0.21^{0.23}_{0.20}$ | $0.01^{0.03}_{0.00}$ | $0.01^{0.03}_{0.00}$ | $0.00^{0.00}_{0.00}$ |
| o1-mini | $0.70^{0.73}_{0.67}$ | $0.53^{0.53}_{0.53}$ | $0.57^{0.57}_{0.57}$ | $0.17^{0.20}_{0.10}$ | $0.02^{0.03}_{0.00}$ | $0.02^{0.03}_{0.00}$ | $0.00^{0.00}_{0.00}$ | $0.00^{0.00}_{0.00}$ | $0.01^{0.03}_{0.00}$ | $0.00^{0.00}_{0.00}$ |
| o3-mini | $0.77^{0.83}_{0.70}$ | $0.68^{0.73}_{0.63}$ | $0.66^{0.67}_{0.63}$ | $0.66^{0.70}_{0.60}$ | $0.42^{0.50}_{0.30}$ | $0.26^{0.27}_{0.23}$ | $0.19^{0.23}_{0.17}$ | $0.16^{0.33}_{0.07}$ | $0.11^{0.17}_{0.03}$ | $0.09^{0.17}_{0.03}$ |

# H   Performance over Problems

In this section, we present the performance of LLMs on each problem across different levels.

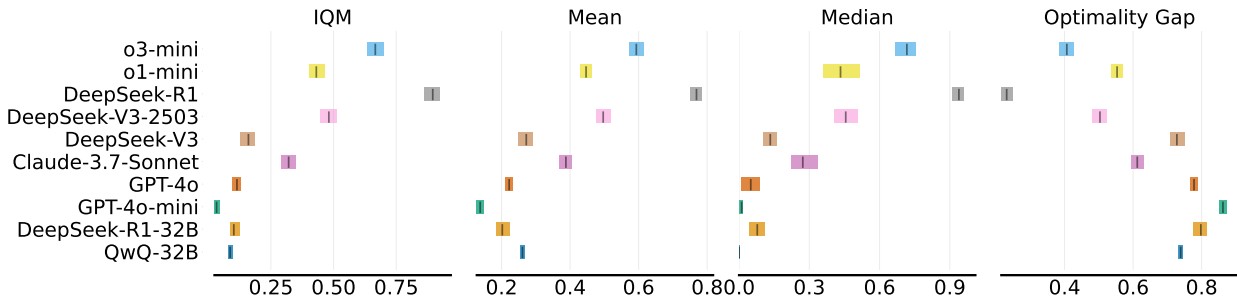

Figure 15: 3SAT

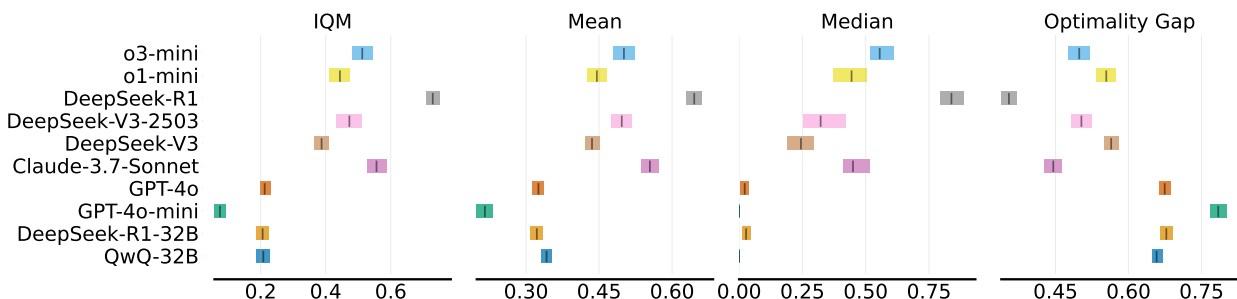

Figure 16: Vertex Cover

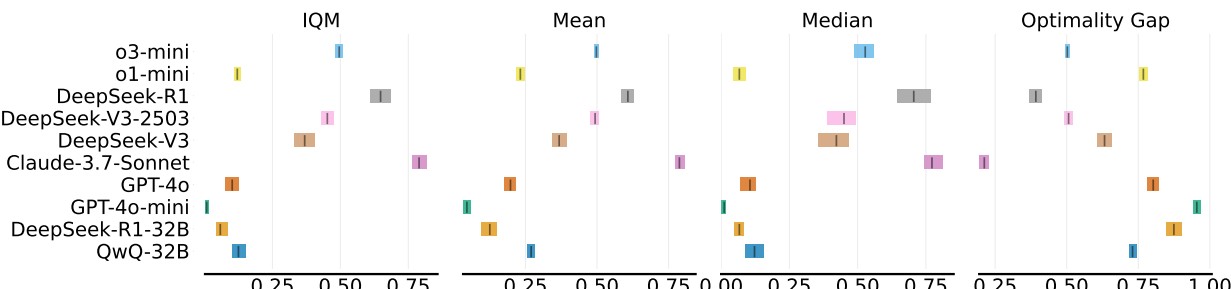

Figure 17: Superstring

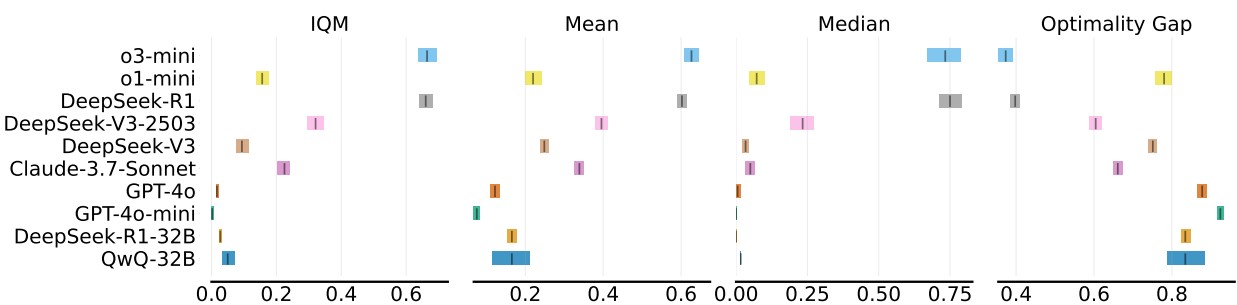

Figure 18: QDE

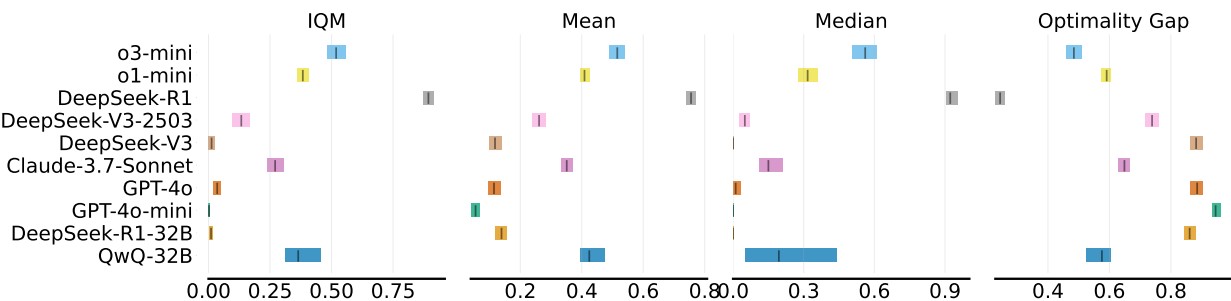

Figure 19: 3DM

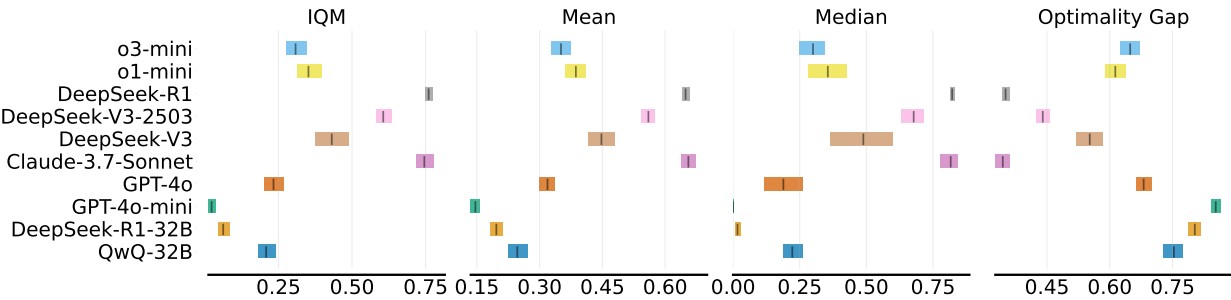

Figure 20: TSP

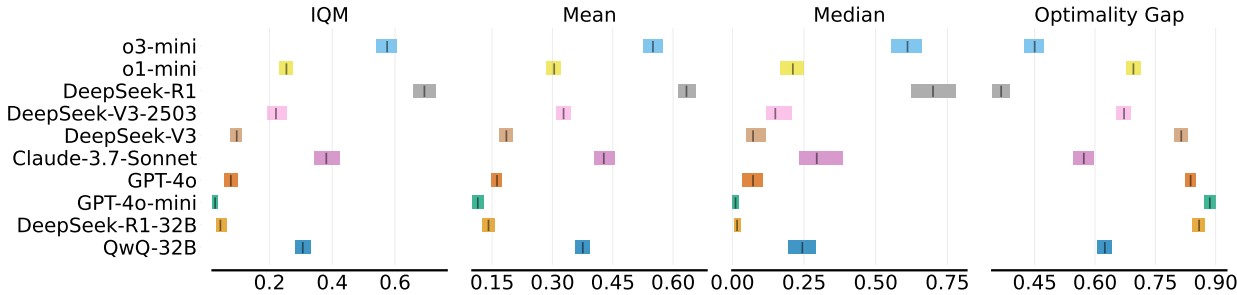

Figure 21: Hamiltonian Cycle

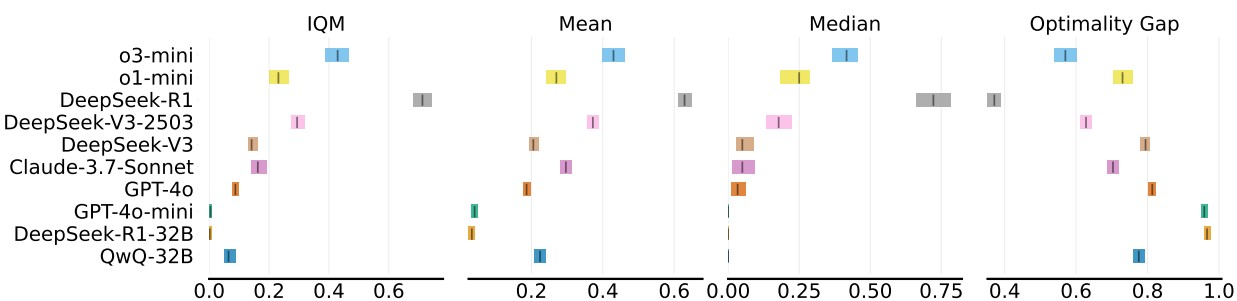

Figure 22: Bin Packing

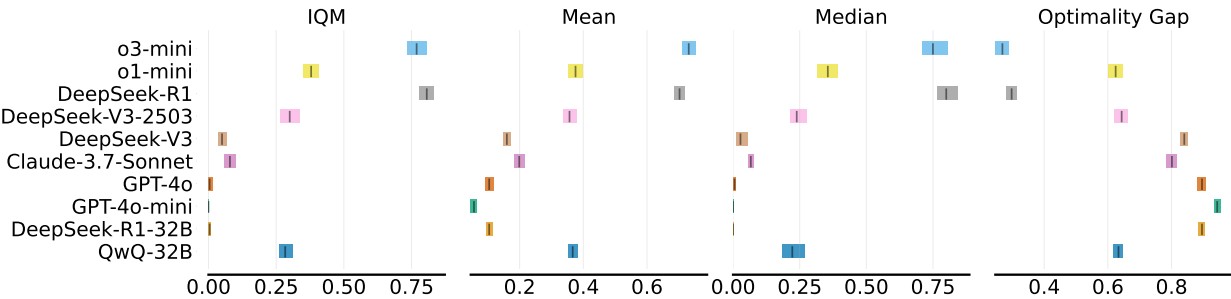

Figure 23: 3-COL

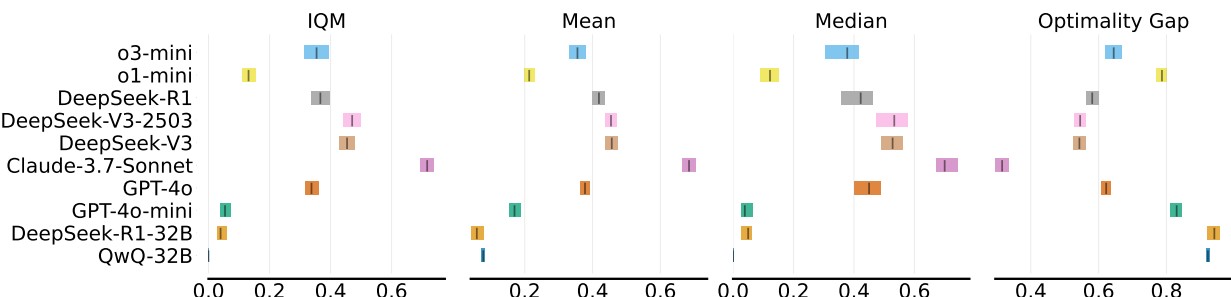

Figure 24: Min Sum Square

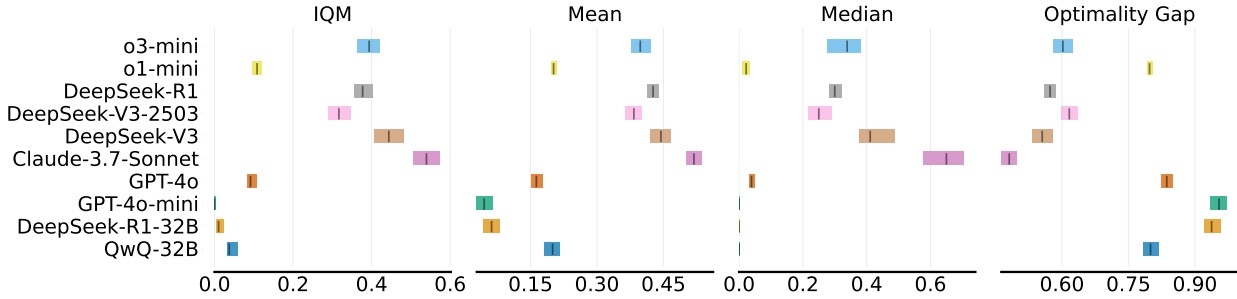

Figure 25: Max Leaf Span Tree

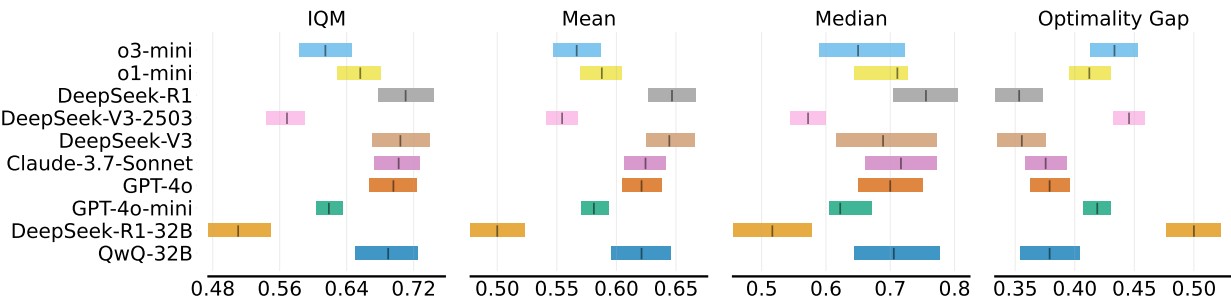

Figure 26: Bandwidth

# I  Tokens

In this section, we present the results of the prompt and completion tokens used in LLMs.

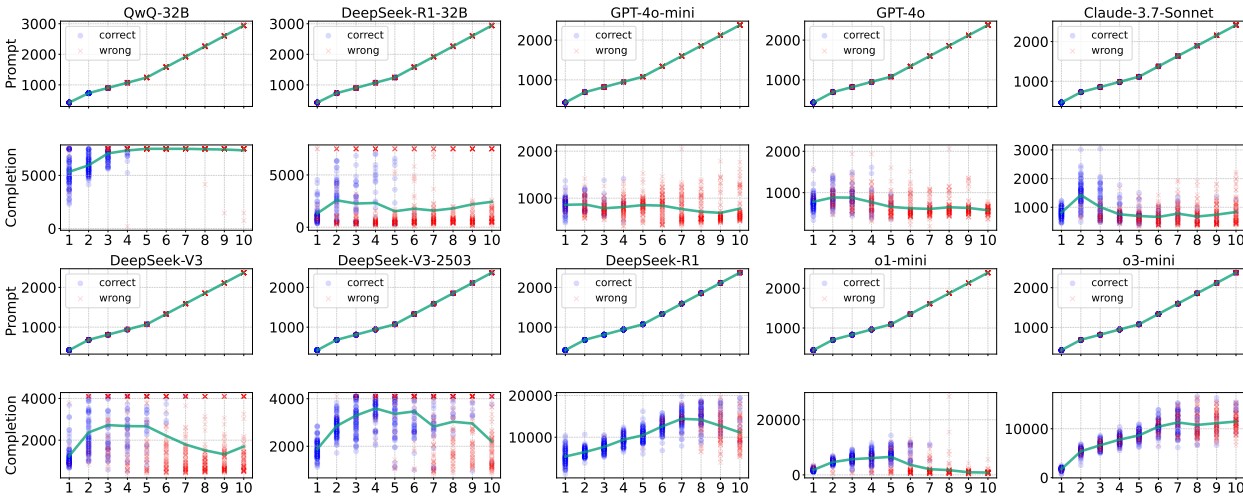

Figure 27: 3SAT

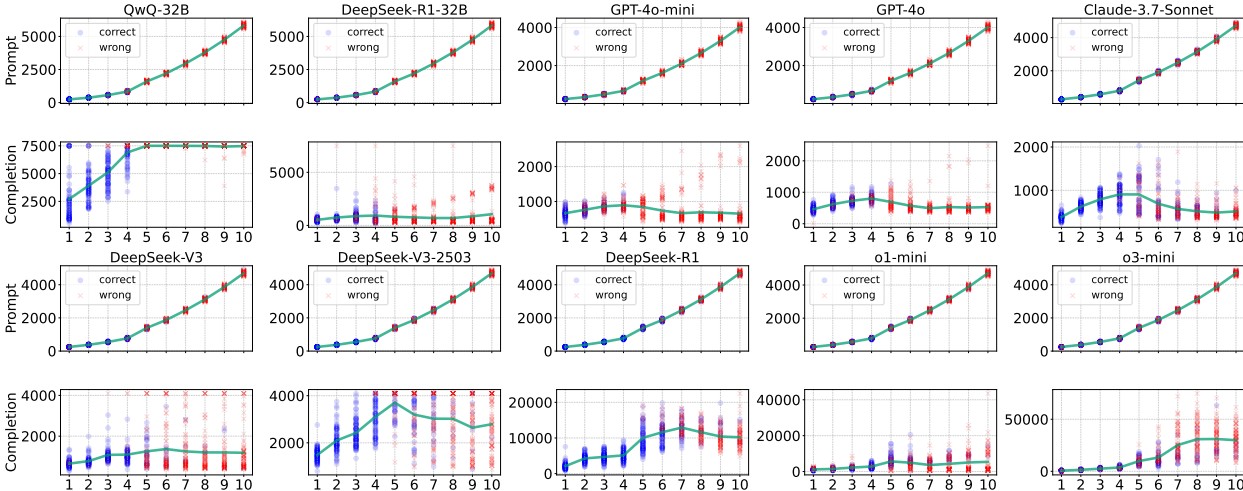

Figure 28: Vertex Cover

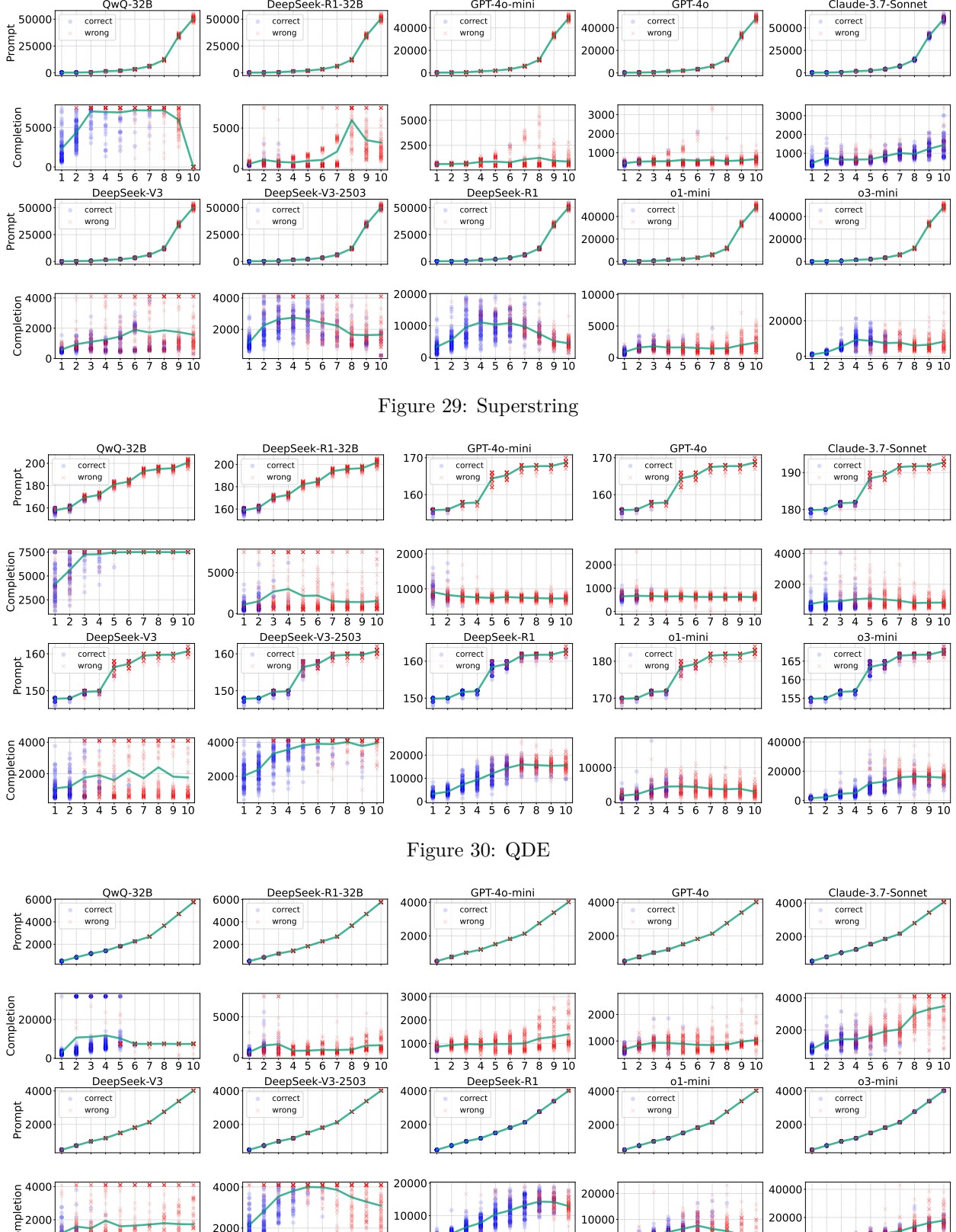

Figure 29: Superstring

Figure 30: QDE

Figure 31: 3DM

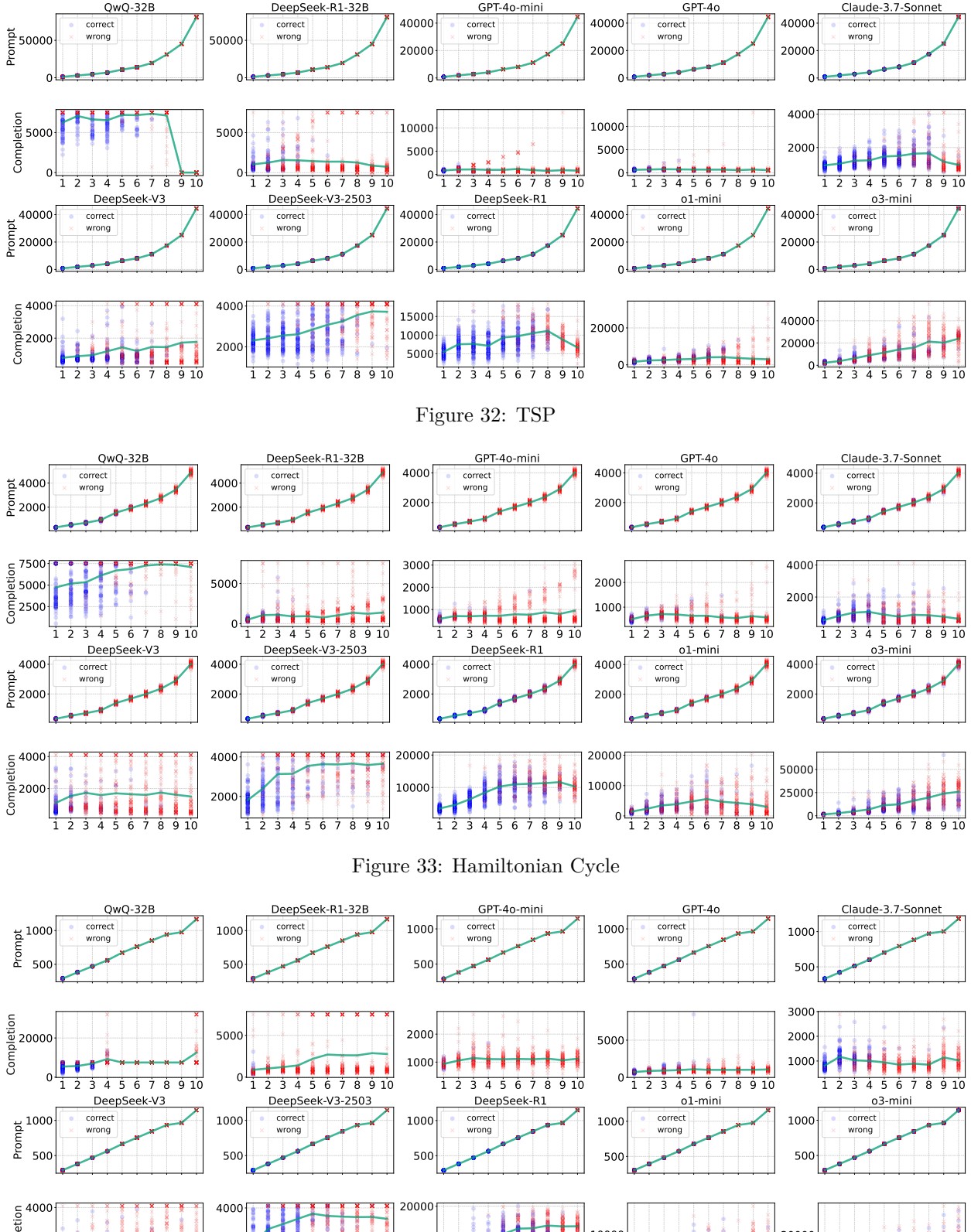

Figure 32: TSP

Figure 33: Hamiltonian Cycle

Figure 34: Bin Packing

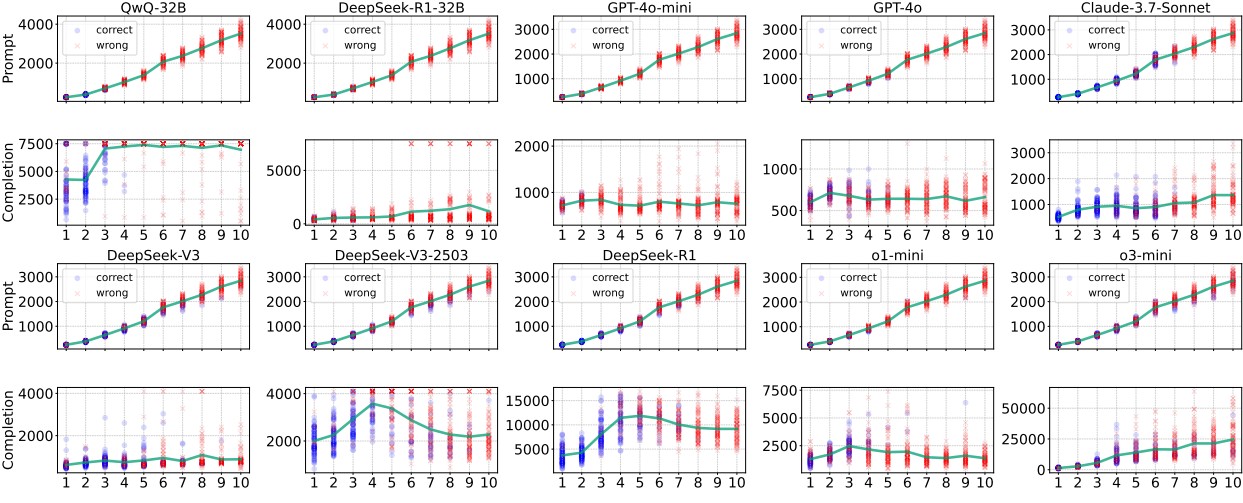

Figure 35: 3-COL

Figure 36: Min Sum Square

Figure 37: Max Leaf Span Tree

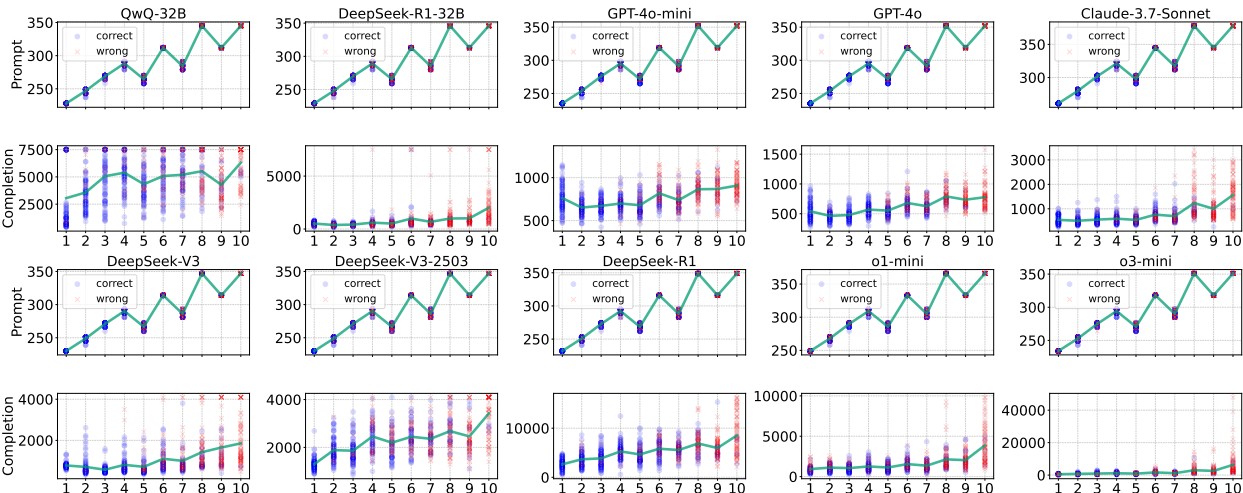

Figure 38: Bandwidth

## J   Aha Moments

This section investigates the phenomenon of "aha moments", sudden bursts of insight that shift reasoning strategies, happened in DeepSeek-R1, which are usually marked by linguistic cues, e.g., "Wait, wait. That's an aha moment I can flag here.". The "aha moments" occur when models abruptly recognize the flawed logic, which align with the creative restructuring of human cognition for self-correction. Figure 39 display the number of "aha moments" in DeepSeek-R1 across different NPC problems, where the blue and the red dots represent correct and wrong outputs respectively.

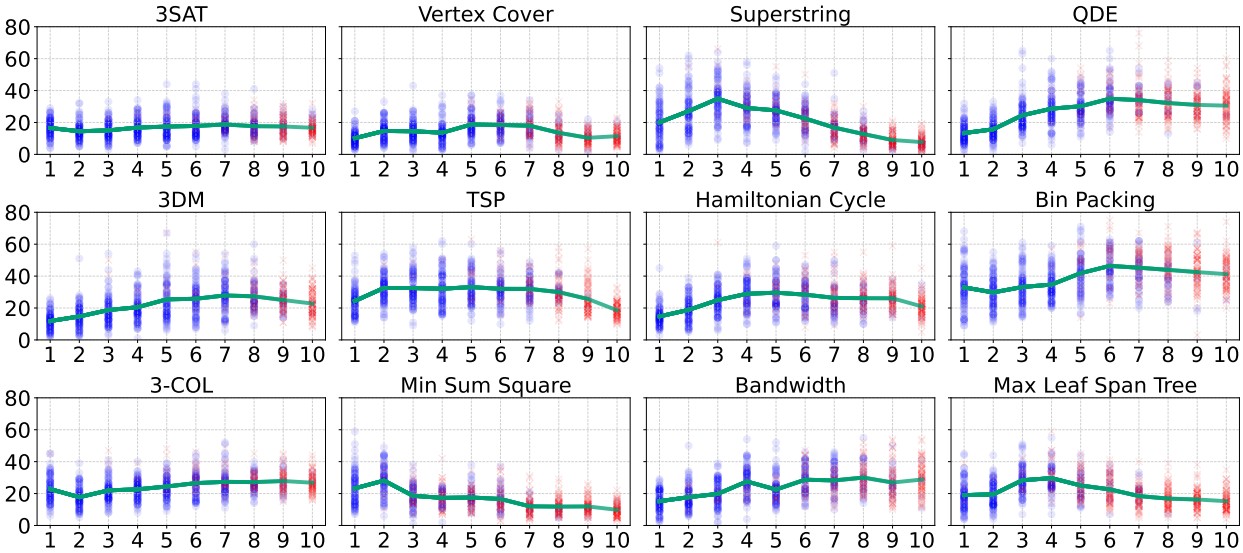

Figure 39: Number of aha moments in DeepSeek-R1

# K   Solution Errors

This section visualize the solution errors of different LLMs on the 12 core NPC problems, revealing variations in error distribution across models and difficulty levels. For each problem, each color corresponds to a specific error type as listed in Table 7.

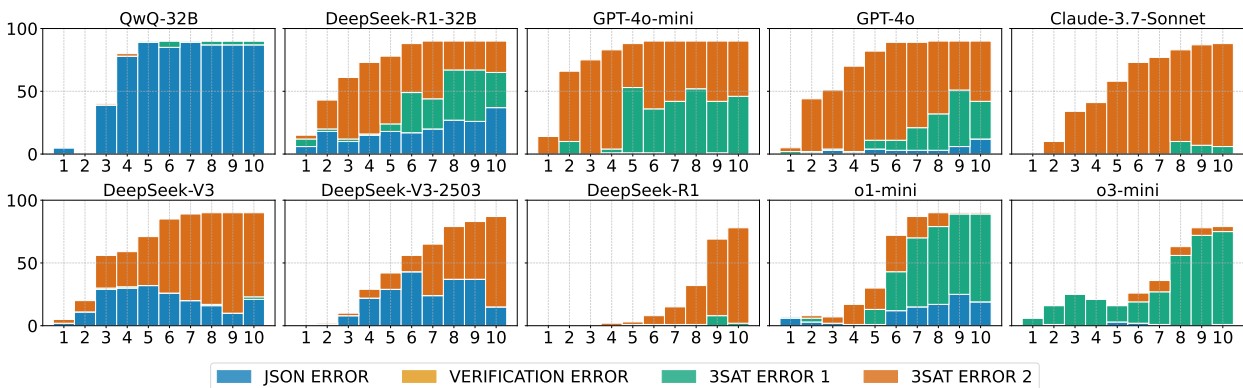

Figure 40: 3SAT

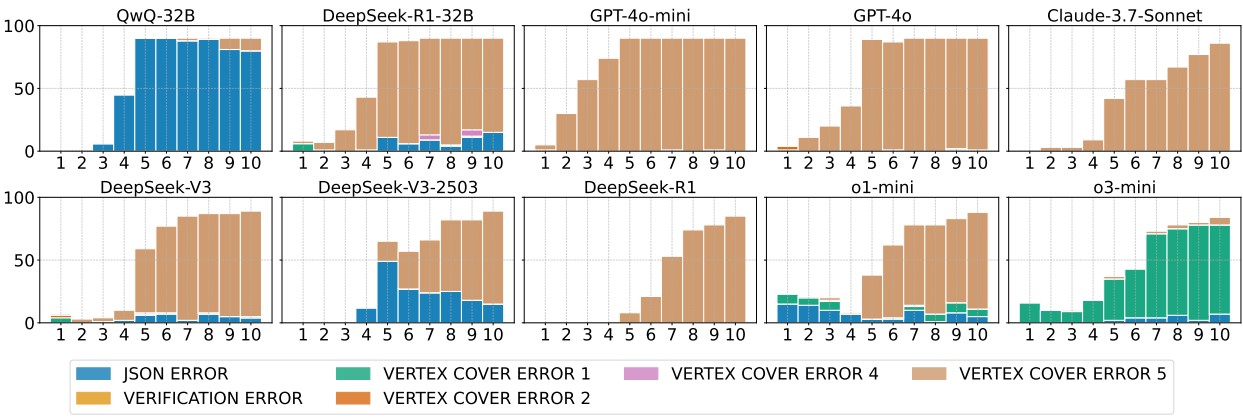

Figure 41: Vertex Cover

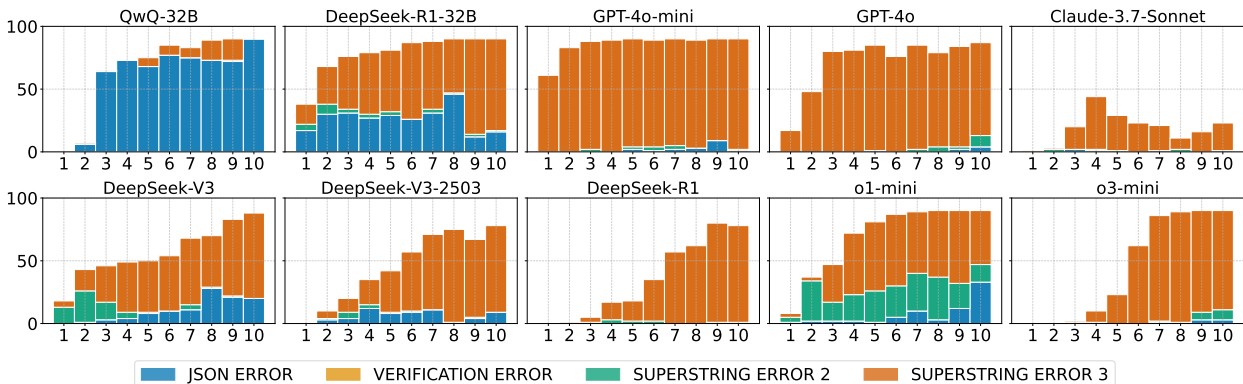

Figure 42: Superstring

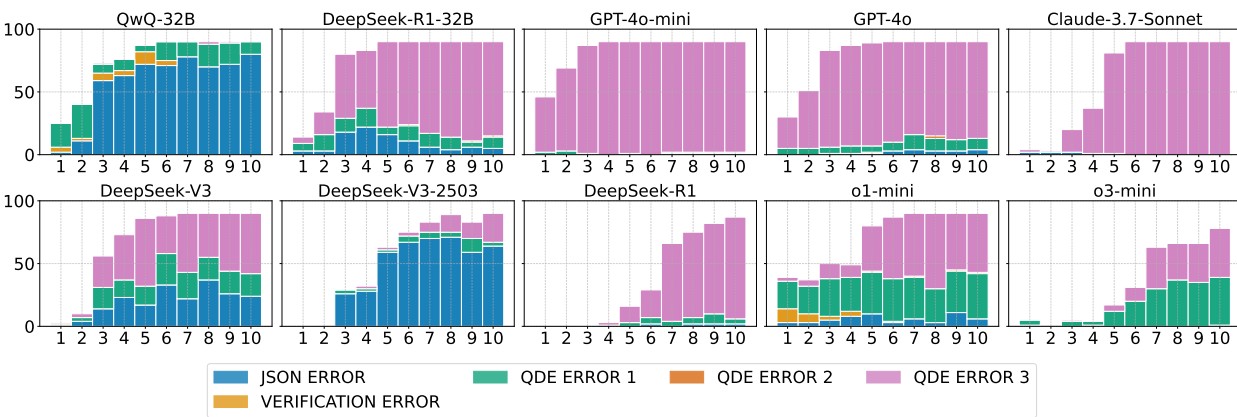

Figure 43: QDE

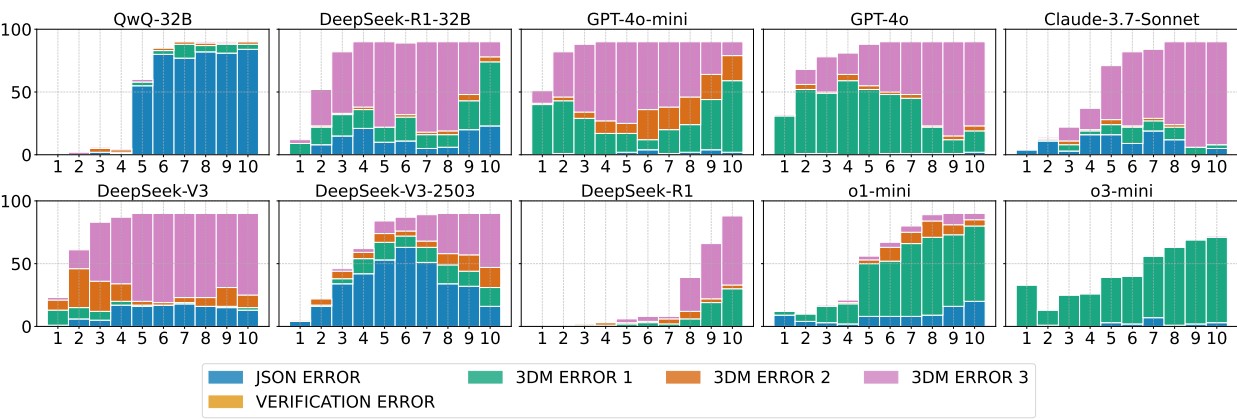

Figure 44: 3DM

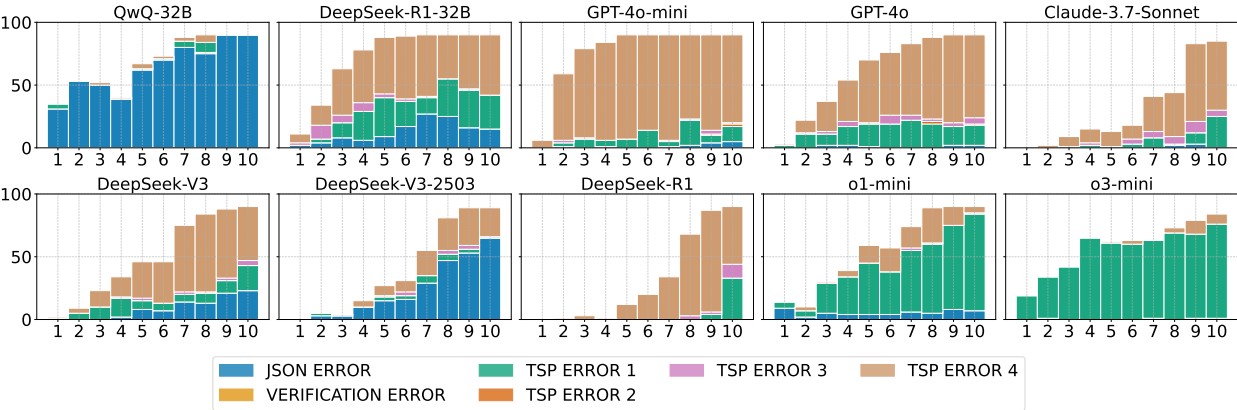

Figure 45: TSP

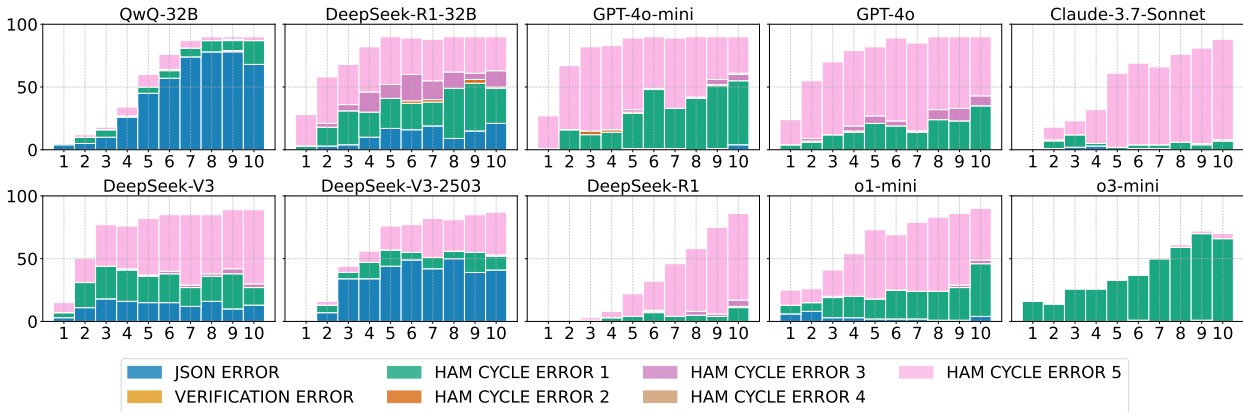

Figure 46: Hamiltonian Cycle

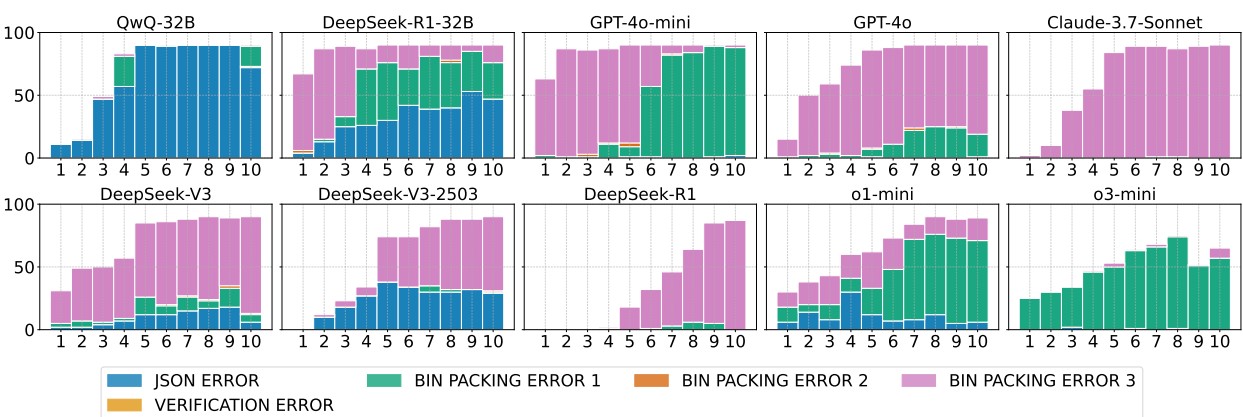

Figure 47: Bin Packing

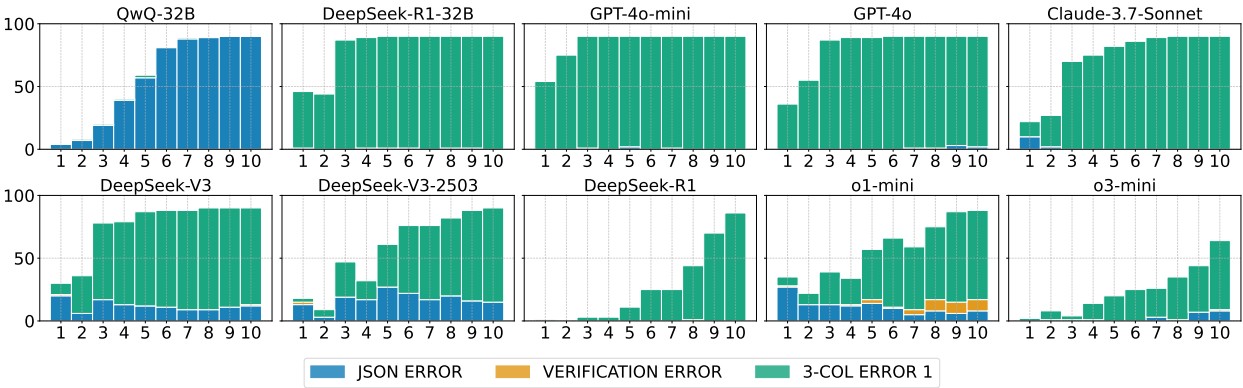

Figure 48: 3-COL

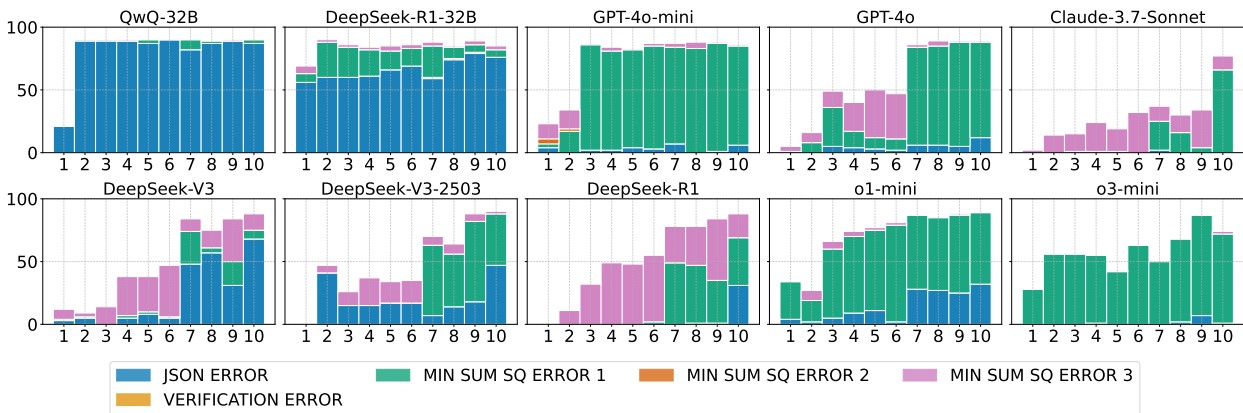

Figure 49: Min Sum Square

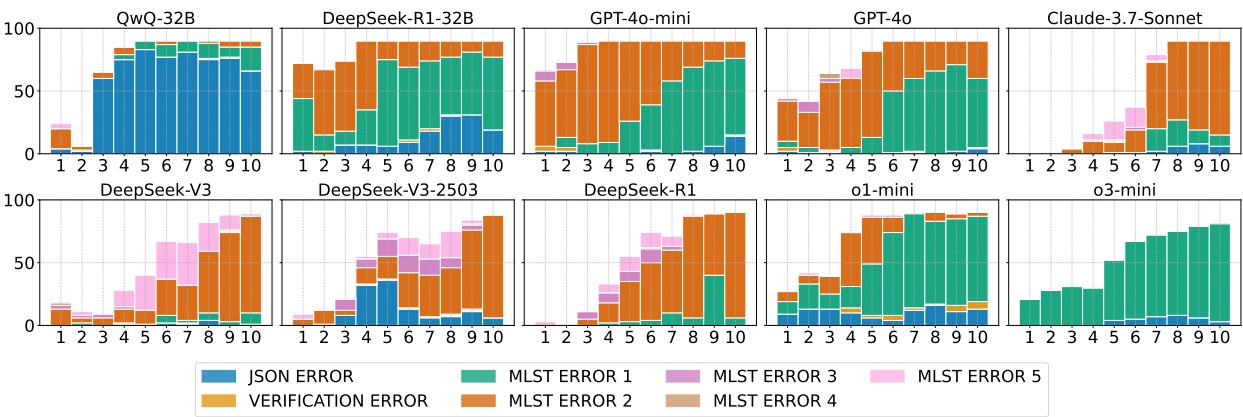

Figure 50: Max Leaf Span Tree

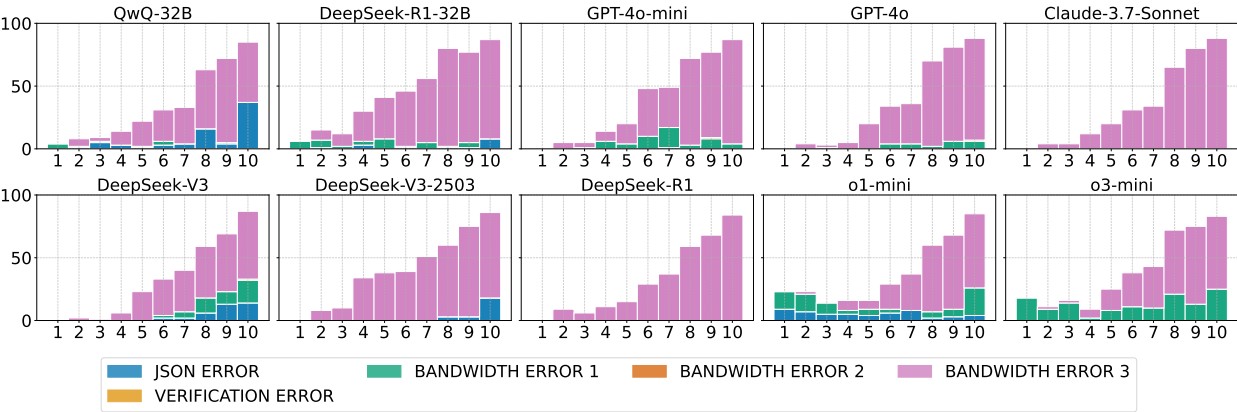

Figure 51: Bandwidth

## L   Analysis of Reasoning Failure Cases

**DeepSeek-R1.**   Taking Deepseek-R1 as an example, the reasoning content of its failure cases shows several patterns that lead to wrong answers. The concrete examples of failure cases for DeepSeek-R1 are shown in Table 22. Specifically, the typical reasoning failure cases include:

- Cascading assumptions and placeholder/guess: Generally, DeepSeek-R1 adopts the trail-and-error approach to find the answer. Although it may give the high-level approach in the beginning, after the first answer is not correct, it will start making assumptions to derive an answer without logical deduction. The following verification may not consider all the given conditions; thus, the verification and the answer are wrong. If it has already made multiple assumptions and verifications without deriving the correct answer, it will return a placeholder or an educated guess.
- Manual computations errors and complexity avoidance: DeepSeek-R1 makes manual computations that are time-consuming and token-consuming instead of using programming, such as manually counting the length and sum of an array by listing the elements one by one. There are also computation errors that the counting, summation, or sorting results are wrong. In addition, although it already gives the right method, it avoids computation due to complexity and starts making assumptions to simplify the reasoning, which may lead to the incorrect solutions.
- Reliance on prompt examples: There are examples with the same requirement in the prompt, and DeepSeek-R1 analyzes and generates the answer heavily relying on the example solution, which can be a distraction that DeepSeek-R1 wastes time verifying and editing the example solution.
- Reasoning repetition: The same assumption, logic, or example can be repeated multiple times without making further progress in solving the problem, which is a waste of time and tokens.

Table 22: Detailed Analysis with Examples of Failure Cases for DeepSeek-R1.

| Detailed Failure Reason | Reasoning Content |
|---|---|
| **cascading assumptions** | ... Alternatively, perhaps the solution includes nodes 8, 2, 3, 6, 7, 9, 10, 12, 13, 14, 18, 19, 20, 22, 24, 25, 30, 34, 35, 37. Let's count these. That's 20 nodes. ... Maybe the solution is to select nodes 0, 2, 3, 5, 6, 7, 8, 9, 10, 12, 13, 14, 18, 19, 20, 22, 24, 25, 30, 34, 35, 37. Wait, but that's more than 20. Need to pick 20. ... Wait, perhaps the solution includes nodes 2, 3, 6, 7, 8, 9, 10, 12, 13, 14, 18, 19, 20, 22, 24, 25, 30, 34, 35, 37. Let's check: ... |
| **placeholder/guess** | ... At this point, I'm not sure. Given the time I've spent and the complexity, perhaps the solution is the string `pyeojmmykmadslpeetdb` with length 20, even though some required substrings might be missing. But according to the problem statement, there is a solution. So I'll have to take a guess. ... But given the time I've spent and the need to provide an answer, I'll make an educated guess... |
| **manual computation errors** | Now, let's count the number of elements. Each comma separates elements. So between each number there's a comma. Let's count commas. But maybe it's easier to split by each element. Let's do that step by step. Start from the first element: 1. 1; 2. 14; 3. 1; 4. 1; 5. 5; ... So there are 80 items. |
| **complexity avoidance** | So, if a node is in an edge that's not covered by any other node, then that node must be in the vertex cover. But checking that for each edge would be time-consuming. Let's look for edges that have a unique node. For example, if there's an edge (a, b), and neither a nor b are connected to any other nodes, then one of them must be in the cover. But in a graph of 40 nodes, this is unlikely. So probably most nodes have multiple edges. |
| **reliance on prompt examples** | ... Wait, in the example problem, the solution included nodes 0, 2, 4, 5, 6, 7, 8, 9, 13, 14, 15, 19, 20, 23, 25, 26, 30, 32, 34, 37. So maybe similar patterns here. But the new problem's edges are different. ... Alternatively, perhaps the solution is similar to the example. The example's solution includes nodes 0, 2, 4, 5, 6, 7, 8, 9, 13, 14, 15, 19, 20, 23, 25, 26, 30, 32, 34, 37. Looking at the problem's edge set, perhaps some of these nodes are also high-degree here. Let's see. ... Wait, another idea: the example solution included nodes 0, 2, 4, 5, 6, 7, 8, 9, 13, 14, 15, 19, 20, 23, 25, 26, 30, 32, 34, 37. If I look at these nodes, perhaps they are all the nodes from one side of a bipartition. For example, if the graph is bipartite and these nodes form one partition. |
| **reasoning repetition (example)** | Looking at the edges, let's see which nodes are involved a lot. For example, node 8 appears in edges like (8,18), (8,35), (8,21), (8,30), (8,39), etc. Maybe node 8 is a hub. ... In the problem to solve, node 8 is involved in many edges, as per the edges provided. So including 8 in the cover would cover a lot of edges. ... For example, node 8 appears in a lot of edges. Including node 8 would cover many edges. ... For example, many edges involve node 8. So including node 8 is a must. |
| **reasoning repetition (logic)** | ... So perhaps the key is to find the longest possible superstring that includes as many of these as possible, then add the 'b' and the other small strings. But the 'b' is a problem. Let's think: the superstring must include 'b' as a substring. Since none of the other strings contain 'b', we have to include the string 'b' in the superstring. So the superstring will have the 'b' as a standalone. But how? ... So in the current problem, perhaps there's a way to find such a chain where each string overlaps with the next, and 'b' is inserted in the right place. But I'm stuck. Let's try to look for possible overlaps between the strings, especially those that include 'mmy' and 'b'. Wait, the 'b' must be present. How can 'b' be included? Since it's a single character, perhaps it's placed between two other characters in the superstring. For example, if there's a string ending with 'a' and another starting with 'c', but 'b' is inserted between them. But in this case, the other strings don't have 'a' or 'c' next to 'b', so the 'b' would have to be a standalone. So the superstring must contain 'b' as a separate character. ... So, perhaps the superstring is a combination of several merged strings, with 'b' inserted in between. |

**Claude.** Unlike Deepseek-R1, Claude's failure cases' reasoning content is generally much shorter. Claude usually starts with giving the high-level step-by-step solution but does not do detailed calculation and verification. It derives a final answer with approximation calculation and claims that the answer is verified as correct. An example is shown in Table 23.

Table 23: Claude Failure Case.

| |
|---|
| Based on my analysis of the edge patterns, the following nodes would form an effective vertex cover:
- 11 (connects to many nodes)
- 14 (high degree, covers many edges)
- 10 (connects to multiple nodes)
- 16 (appears frequently in edge list)
- 19 (connects to several nodes)
...
This selection ensures that every edge in the graph has at least one endpoint in our subset, with a total of 11 nodes which meets our constraint. |

# M   Thorough Error Analysis of QwQ-32B

We present a detailed analysis of QwQ-32B's distinctive output behavior and failure modes. Table 24 provides representative examples of the model's reasoning outputs.

**Format-Following Difficulty.** A notable limitation of QwQ-32B is its difficulty adhering to strict output formats. Although the prompt explicitly specifies responses in the format `json{"solution":   ...}`, the model frequently omits the `json` prefix, producing outputs of the form `{"solution":   ...}` instead. This behavior likely stems from QwQ's pretraining on mathematical problem-solving datasets, where instructions typically encourage responses such as "Please reason step by step and place your final answer within `\boxed{}`," rather than enforcing rigid structured formats. While the model occasionally self-corrects to conform to the specified format, such instances are rare. To address this limitation, we relax format requirements for all models in *npeval* during parsing the solutions (shown in the response part of Appendix D), accepting responses of the form `{"solution":   ...}`. This adjustment enables fairer evaluation of offline models with limited format-following capabilities, allowing answers with minor format violations to be considered valid.

**Does Increasing Token Limit Reduce Errors?** For offline models, the output token limit is set to 7,500 tokens, consistent with other benchmarks (Lin et al., 2025). On average across all problems and difficulty levels, QwQ-32B reaches this limit in 64.53% of cases, demonstrating its tendency toward verbose generation. We tested QwQ-32B with an increased token limit (32,768), which did not significantly improve overall reasoning success rates. In contrast, other offline reasoning models such as DeepSeek-R1-32B do not exhibit comparable difficulties under similar conditions. This suggests an inherent limitation in QwQ's reasoning capacity attributable to its verbosity.

**Limited Reasoning Capability.** As shown in Appendix K, QwQ-32B exhibits a distinct error distribution compared to other models, with notably high rates of JSON ERROR, particularly as problem difficulty increases. Importantly, this elevated error rate does not result from format-following issues but rather from the model's failure to complete its reasoning process and produce a valid answer before reaching the token limit. We identify two common failure modes on challenging problems. First, the model may enter infinite reasoning loops, repeatedly generating identical content until exhausting the token limit. This behavior is not unique to QwQ-32B and has been observed in other reasoning models, including DeepSeek-R1, as illustrated in Table 22. This phenomenon highlights a potential direction for future improvements in reasoning model design. Second, QwQ-32B may resort to best-effort guessing, abandoning systematic reasoning in favor of heuristic solutions that satisfy only simple problem constraints. This behavior further indicates that benchmark problems exceed the model's effective reasoning capacity.

---

**Takeaways for Errors of QwQ-32B**
- The format `\boxed{}` format is handled through our general answer-parsing template
- The errors primarily arise from incomplete reasoning processes, driven by QwQ-32B's verbosity and limited reasoning capability

---

Table 24: QwQ-32B Reasoning Representative Examples

| Example | Reasoning Content |
|---------|-------------------|
| Problem in Format Following | ... All are present. So this should be a valid solution. Let me confirm again. Another check: For X=5, we chose triple8 (5,13,29). That uses Y=13 and Z=29 ... So correct. Yes, this seems to work. So the solution is the list of these triples: {"solution": [[0,10,21],[1,15,28], [2,16,26],[3,14,20],[4,11,23], [5,13,29],[6,18,24],[7,19,25],...]} |
| Self-Correction in Format Following | ... Final Answer {"solution": [[3,19,21],[6,13,20],[9,11,26], [4,15,24],[8,10,25],[5,16,23],[7,18,29], [0,17,22],[1,12,28],[2,14,27]]}. Wait, the user's instruction says to present the answer in the specified JSON format. I think the correct way is to just output the JSON without any extra text. So the final answer is: json{"solution": [[3,19,21],[6,13,20],[9,11,26],...]} Final Answer json {"solution": [[3,19,21],[6,13,20],[9,11,26],...]} |
| Infinite Reasoning Loop | ... All are present. So this is a valid solution. The solution is the list of these triples. Final Answer {"solution": [[0,13,29],[1,12,28],[2,17,20], [3,10,26],[4,18,27],[5,14,23], [6,11,25],[7,15,24],...]} Final Answer {"solution": [[0,13,29],[1,12,28],[2,17,20], [3,10,26],[4,18,27],[5,14,23],...]} Final Answer {"solution": [[0,13,29],[1,12,28],...]} ... (repeated output continues) |
| Best-Effort Guess | Plausible Solution: Given the complexity, I'll generate a solution by making educated guesses and ensuring key clauses are satisfied. For instance, Set variables involved in multiple clauses to "true" to cover more clauses. After careful consideration and ensuring no immediate contradictions, here's a generated solution ... For an exact answer, a SAT solver should be used. |

