# OpenReview forum: "Nondeterministic Polynomial-time Problem Challenge: An Ever-Scaling Reasoning Benchmark for LLMs"
_TMLR — Accepted by TMLR_

### Review · Reviewer_JPDw · 2025-12-09

**Summary Of Contributions:**

This paper introduces the Nondeterministic Polynomial-time Problem Challenge (NPPC), a reasoning benchmark designed to address the rapid saturation and potential data contamination of existing LLM benchmarks. NPPC is a benchmark that can scale over complexity, instance, oversight, and coverage to ensure continuous differentiation among LLMs over extended periods, identifying fundamental limitations for further improvement. The benchmark leverages 25 NP-complete (NPC) problems (e.g., 3SAT, Vertex Cover, 3DM) to generate procedurally verifiable instances of arbitrary difficulty. Three components, including npgym, npsolver, and npeval are designed for problem generation, unified model interaction, and statistical evaluation, respectively. Extensive experiments over widely-used LLMs demonstrate that NPPC can reduce model accuracy to below 10% at high difficulty levels, validating its resistance to saturation.

### Strengths
1. The benchmark utilizes the unique properties of NP-complete problems—specifically that they are computationally hard to solve but easy to verify to create an ever-scaling benchmark. This ensures the benchmark can scale indefinitely in complexity without requiring human curation.
2. The framework, comprising three modules (npgym, npsolver, npeval), offers a complete pipeline for generation, interaction, and rigorous statistical analysis. Meanwhile, the framework supports both online (API-based) and offline (locally deployed) models, facilitating broad comparisons between proprietary and open-weights models.
3. Extensive experiments and in-depth analysis verify that NPPC is a ever-scaling benchmark for reliable and rigorous evaluation of the reasoning limits of LLMs.

### Weaknesses
1. The evaluation protocol strictly requires responses in a specific JSON format. The analysis reveals that for some models (e.g., QwQ-32B), "JSON ERROR" dominates the failure modes, meaning the model might have reasoned correctly but failed the format check. This obscures the true reasoning capability of the models. A reasoning benchmark should isolate reasoning ability; strict formatting penalties introduce confounding variables, especially for smaller or offline models
2. The prompt template is rigid regarding the output format, and while offline models have defined parameters (Table 8), the lack of a "retry" mechanism or flexible parsing for syntax errors (beyond basic regex) limits the fairness of the comparison.
3. The authors claim "scaling over coverage" ensures relevance to real-world problems. While NPC problems are theoretically foundational, the paper does not empirically demonstrate that improving on abstract 3SAT or Hamiltonian Cycle tasks correlates with performance on practical downstream tasks.

**Audience:**

Yes

**Audience Explanation:**

The paper addresses the critical issues of benchmark saturation and potential data contamination by introducing the Nondeterministic Polynomial-time Problem Challenge (NPPC), an ever-scaling framework that utilizes procedurally generated NPC problems to resist obsolescence. The submission offers relevant empirical data through a rigorous evaluation of state-of-the-art reasoning models, including DeepSeek-R1, Claude-3.7-Sonnet, and o1/o3-mini, identifying performance limits that current static benchmarks fail to capture. Furthermore, the work provides novel methodological contributions regarding scalable oversight, leveraging the "hard to solve, easy to verify" property of NPC problems, and delivers deep insights into inference-time compute behaviors, such as token usage and "aha moments". Finally, the detailed taxonomy of reasoning failures offers actionable feedback crucial for researchers aiming to advance model architectures and training strategies.

**Claims And Evidence:**

Yes

**Claims Explanation:**

The submission is distinguished by an accurate and robust theoretical foundation, leveraging the definition of NPC problems, specifically their property of being computationally hard to solve but easy to verify, to logically support its framework for scalable oversight and complexity. The efficacy of the authors' two-stage difficulty calibration, which combines expert configuration with empirical validation, is convincingly evidenced by the monotonic degradation of model performance observed in the results, ensuring generated instances accurately reflect increasing complexity. Furthermore, the claim of the benchmark's "uncrushable" nature is supported by clear experimental data showing a consistent performance collapse across all tested models, including state-of-the-art reasoning models, as difficulty scales. Finally, the analysis offers granular insights into model behavior through clear evidence regarding token consumption and "aha moments," visualizing distinct trends during inference.

**Requested Changes:**

1. (Major) The current evaluation protocol conflates reasoning capabilities with instruction-following strictness. Implement a more robust parsing mechanism (e.g., using an LLM-based extraction step for malformed JSON or more permissive regex) to separate reasoning accuracy from formatting adherence. Alternatively, report "Reasoning Accuracy" (manual/soft-check) and "Format Compliance" as separate metrics to provide a fairer assessment of offline models like QwQ-32B.
2. (Minor) Include a correlation analysis comparing model performance on NPPC against established applied benchmarks. This would empirically validate the claim that "coverage" of NPC problems translates to "practical value" and "downstream tasks" as argued in the introduction.

---

> ### Author Response · Authors · 2025-12-16
> **Response from Authors**
>
> We sincerely appreciate the reviewers for their valuable feedback and insightful comments. We hope our following answers will clear up the doubts about our work, and please let us know if there is any other clarification we can provide.
>
> ---
>
> **1. JSON Format**
>
> JSON format serves as the canonical output format for advanced LLM models, with instruction following representing one of their core capabilities. Our prompt template follows the design used in ZebraLogic (Lin et al. 2025), while our solution parsing template (detailed in Appendix D) is intentionally designed to be maximally general, enabling robust extraction of solutions from diverse response formats. We also want to note that as we need the program to automatically verify the solution, we have to specify the format of the solution for auto verification, and using LLMs to verify the solution may not be reliable.
>
> Our analysis reveals a clear performance pattern related to problem difficulty. For easier problems, LLMs consistently produce solutions in the correct format, even when the solutions themselves are incorrect. However, as problem complexity increases and extended reasoning processes become necessary, LLMs increasingly struggle to maintain proper formatting compliance. This trend is particularly pronounced in Figure 9, with DeepSeek-v3 and o1-mini exhibiting notably higher format error rates on challenging problems.
>
> ---
>
>
> **2. Analysis of QwQ-32 JSON Errors**
>
> Following the reviewer's suggestion, we conducted a comprehensive analysis of the elevated JSON error rates observed in the QwQ-32B model, which is displayed in Appendix M. Specifically,
>
> 1. we first reaffirm that the general parsing template can help to mitigate the errors of generating incorrect formats of the solutions, which is particularly useful for QwQ-32B model, as it is trained with different formats of the solutions, i.e., \box{}.
>
> 2. we then double-check the influence of the token limits. We set 7500 tokens in the experiments by following the existing work (Lin et al., 2025) and 64.53% responses reach the limit. We then increase the token limit to 32,768, and do not see the significant improvement of the performance. The same token limit is applied to DeepSeek-R1-32B, and we do not observe so many JSON errors.
>
> 3. Finally, we report that QwQ-32B is very verbose, where it may cost many tokens without advancing the reasoning, and therefore, the model cannot complete the reasoning within the token limit. We also identify the two common failures, i.e., repeatedly reasoning loop and best-effort guessing without efficient reasoning.  We present the comprehensive analysis including examples in Appendix M, and we hope this thorough analysis can address the reviewers’ concerns.
>
> ---
>
> **3. Correlation with Established Benchmarks**
>
> To assess NPPC's relationship with existing evaluation frameworks, we performed correlation analysis against three well-established benchmarks: MMLU, GPQA-Diamond, and LMArena. The results (displayed in Section A.9) demonstrate strong positive correlations between NPPC and these benchmarks, supporting NPPC's validity as a reasoning assessment tool.
>
> ---
>
> **4. NP-Complete Problems for Training Data Generation**
>
> We note that several recent studies have leveraged NP-complete problems to generate reasoning data for model training, with the goal of improving general reasoning capabilities. This growing research direction is discussed in the third paragraph of Section 6, highlighting the broader utility of NP-complete problems in advancing LLM reasoning abilities.

---

### Review · Reviewer_gQLy · 2025-12-12

**Summary Of Contributions:**

**Sumary**

The paper introduces the Nondeterministic Polynomial-time Problem Challenge (NPPC), a new benchmark designed to evaluate the reasoning capabilities of LLMs. The authors argue that existing benchmarks suffer from saturation (crushing) or contamination (hacking), resulting in several models achieving very high performance in a short period after the benchmarks are released. To address this, the authors define ever-scalingness, four dimensions for the building of an effective benchmark: (i) complexity, adjustable and increasing difficulty, (ii) instances, infinite generation of evaluation samples, (iii) oversight, fast verification of the predictions, (iv) coverage, inclusion of diverse domains. To this end, the NPPC framework is compiled, comprising 25 NP-complete problems (12 core, 13 extension) that cover the aforementioned requirements. NPPC framework is implemented via: (i) npgym, which generates instances and verifies the model predictions, ii) npsolver, which provides a standardized pipeline for prompting and output parsing of various models, and iii) npeval, which systematically evaluates the performances of LLMs using various metrics (e.g. mean, IQM) and on various aspects (e.g. token usage, aha moment counts, and error types). Several contemporary offline and online models are evaluated, highlighting that the proposed benchmark is far from saturation, with the best models achieving an accuracy of less than 10%, and performance degrading with an increase in difficulty.

**Strengths**

* The contribution of the paper is significant. The NPPC is an evidently scalable benchmark that comprises many NPC problems, which are naturally of increasing complexity and have an infinite pool of instances. Additionally, the difficulty of the problems can be further increased, making it a long-lasting benchmark and an excellent resource for future research. Hence, it is of great value for the research community working on reasoning-related problems, a very contemporary topic.
* The definition of the ever-scalingness is sound and interesting. All four dimensions are deemed necessary for building an effective benchmark that cannot (or at least it is hard to) be saturated, and also reflect real-world reasoning.
* The framework implementation is technically rigorous, thorough, and will be fully open-sourced. The three components (npgym, npsolver, npeval) are well designed, and their functionality is clear. The automatic generation and verification of the npgym is clear, the prompting and completion of the npsolver is straightforward, and the evaluation protocol of the npeval covers various performance aspects.
* An extensive experimental study is provided based on 10 recent LLM models. The models are used either directly offline or via their online APIs, and reasoning and non-reasoning LLMs are evaluated. Also, an insightful analysis of performance is provided, breaking down the types of errors made by the models, the number of output tokens, as well as the aha moments of DeepSeek-R1.

**Weaknesses**

All weaknesses stated below focus on clarity and presentation of the paper.
1. The coverage dimension of the dataset is not sufficiently discussed. Although the other three dimensions are well covered due to the nature of NPC problems, it is very clear how and why high performance on such tasks translates to real-world reasoning. Especially, the high difficulty regime might result in artificial problems with limited practical application. Addressing such concerns would strengthen the paper's motivation, and the benchmark would reach a broader audience.

2. It is not very clear how the aha moments are defined, how they are quantified, and what they capture. Several references to them appear throughout the paper without a proper introduction. Additionally, there is no discussion of the results, and it is unclear what the outcome of such an analysis is. In Figure 36, the aha moments seem to be relatively uniform without following a particular pattern. In the same spirit, the importance of the number of output tokens is not justified early in the paper; however, its quantification is straightforward, and its impact is adequately discussed.

3. Could the tool usage by models be a game-changer for this benchmark? There is relevant discussion in the paper, but it is unclear whether LLMs that have access to specialized solvers (or even code execution) for the particular problems can easily exploit them to solve the tasks. Although this may not be deemed as actual reasoning, it can be a way to hack the benchmark.

4. The definition of difficulty needs to be better clarified. There is a distinction between "Computational Hardness" and "LLM Hardness", and it is not clear how well they are aligned. While the authors acknowledge that "The difficulty experienced by traditional symbolic solvers does not necessarily translate to difficulty for LLMs", the paper would benefit from a closer discussion on why the specific parameters were chosen for each problem. It appears that the hardness for LLMs comes from context length and variable tracking rather than combinatorial explosion in the traditional sense. Moreover, it is mentioned that " the difficulty levels are further calibrated through systematic empirical testing with state-of-the-art LLMs". Could this introduce bias against the models used for calibration, thereby giving future models an unintended advantage over the current ones?

5. Although the paper contains extensive implementation details, it lacks a dedicated section that consolidates them in an accessible manner. As a result, readers do not have a reference point for locating specific information, since these details are sparsely presented throughout the paper. For example, the compared models are categorized into reasoning vs. non-reasoning and offline vs. online, which can be difficult to follow for readers unfamiliar with these distinctions.

6. Some models have issues with the JSON formatting, raising corresponding errors. It is unclear whether this high failure rate is due to reasoning failure or simply an inability to follow the JSON schema.

**Audience:**

Yes

**Audience Explanation:**

Researchers working in the field of reasoning and evaluation of LLMs will greatly benefit from the proposed benchmark.

**Broader Impact Concerns:**

No Broader Impact Statement is provided. Some topics that could/should be discussed are the following:
1. Compute and environmental cost. The large-scale evaluations encouraged by NPPC may increase consumption requirements, especially as the difficulty increases. Therefore, access to significant computational resources is necessary, which may have an environmental impact.
2. Overemphasis on synthetic NPC tasks. Models may be trained to overfit NPPC, potentially diverting attention from real-world robustness and safety issues.
3. Risk of benchmark misinterpretation. High NPPC scores could be misconstrued as general reasoning capabilities.

**Claims And Evidence:**

Yes

**Claims Explanation:**

The main claim is that the proposed benchmark is far from saturation, which is effectively demonstrated by evaluating 10 widely used LLMs that performed poorly in high difficulty settings. Also, the importance of ever-scalingness is convincing and is adequately satisfied by the proposed benchmark.

**Requested Changes:**

All suggestions below would strengthen the work in the reviewer's view:

1. Better motivation for the coverage aspect of the benchmark. An extension of the existing segment in the introduction (or even a new paragraph) would suffice.
2. Introduction of the aha moments in an early stage of the paper. Discussion of the results, if there is any useful conclusion. Otherwise, consider removing the corresponding results altogether, as they are only with a single model. This would reduce clutter in the presentation and improve clarity. Additionally, it would be more beneficial to focus on analyzing the different error types and provide a more in-depth discussion about what they represent.
3. A small-scale ablation with some models with tool usage would address such concerns. Otherwise, some relevant discussion could be added in the corresponding segment.
4. A more detailed explanation of the chosen parameters would help clarify how they correspond to different levels of difficulty.
5. A brief summary in the main paper (or a table/figure) of the evaluation protocol would address the sparsity of the implementation details. For example, a paragraph in the results (Section 5) that includes the core details, such as the number of instances per problem, and provides references to the relevant sections in the Appendix would resolve the issue. Also, including a table that summarizes the compared LLMs and their types, in the context of reasoning/non-reasoning, offline/online, or any other categorization, would also improve clarity.
6. A small-scale ablation of the different prompt templates would address the formatting issue. Such ablation could be focused solely on the LLMs that exhibit these issues.
7. Some very minor suggestions regarding visuals in the paper, for better consistency and clarity:
    * Figure 6: Although visually appealing, it reiterates the same information shown in Figure 5. Is there anything new that can be derived from this plot?
    * Figure 5: Some notation in the plot's caption to distinguish reasoning/non-reasoning and offline/online would help.
    * Figure 7: Indicating whether each metric is "higher is better" or "lower is better" would make the comparison more accessible. A simple arrow next to each metric name could be sufficient. Additionally, adding borders or clearer separation between the plots would help distinguish them visually.
    * Figure 8, 9: For clarity and visual consistency, it would be preferable to be top-aligned.

---

> ### Author Response · Authors · 2025-12-16
> **Response from Authors**
>
> We sincerely appreciate the reviewers for their valuable feedback and insightful comments. We hope our following answers will clear up the doubts about our work, and please let us know if there is any other clarification we can provide.
>
> ---
>
> **1. The necessities of solving large-scale NPC problems**
>
> We add the discussion about the necessities of solving large-scaling NPC problems in Appendix A.2. Many critical applications in logistics and supply chain optimization, computational biology, and telecommunications inherently require solving NPC problems at scale. The computational challenges posed by these domains grow naturally with technological advancement and increasing data availability. Therefore, our ever-scaling benchmark design directly reflects the genuine need to develop AI systems capable of tackling NPC problems of progressively increasing complexity, reflecting the growing demands of real-world applications.
>
> ---
>
> **2. Tool Use for NPPC**
>
> Conducting comprehensive tool use experiments presents two practical challenges for current NPPC: i) The first primary challenge lies in tool selection: different NP-complete problems naturally require different tools, and determining the appropriate tool set for each problem instance would require substantial additional framework design, and ii) the second challenge is incorporating tools during problem-solving introduces an additional dimension of evaluation, i.e., the tool calling capabilities of LLMs, which extends beyond our primary objective of benchmarking pure reasoning capabilities.
>
> Therefore, following the reviewer’s suggestions, we add a comprehensive discussion about the influence of tool use on NPPC in Appendix A.5. Our position is that while tool augmentation can substantially improve LLM performance on NPPC tasks, it does not invalidate the benchmark's core value. Critically, even with tool assistance, NPC problems remain computationally intractable, i.e., they cannot be solved efficiently in polynomial time. Therefore, tool-augmented LLMs still face the fundamental computational barriers that make NPPC challenging, and the benchmark continues to provide meaningful evaluation of reasoning capabilities in tool-enhanced settings. In this sense, NPPC remains valuable for assessing both standard and tool-augmented LLMs, with the understanding that tool availability shifts but does not eliminate the fundamental complexity challenges.
>
> ---
>
> **3. The specification of the parameters of NPC problems, as well as the determining of difficulty levels**
>
> We add more discussion about the specification of parameters in the difficulty levels of NPC problems. Specifically, Our methodology uses a two-stage process:
> 1.  Human-defined difficulty parameters: We start with interpretable problem parameters. For SAT, this means specifying (num_variables, num_clauses) such as (3, 5), (4, 5), or (100, 100). While we can confidently say (100, 100) is harder than (5, 5), distinguishing between similar configurations like (5, 4) versus (4, 5) is non-trivial;
> 2. LLM-based calibration: We use model performance for two specific purposes: i) Fine-grained ordering: Sorting problems of similar complexity for clearer visualization (e.g., Figure 5), particularly when human intuition cannot definitively rank them. ii) Range validation: Ensuring difficulty levels fall within a meaningful range, i.e., avoiding settings where all models achieve 100% (too easy) or 0% (too hard) accuracy, as neither scenario effectively benchmarks capabilities.
>
> We carefully revise Section 4.1 and Appendix A.6 to clearly explain the process of determining the difficulty levels to avoid any confusion.
>
> ---
>
> **4. JSON format error**
>
> JSON format serves as the canonical output format for advanced LLM models, with instruction following representing one of their core capabilities. Our prompt template follows the design used in ZebraLogic (Lin et al. 2025), while our solution parsing template (detailed in Appendix D) is intentionally designed to be maximally general, enabling robust extraction of solutions from diverse response formats.
>
> Our analysis reveals a clear performance pattern related to problem difficulty. For easier problems, LLMs consistently produce solutions in the correct format, even when the solutions themselves are incorrect. However, as problem complexity increases and extended reasoning processes become necessary, LLMs increasingly struggle to maintain proper formatting compliance. This trend is particularly observed in Figure 9, with DeepSeek-v3 and o1-mini exhibiting notably higher format error rates on challenging problems.

---

> > ### Author Response · Authors · 2025-12-16
> > **Response from Authors (cont.)**
> >
> > **5. Analysis of QwQ-32 JSON Errors**
> >
> > We conducted a comprehensive analysis of the elevated JSON error rates observed in the QwQ-32B model, which is displayed in Appendix M. Specifically,
> > 1. we first reaffirm that the general parsing template can help to mitigate the errors of generating incorrect formats of the solutions, which is particularly useful for QwQ-32B model, as it is trained with different formats of the solutions, i.e., \box{}.
> >
> > 2. we then double-check the influence of the token limits. We set 7500 tokens in the experiments by following the existing work (Lin et al., 2025) and 64.53% responses reach the limit. We then increase the token limit to 32,768, and do not see the significant improvement of the performance. The same token limit is applied to DeepSeek-R1-32B, and we do not observe so many JSON errors.
> >
> > 3. Finally, we report that QwQ-32B is very verbose, where it may cost many tokens without advancing the reasoning, and therefore, the model cannot complete the reasoning within the token limit. We also identify the two common failures, i.e., repeatedly reasoning loop and best-effort guessing without efficient reasoning.
> >
> > We present the comprehensive analysis including examples in Appendix M, and we hope this thorough analysis can address the reviewers’ concerns.
> >
> >
> > ---
> >
> > **6. Other changes following the reviewer’s suggestions**
> >
> > We also introduce the following changes in the manuscript following the reviewer’s suggestions:
> >
> > a.	Revising the introduction section to better motivate the scaling of coverage
> >
> > b.	Moving the analysis of aha moment to the appendix to make the main context more readable and focusing
> >
> > c.	Adding a brief summary of the experiments at the beginning of Section 5.
> >
> > d.	We also revise the figures to make it more readable and informative
> >
> > ---
> >
> > **7. Broader impact statement**
> >
> > We also add the broader impact statement (Appendix A.10) to address the potential concerns about NPPC.

---

### Review · Reviewer_GmXh · 2025-12-13

**Summary Of Contributions:**

Paper introduce a novel NPPC benchmarking framework to evaluate the LLM's reasoning capabilities. It involves three modules: npgym, npsolver, and npeval targeting problem generation, verification, and analysis of the performance tools, respectively.

The idea is to draw problems from the class of NP-complete problems, which are difficult to solve but easy to verify, and evaluate the performance of LLMs over these problems for reasoning. NP-complete class properties are exploited to generate problems with increasing "complexities" by a two-stage method: manual configuration by experts followed by empirical calibration.

My main concern is about the way difficulty levels are constructed. Once mathematical complexity of the problems is identified, the calibration step seems cyclic. Calibration decides problems as "difficult" if they lower the LLMs performance. The final conclusion then says that as the difficulty levels increase, LLMs performance decreases.

**Audience:**

Yes

**Audience Explanation:**

The topic of benchmarking is of interest to many in the field as reasoning models are becoming increasingly important and there is a need of benchmarking frameworks that can scale indefinitely. But, the logic of difficulty notion defined in the paper needs more clarification.

**Broader Impact Concerns:**

No broader impact concerns.

**Claims And Evidence:**

No

**Claims Explanation:**

The logic of difficulty Vs performance seems circular. There should be comparison with the results with computational complexity (step I (theoretically hard assigned by experts)) Vs LLMs performance, with the current plots of difficulty levels (step I (theoretically hard assigned by experts) + II (LLM calibration)). In the current settings, the plots seem redundant as the problems are labelled difficult by that same performance condition.

**Requested Changes:**

The notion of "difficulty levels" needs to be justified for its circular logic, experiments with "mathematical difficulty levels" Vs performance should be considered. (critical)

There are typos in the paper (such as Laderboard Vs leaderboard, etc.) (minor)

---

> ### Author Response · Authors · 2025-12-16
> **Response from Authors**
>
> We sincerely appreciate the reviewers for their valuable feedback and insightful comments. We hope our following answers will clear up the doubts about our work, and please let us know if there is any other clarification we can provide.
>
> ---
>
> **1. Response to Difficulty Level Concerns**
>
> We want to clarify our approach to determining difficulty levels in NPPC. Our methodology uses a two-stage process:
>
> 1) Human-defined difficulty parameters: We start with interpretable problem parameters. For SAT, this means specifying (num_variables, num_clauses) such as (3, 5), (4, 5), or (100, 100). While we can confidently say (100, 100) is harder than (5, 5), distinguishing between similar configurations like (5, 4) versus (4, 5) is non-trivial.
>
> 2) LLM-based calibration: We use model performance for two specific purposes: i) Fine-grained ordering: Sorting problems of similar complexity for clearer visualization (e.g., Figure 5), particularly when human intuition cannot definitively rank them. Ii) Range validation: Ensuring difficulty levels fall within a meaningful range, i.e., avoiding settings where all models achieve 100% (too easy) or 0% (too hard) accuracy, as neither scenario effectively benchmarks capabilities.
>
> We want to note that this is not circular reasoning. We are not using LLM performance to define what makes problems hard; rather, we use it as a practical tool for (1) ordering problems within human-defined difficulty ranges and (2) validating that our chosen parameter ranges enable effective differentiation between models. We carefully revise Section 4.1 and Appendix A.6 to clearly explain the process of determining the difficulty levels to avoid any confusion.
>
> ---
>
> **2. On the value of difficult problems**
>
> Beyond simply decreasing the performance, we want to highlight the two primary objectives of NPPC:
> 1. Future-proofing benchmarks: As LLMs become increasingly capable, having appropriately difficult problems is essential for distinguishing between state-of-the-art models and tracking progress.
> 2. Revealing failure modes: Lower performance on harder instances exposes systematic weaknesses in LLM reasoning (as discussed in Section 5.4), providing actionable insights for improving model architectures and training approaches.
>
> ---
>
> **3. Regarding typos**
>
> We apologize for the grammatical errors in the original submission. We have carefully reviewed the manuscript and corrected all identified issues.

---

### Author Response · Authors · 2025-12-16
**General Response**

We thank the reviewers for their valuable and insightful review. Additional to the separate response, we would like to summarize the main changes (highlighted as blue)  in the revised manuscript:

1.	A revision of the scaling of coverage (Section 1 Introduction)

2.	More clarification of the determining of difficulty levels (Section 4.1Problem Suite: npgym)

3.	Brief overview of the experiments (Section 5 Results)

4.	Revision of the analysis of the solution errors (Section 5.3 Analysis of Solution Errors)

5.	Discussion about the reasoning data from NPPC for general reasoning (Section 6 Limitations and Future Work)

6.	Discussion of the real-world relevance of large-scale NPC problems (Appendix A.2 Why Focusing on NP (Specifically NPC) Problems?)

7.	Discussion of the influence of tool use on NPPC (Appendix A.5 Can Tool Use Crush NPPC?)

8.	More layman explanation of determining the difficulty levels (Appendix A.6 Determining the Difficulty Levels)

9.	Correlation between NPPC and other benchmarks (Appendix A.9 Correlation of Performance between NPPC and Other Benchmarks)

10.	Broader Impact Statement (Appendix A.10)

11.	More explanation of the response parsing template (Appendix D Prompts and Responses)

12.	A thorough analysis of the errors of QwQ-32B (Appendix M Thorough Error Analysis of QwQ-32B)

We sincerely hope that this thorough revision of the manuscript effectively addresses all of the reviewers' concerns and significantly strengthens the paper for publication. We deeply appreciate the reviewers' thoughtful feedback, which has been invaluable in improving our work. We remain fully open to addressing any additional concerns or suggestions the reviewers may have, and we welcome further discussion to ensure the highest quality of our manuscript.

---

### Decision · Action_Editor_Qx7k · 2026-01-14

**Recommendation:** Accept as is

**Audience:**

Yes

**Audience Explanation:**

Reasoning is a very important capability for current LLMs and the paper focuses on two main gaps of the current benchmarks : i) these benchmarks can be crushed in a short time (less than 1 year), and ii) these benchmarks may be easily hacked.

**Claims And Evidence:**

Yes

**Claims Explanation:**

The authors have addressed all concerns and questions raised by the reviewers during the rebuttal phase.